# Extension of correlation coefficient based TOPSIS technique for interval-valued Pythagorean fuzzy soft set: A case study in extract, transform, and load techniques

**Rana Muhammad Zulqarnain**[1], **Imran Siddique**[2], **Muhammad Asif**[3], **Hijaz Ahmad**[4,5], **Sameh Askar**[6], **Shahid Hussain Gurmani**[1] *

**1** School of Mathematical Sciences, Zhejiang Normal University, Jinhua, Zhejiang, China, **2** Department of Mathematics, University of Management and Technology, Lahore, Pakistan, **3** Department of Mathematics, University of Management and Technology, Sialkot Campus, Sialkot, Pakistan, **4** Section of Mathematics, International Telematic University Uninettuno, Corso Vittorio Emanuele II, Roma, Italy, **5** Operational Research Center in Healthcare, Near East University, Nicosia, Turkey, **6** Department of Statistics and Operations Research, College of Science, King Saud University, Riyadh, Saudi Arabia

* shahidgurmani07@gmail.com

**Data Availability Statement:** All relevant data are within the manuscript.

## Abstract

Correlation is an essential statistical concept for analyzing two dissimilar variables' relationships. Although the correlation coefficient is a well-known indicator, it has not been applied to interval-valued Pythagorean fuzzy soft sets (IVPFSS) data. IVPFSS is a generalized form of interval-valued intuitionistic fuzzy soft sets and a refined extension of Pythagorean fuzzy soft sets. In this study, we propose the correlation coefficient (CC) and weighted correlation coefficient (WCC) for IVPFSS and examine their necessary properties. Based on the proposed correlation measures, we develop a prioritization technique for order preference by similarity to the ideal solution (TOPSIS). We use the Extract, Transform, and Load (ETL) software selection as an example to demonstrate the application of these measures and construct a prioritization technique for order preference by similarity to the ideal solution (TOPSIS) model. The method investigates the challenge of optimizing ETL software selection for business intelligence (BI). This study offers to illuminate the significance of using correlation measures to make decisions in uncertain and complex settings. The multi-attribute decision-making (MADM) approach is a powerful instrument with many applications. This expansion is predicted to conclude in a more reliable decision-making structure. Using a sensitivity analysis, we contributed empirical studies to determine the most significant decision processes. The proposed algorithm's productivity is more consistent than prevalent models in controlling the adequate conformations of the anticipated study. Therefore, this research is expected to contribute significantly to statistics and decision-making.

**Funding:** Authors received the funding for this research from Research Supporting Project number (RSP2023R167), King Saud University, Riyadh, Saudi Arabia.

**Competing interests:** The authors have declared that no competing interests exist

## 1. Introduction

Multi-criteria decision-making (MCDM) is a technique used for decision-making when there are multiple measures to consider. MCDM can be divided into multi-attribute decision-making (MADM) and multi-objective decision-making (MODM). MADM approaches are used to select the most desirable option based on various attributes, while MODM methods optimize the design of different options based on the decision maker's multiple objectives. MADM plans have many applications, including transportation, logistics, urban planning, health, environmental safety, marketing, finance, education, economics, energy management, and water resource management. MADM techniques can be classified into various approaches such as AHP, weighted sum method (WSM), weighted product model (WPM), a technique for order preference by similarity to ideal solution (TOPSIS), modified TOPSIS, fuzzy approach, Elimination Et Choice Translating Reality (ELECTRE), VIseKriterijumska Optimizacija I Kompromisno Resenj (VIKOR), among others, as described by Das and Srinivas [1]. MADM is a field of study that focuses on developing methodologies and approaches for making decisions that involve multiple criteria or attributes. These techniques help decision-makers evaluate and compare alternatives based on various factors and select the best option. Numerous researchers have explored its plans, and their outcomes have been effectively employed to address a variety of decision-making (DM) [2–4]. The selection of BI tools is a critical factor that can significantly impact the success of a business. Among these tools, selecting ETL software is an important research topic that needs to be addressed appropriately. We have developed a practical approach for evaluating and selecting ETL software to address this issue.

In statistics and industrial engineering, a correlation coefficient is a significant tool to assess the relationship between two variables. Despite the widespread use of probabilistic methods to address practical industrial issues, they encounter difficulties when handling vast amounts of arbitrary data. In many cases, the imprecise nature of composite structures makes it difficult to achieve accurate probability relationships. Furthermore, statistical information may not always be sufficient for practical use. As a result, outcomes based on the probability model may not always be useful for professionals. Therefore, probabilistic techniques may not adequately account for the inherent vagueness of the data. MADM has been identified as a valuable approach to identifying appropriate alternatives in uncertain or imperfect information. By considering representative objectives and constraints, a comprehensive assessment can be achieved. However, in situations where objectives and boundaries are unclear, fuzzy mathematical models, such as fuzzy sets (FS) [5] and interval-valued FS (IVFS) [6], can handle vague and uncertain data in decision-making processes. Ansari et al. [7] developed a robust precautionary engineering technique for the construction of dependable software for healthcare. Ashtiani et al. [8] expanded the use of the TOPSIS method in the IVFS context to address challenges in multiple criteria decision-making. While existing FS and IVFS have limitations, intuitionistic fuzzy sets (IFS) [9] and interval-valued IFS (IVIFS) [10] have been developed to overcome these limitations. Despite these advancements, the current IFS cannot handle inconsistent and conflicting data when a team of expert's membership degree (MD) and non-membership degree (NMD) values exceed a total value of 1. The conventional FS and IVFS approaches cannot adequately address the complex nature of MD and NMD in DM assessment. Rouyendegh et al. [11] applied the TOPSIS method for IFS to solve MCDM problems in green supply chain management. Mahmood et al. [12] developed several aggregation operators (AOs) for complex intuitionistic fuzzy sets and their useful consequences. Mahmood and Ali [13] introduced the Aczel–Alsina power AOs for complex intuitionistic fuzzy settings and established a MADM technique. A centroid approach was proposed by Hung and Wu [14] for computing the correlation coefficient (CC) of intuitionistic fuzzy sets (IFS), which was then

extended to interval-valued IFS (IVIFS). The CC for IVIFS was demonstrated by Bustince and Burillo [15], who also presented proposals for its decomposition. Decomposition theorems for correlation coefficients in intuitionistic fuzzy sets (IFS) and interval-valued intuitionistic fuzzy sets (IVIFS) were proposed by Mitchell [16] and Hong [17], respectively. To solve MADM problems, Zhang and Yu [18] proposed a TOPSIS technique for IVIFS. Shi et al. [19] developed the Aczel–Alsina power AOs for interval-valued Atanassov-intuitionistic fuzzy sets. Jana et al. [20] presented hybrid Dombi aggregation operators (AOs) for IFS and utilized these operators to develop a MADM technique.

Yager [21] extended the idea of Pythagorean fuzzy sets (PFS) and used it in decision-making complications. It was developed to report the confines of existing FS theories in handling inconsistent and uncertain data. The basic condition $\mathcal{T} + \mathcal{J} \leq 1$ was revised to $\mathcal{T}^2 + \mathcal{J}^2 \leq 1$ to correct these errors. Subsequent researchers expanded on the PFS theory, with Rahman et al. [22] proposing Einstein-weighted geometric AOs for multi-attribute group decision-making (MAGDM) and Wei and Lu [23] introducing power aggregation operators for MADM. To evaluate risk assessments in failure modes and effects analysis, Akram et al. [24] constructed two improved methods for analyzing Pythagorean fuzzy sets: the Pythagorean fuzzy hybrid TOPSIS method and the Pythagorean fuzzy hybrid ELimination and Choice Translating REality I (ELECTRE I) method. Wang and Li [25] investigated interactions between Pythagorean fuzzy numbers (PFNs) and power Bonferroni mean operators. Khan et al. [26] delivered a novel dissimilarity measure for PFS and refined the VIKOR method. To address the MADM concern, Zhan et al. [27] proposed two different Pythagorean fuzzy TOPSIS approaches. Zhang [28] proposed a DM method using similarity measures to address MCGDM obstacles in PFS scenarios. Akram et al. [29] presented an ELECTRE-I methodology in a hesitant Pythagorean fuzzy environment to measure and rank risk. Peng and Yang [30] extended the PFS theory to include interval-valued Pythagorean fuzzy sets (IVPFS), proposing a DM system based on their method. Rahman et al. [31] expanded on this by developing a DM method using weighted geometric AOs for IVPFS. However, these methods have limitations in dealing with uncertainties and vagueness in parametric chemistry. The structures above are insufficient when handling the parametric values of alternatives. To address this shortcoming and tackle problems that are ambiguous, obscure, or equivocal, Molodtsov [32] introduced soft sets (SS), a general mathematical tool. Additionally, Maji et al. [33] combined fuzzy sets (FS) and SS to create fuzzy soft sets (FSS), which were later extended to intuitionistic fuzzy soft sets (IFSS) [34] with essential operations and properties. Santos-García et al. [35] introduced ranked soft sets as an innovative concept of imprecise expertise and efficient soft set development. Garg and Arora [36] protracted the TOPSIS technique based on CC to resolve MADM complexities. Jiang et al. [37] prolonged the IFSS to interval-valued IFSS (IVIFSS) and presented its fundamental operations. Ma et al. [38] utilized the choice and score values and developed an innovative DM method for IVIFSS. Khan et al. [39] developed a MADM approach for generalized IVIFSS with basic operations to address realistic complications. Zulqarnain et al. [40] established the TOPSIS technique and AOs for IVIFSS to deal with MADM issues. Akram and Adeel [41] proposed the interval-valued hesitant fuzzy N-soft sets and extended the TOPSIS approach to resolving MAGDM complications. Garg and Arora [42] offered a nonlinear-programming to determine MADM convolutions under IVIFSS.

Recently, there has been significant attention on soft sets' wide-ranging applications and extensions. To address this, Peng et al. [43] proposed Pythagorean fuzzy soft sets (PFSSs), which combine two existing theories, PFS and SS, with basic operations that possess desirable characteristics. Athira et al. [44] developed entropy and distance measures for PFSSs and utilized them for decision-making. They also developed new AOs [45] for PFSS that offer more

flexibility than IFSS or FSS. Naeem et al. [46] extended the TOPSIS and VIKOR techniques for linguistic PFSS information in the stock exchange investment domain. Riaz et al. [47, 48] developed the TOPSIS technique for m polar PFSS and introduced a similarity measure. Han et al. [49] improved the TOPSIS approach for solving MAGDM problems using PFSS information. Hua et al. [50] extended PFSS to possibility PFSS and introduced a similarity measure for comparing any two possibility PFSS. To solve complex real-life problems, Zulqarnain et al. [51] extended the Einstein-ordered operational laws for PFSS and introduced the Einstein-ordered weighted ordered geometric aggregation operator for PFSS. They also established a MAGDM technique that utilizes this operator. In addition, Zulqarnain et al. [52] proposed the Einstein-ordered weighted average aggregation operator for PFSS and developed a DM technique based on their operator. In another study, Zulqarnain et al. [53] applied the CC to settle the TOPSIS method for PFSS and developed a MADM approach to resolve DM obstacles. The extension of PFSS to IVPFSS is necessary to deal with situations where the decision-maker cannot provide a precise value for the MD and NMD. In such cases, the decision-maker may only be able to provide an interval of possible values for the MD and NMD. Therefore, IVPFSS allows for a more realistic representation of uncertainty and imprecision in DM processes. Moreover, IVPFSS can be used in DM problems involving interval-valued data, common in real-world applications such as finance, economics, and engineering. Zulqarnain et al. [54] settled AOs for IVPFSS and presented a MAGDM approach for solving real-world problems.

## 1.1 Motivation

Interval-valued Pythagorean fuzzy soft sets have become crucial in DM, particularly for dealing with incomplete information and uncertainty. IVPFSS combines the benefits of both SS and IVPFS, making it a potent method for dealing with hesitancy, inconsistencies, and inadequate data. Significant progress has been made in utilizing IVPFSS to address these challenges in recent years. Although the TOPSIS method is crucial for solving DM problems, the CC has not been used in the hybridizing SS and IVPFS literature. This means the prevalent TOPSIS technique for IVIFSS [40] cannot handle situations when the $MD+NMD>1$. Similarly, the correlation-based TOPSIS method for PFSS [53] cannot measure composite IVPFSN or intentionally associate with MD and NMD. Moreover, the model's results are limited, and the presentation partiality of substitutes cannot be resolute. Given these limitations, there is a strong motivation to develop a more capable approach that can handle several specialties of alternatives in interval form. We propose a solution to address the limitations of existing DM techniques when handling IVPFSS. Our approach involves introducing CC and WCC measures explicitly intended for IVPFSS, which allows for ranking preferences based on their similarity to the ideal solution. We then use the developed correlation measures to apply the TOPSIS method to MADM problems. This approach outperforms existing TOPSIS techniques, as it is tailored to the unique challenges of IVPFSS. We demonstrate the effectiveness of our methodology over a numerical case in point and present a comparative analysis to endorse its realism and effectiveness. Our contribution is a novel DM strategy for IVPFSS that is more consistent than prevalent replicas.

## 1.2 Contribution

Recent research in decision-making (DM) has emphasized the need to account for uncertainty and incomplete information in DM protocols. This is because real-world problems often involve imprecise or ambiguous data, making it difficult to make informed decisions. One significant tool for addressing these contests is IVPFSS, which combines the advantages of both SS and IVPFS. In the context of DM protocols using IVPFSS, the frequently used CC measure may not perfectly replicate the viable assessment of alternatives, as it fails to interpret for

assured aspects. IVPFSS is a systematic technique most beneficial for handling hesitancy, inconsistencies, and imperfect data. To address this issue, this research aims to introduce new CC and weighted CC (WCC) measures based on imprecise information in the context of IVPFSS. The objectives of this study are as follows:

⁜ One of the contributions of this study is the development of a framework that enables the comprehension of the informational energies present in IVPFSS settings. These energies denote the quantity of information in an FS, and their understanding is crucial for creating reliable CC and WCC measures.

⁜ The study employs informational energies and correlation measures to create novel CC and WCC measures for IVPFSS. These measures consider the imprecise information in IVPFSS and offer a more precise assessment of the reasonable value of substitutes in DM practices.

⁜ The study aims to improve the effectiveness and robustness of the TOPSIS technique for DM complications, including multiple features, by introducing a modified version incorporating the settled CC and WCC. This approach will provide a more accurate reflection of the competitive value of alternatives, taking into account the imprecise information in IVPFSS.

⁜ Applying the TOPSIS method to explain MADM complications, evaluating DM distraction and ETL software selection, and selecting the most practical for benchmarking will substantially provide the hypothetical structure of FS in DM.

⁜ Perform comparative analyses to assess the effectiveness of the proposed methodology compared to prevailing methodologies. These studies will determine the strengths and robustness of the developed TOPSIS approach and its ability to perform other methods in solving MADM problems.

This research is organized into seven sections, each with a specific objective and contribution. The first section introduces the importance of accounting for uncertainty and incomplete information in decision-making processes. This section also highlights the limitations of the commonly used CC measure in addressing DM problems. Section 2 overviews the basic concepts and principles underpinning the organizational development follow-up study. This section sets the stage for the rest of the proposal by providing a foundation for understanding the complexities of DM problems and the need for a more reliable and accurate approach. The concept of informational energies is introduced, and their application to CC measures for IVPFSS is discussed in section 3. The essential properties of this approach are explained, and how it enhances the accuracy and reliability of DM processes involving uncertainty and incomplete information in the same section. The WCC is introduced, and its critical properties are discussed, further refining DM processes' accuracy and reliability in the IVPFSS context in section 4. Section 5 comprises the correlation-based TOPSIS technique used to solve MADM complications. A numerical exploration is conducted to demonstrate the practicality of the proposed model in section 6. This exploration identifies the most suitable ETL software selection and provides a real-world example of how the proposed model can be applied to solve practical DM problems. In the seventh section, a comparative analysis is conducted to ensure the pragmatism of the proposed model. This analysis compares the proposed approach with existing models and demonstrates the superiority of the proposed model in terms of accuracy and reliability.

## 2. Preliminaries

In the following section, we will discuss some basic notions to construct the structure of the subsequent research.

**Definition 2.1** [6]

An interval-valued fuzzy set $\mathcal{A}$ in a universe of discourse $U$ is defined as:

$$\mathcal{A} = \{(\mathfrak{u}_i, \mathcal{T}_{\mathcal{A}_j}(\mathfrak{u}_i)) | \mathfrak{u}_i \in U\}$$

Where, $\mathcal{T}_{\mathcal{A}_j}(\mathfrak{u}_i) = [\mathcal{T}^{\ell}_{\mathcal{A}_j}(\mathfrak{u}_i), \mathcal{T}^{\mho}_{\mathcal{A}_j}(\mathfrak{u}_i)]$ be the MD interval, and $\mathcal{T}^{\ell}_{\mathcal{A}_j}(\mathfrak{u}_i), \mathcal{T}^{\mho}_{\mathcal{A}_j}(\mathfrak{u}_i)$ indicating the uncertainty or imprecision in the MD interval.

**Definition 2.2** [10]

An interval-valued intuitionistic fuzzy set $\mathcal{A}$ in a universe of discourse $U$ is defined as:

$$\mathcal{A} = \{(\mathfrak{u}_i, (\mathcal{T}_{\mathcal{A}_j}(\mathfrak{u}_i), \mathcal{J}_{\mathcal{A}_j}(\mathfrak{u}_i))) | \mathfrak{u}_i \in U\}$$

Where, $\mathcal{T}_{\mathcal{A}_j}(\mathfrak{u}_i) = [\mathcal{T}^{\ell}_{\mathcal{A}_j}(\mathfrak{u}_i), \mathcal{T}^{\mho}_{\mathcal{A}_j}(\mathfrak{u}_i)]$ and $\mathcal{J}_{\mathcal{A}_j}(\mathfrak{u}_i) = [\mathcal{J}^{\ell}_{\mathcal{A}_j}(\mathfrak{u}_i), \mathcal{J}^{\mho}_{\mathcal{A}_j}(\mathfrak{u}_i)]$ be the MD and NMD intervals. Also, $[\mathcal{T}^{\ell}_{\mathcal{A}_j}(\mathfrak{u}_i), \mathcal{T}^{\mho}_{\mathcal{A}_j}(\mathfrak{u}_i)] \subseteq [0,1]$ and $[\mathcal{J}^{\ell}_{\mathcal{A}_j}(\mathfrak{u}_i), \mathcal{J}^{\mho}_{\mathcal{A}_j}(\mathfrak{u}_i)] \subseteq [0,1]$, $0 \leq \mathcal{T}^{\ell}_{\mathcal{A}_j}(\mathfrak{u}_i)$, $\mathcal{T}^{\mho}_{\mathcal{A}_j}(\mathfrak{u}_i), \mathcal{J}^{\ell}_{\mathcal{A}_j}(\mathfrak{u}_i), \mathcal{J}^{\mho}_{\mathcal{A}_j}(\mathfrak{u}_i) \leq 1$, such as $0 \leq \mathcal{T}^{\mho}_{\mathcal{A}_j}(\mathfrak{u}_i) + \mathcal{J}^{\mho}_{\mathcal{A}_j}(\mathfrak{u}_i) \leq 1$.

**Definition 2.3** [30]

An interval-valued Pythagorean fuzzy set $\mathcal{A}$ in a universe of discourse $U$ is defined as:

$$\mathcal{A} = \{(\mathfrak{u}_i, (\mathcal{T}_{\mathcal{A}_j}(\mathfrak{u}_i), \mathcal{J}_{\mathcal{A}_j}(\mathfrak{u}_i))) | \mathfrak{u}_i \in U\}$$

Where, $\mathcal{T}_{\mathcal{A}_j}(\mathfrak{u}_i) = [\mathcal{T}^{\ell}_{\mathcal{A}_j}(\mathfrak{u}_i), \mathcal{T}^{\mho}_{\mathcal{A}_j}(\mathfrak{u}_i)]$ and $\mathcal{J}_{\mathcal{A}_j}(\mathfrak{u}_i) = [\mathcal{J}^{\ell}_{\mathcal{A}_j}(\mathfrak{u}_i), \mathcal{J}^{\mho}_{\mathcal{A}_j}(\mathfrak{u}_i)]$ be the MD and NMD intervals. Also, $[\mathcal{T}^{\ell}_{\mathcal{A}_j}(\mathfrak{u}_i), \mathcal{T}^{\mho}_{\mathcal{A}_j}(\mathfrak{u}_i)] \subseteq [0,1]$ and $[\mathcal{J}^{\ell}_{\mathcal{A}_j}(\mathfrak{u}_i), \mathcal{J}^{\mho}_{\mathcal{A}_j}(\mathfrak{u}_i)] \subseteq [0,1]$, $0 \leq \mathcal{T}^{\ell}_{\mathcal{A}_j}(\mathfrak{u}_i)$, $\mathcal{T}^{\mho}_{\mathcal{A}_j}(\mathfrak{u}_i), \mathcal{J}^{\ell}_{\mathcal{A}_j}(\mathfrak{u}_i), \mathcal{J}^{\mho}_{\mathcal{A}_j}(\mathfrak{u}_i) \leq 1$, such as $0 \leq (\mathcal{T}^{\mho}_{\mathcal{A}_j}(\mathfrak{u}_i))^2 + (\mathcal{J}^{\mho}_{\mathcal{A}_j}(\mathfrak{u}_i))^2 \leq 1$.

**Definition 2.4** [32]

Let $U$ and $\varsigma$ be the universe of discourse and set of attributes, $\mathcal{P}(U)$ be the power set of $U$ and $\mathcal{A} \subseteq \varsigma$. Then, a pair $(\mathcal{F}, \mathcal{A})$ is called a soft set over $U$, where $\mathcal{F}$ is a mapping:

$$\mathcal{F} : \mathcal{A} \to \mathcal{P}(U)$$

Also, it can be defined as follows:

$$(\mathcal{F}, \mathcal{A}) = \{\mathcal{F}(\varsigma) \in \mathcal{P}(U) : \varsigma \in \varsigma, \mathcal{F}(\varsigma) = \emptyset \; if \; \varsigma \notin \mathcal{A}\}$$

**Definition 2.5** [37]

Let $U$ and $\varsigma$ be the universe of discourse and set of attributes, $\mathcal{P}(U)$ be the power set of $U$ and $\mathcal{A} \subseteq \varsigma$. Then, a pair $(\mathcal{F}, \mathcal{A})$ is called an IVIFSS over $U$.

$$(\mathcal{F}, \mathcal{A}) = \{(\mathfrak{u}_i, (\mathcal{T}_{\mathcal{A}_j}(\mathfrak{u}_i), \mathcal{J}_{\mathcal{A}_j}(\mathfrak{u}_i))) | \mathfrak{u}_i \in U\}$$

Where $\mathcal{F} : \mathcal{A} \to \mathcal{P}(U)$ be a mapping between a set of attributes and a power set of $U$. Also, $\mathcal{T}_{\mathcal{A}_j}(\mathfrak{u}_i) = [\mathcal{T}^{\ell}_{\mathcal{A}_j}(\mathfrak{u}_i), \mathcal{T}^{\mho}_{\mathcal{A}_j}(\mathfrak{u}_i)]$ and $\mathcal{J}_{\mathcal{A}_j}(\mathfrak{u}_i) = [\mathcal{J}^{\ell}_{\mathcal{A}_j}(\mathfrak{u}_i), \mathcal{J}^{\mho}_{\mathcal{A}_j}(\mathfrak{u}_i)]$ be the MD and NMD intervals, such as $[\mathcal{T}^{\ell}_{\mathcal{A}_j}(\mathfrak{u}_i), \mathcal{T}^{\mho}_{\mathcal{A}_j}(\mathfrak{u}_i)] \subseteq [0,1]$ and $[\mathcal{J}^{\ell}_{\mathcal{A}_j}(\mathfrak{u}_i), \mathcal{J}^{\mho}_{\mathcal{A}_j}(\mathfrak{u}_i)] \subseteq [0,1]$, $0 \leq \mathcal{T}^{\ell}_{\mathcal{A}_j}(\mathfrak{u}_i), \mathcal{T}^{\mho}_{\mathcal{A}_j}(\mathfrak{u}_i)$, $\mathcal{J}^{\ell}_{\mathcal{A}_j}(\mathfrak{u}_i), \mathcal{J}^{\mho}_{\mathcal{A}_j}(\mathfrak{u}_i) \leq 1$, and $0 \leq \mathcal{T}^{\mho}_{\mathcal{A}_j}(\mathfrak{u}_i) + \mathcal{J}^{\mho}_{\mathcal{A}_j}(\mathfrak{u}_i) \leq 1$.

**Definition 2.6** [54]

Let $U$ and $\varsigma$ be the universe of discourse and set of attributes, $\mathcal{P}(U)$ be the power set of $U$ and $\mathcal{A} \subseteq \varsigma$. Then, a pair $(\mathcal{F}, \mathcal{A})$ is called an IVPFSS over $U$.

$$(\mathcal{F}, \mathcal{A}) = \{(\mathfrak{u}_i, (\mathcal{T}_{\mathcal{A}_j}(\mathfrak{u}_i), \mathcal{J}_{\mathcal{A}_j}(\mathfrak{u}_i))) | \mathfrak{u}_i \in U\}$$

Where $\mathcal{F} : \mathcal{A} \to \mathcal{P}(U)$ be a mapping between a set of attributes and a power set of $U$. Also,

$\mathcal{T}_{\mathcal{A}_j}(\mathfrak{u}_i) = [\mathcal{T}^{\ell}_{\mathcal{A}_j}(\mathfrak{u}_i), \mathcal{T}^{\mho}_{\mathcal{A}_j}(\mathfrak{u}_i)]$ and $\mathcal{J}_{\mathcal{A}_j}(\mathfrak{u}_i) = [\mathcal{J}^{\ell}_{\mathcal{A}_j}(\mathfrak{u}_i), \mathcal{J}^{\mho}_{\mathcal{A}_j}(\mathfrak{u}_i)]$ be the MD and NMD intervals, such as $[\mathcal{T}^{\ell}_{\mathcal{A}_j}(\mathfrak{u}_i), \mathcal{T}^{\mho}_{\mathcal{A}_j}(\mathfrak{u}_i)] \subseteq [0, 1]$ and $[\mathcal{J}^{\ell}_{\mathcal{A}_j}(\mathfrak{u}_i), \mathcal{J}^{\mho}_{\mathcal{A}_j}(\mathfrak{u}_i)] \subseteq [0, 1], 0 \le \mathcal{T}^{\ell}_{\mathcal{A}_j}(\mathfrak{u}_i), \mathcal{T}^{\mho}_{\mathcal{A}_j}(\mathfrak{u}_i),$ $\mathcal{J}^{\ell}_{\mathcal{A}_j}(\mathfrak{u}_i), \mathcal{J}^{\mho}_{\mathcal{A}_j}(\mathfrak{u}_i) \le 1$, and $0 \le (\mathcal{T}^{\mho}_{\mathcal{A}_j}(\mathfrak{u}_i))^2 + (\mathcal{J}^{\mho}_{\mathcal{A}_j}(\mathfrak{u}_i))^2 \le 1$. For readers convience, $\mathcal{F}_{\mathcal{A}_j}(\mathfrak{u}_i) = \{(\mathfrak{u}_i, ([\mathcal{T}^{\ell}_{\mathcal{A}_j}(\mathfrak{u}_i), \mathcal{T}^{\mho}_{\mathcal{A}_j}(\mathfrak{u}_i)], [\mathcal{J}^{\ell}_{\mathcal{A}_j}(\mathfrak{u}_i), \mathcal{J}^{\mho}_{\mathcal{A}_j}(\mathfrak{u}_i)]))|\mathfrak{u}_i \in U\}$ can be written as $\mathcal{F}_{\mathcal{A}} = ([\mathcal{T}^{\ell}_{\mathcal{A}}, \mathcal{T}^{\mho}_{\mathcal{A}}], [\mathcal{J}^{\ell}_{\mathcal{A}}, \mathcal{J}^{\mho}_{\mathcal{A}}])$ known as interval-valued Pythagorean fuzzy soft number (IVPFSN). In the process of applying IVPFSNs in actual problems, it is essential to rank them. For this, the scoring function of $\mathcal{F}_{\mathcal{A}}$ is defined as follows:

$$S(\mathcal{F}_{\mathcal{A}}) = \frac{(\mathcal{T}^{\ell}_{\mathcal{A}})^2 + (\mathcal{T}^{\mho}_{\mathcal{A}})^2 - (\mathcal{J}^{\ell}_{\mathcal{A}})^2 - (\mathcal{J}^{\mho}_{\mathcal{A}})^2}{2}$$

$$A(\mathcal{F}_{\mathcal{A}}) = \frac{(\mathcal{T}^{\ell}_{\mathcal{A}})^2 + (\mathcal{T}^{\mho}_{\mathcal{A}})^2 + (\mathcal{J}^{\ell}_{\mathcal{A}})^2 + (\mathcal{J}^{\mho}_{\mathcal{A}})^2}{2}$$

**Definition 2.7** [54]

Let $\mathcal{F}_{\varsigma} = ([\mathcal{T}^{\ell}, \mathcal{T}^{\mho}], [\mathcal{J}^{\ell}, \mathcal{J}^{\mho}]), \mathcal{F}_{\varsigma_{11}} = ([\mathcal{T}^{\ell}_{\varsigma_{11}}, \mathcal{T}^{\mho}_{\varsigma_{11}}], [\mathcal{J}^{\ell}_{\varsigma_{11}}, \mathcal{J}^{\mho}_{\varsigma_{11}}])$, and $\mathcal{F}_{\varsigma_{12}} = ([\mathcal{T}^{\ell}_{\varsigma_{12}}, \mathcal{T}^{\mho}_{\varsigma_{12}}], [\mathcal{J}^{\ell}_{\varsigma_{12}}, \mathcal{J}^{\mho}_{\varsigma_{12}}])$ be the IVPFSNs and $\beta > 0$. Then, the algebraic operational laws for IVPFSNs are given:

1) $\mathcal{F}_{\varsigma_{11}} \oplus \mathcal{F}_{\varsigma_{12}} = ([\sqrt{(\mathcal{T}^{\ell}_{\varsigma_{11}})^2 + (\mathcal{T}^{\ell}_{\varsigma_{12}})^2 - (\mathcal{T}^{\ell}_{\varsigma_{11}})^2 (\mathcal{T}^{\ell}_{\varsigma_{12}})^2},$
$\sqrt{(\mathcal{T}^{\mho}_{\varsigma_{11}})^2 + (\mathcal{T}^{\mho}_{\varsigma_{12}})^2 - (\mathcal{T}^{\mho}_{\varsigma_{11}})^2 (\mathcal{T}^{\mho}_{\varsigma_{12}})^2}], [\mathcal{J}^{\ell}_{\varsigma_{11}} \mathcal{J}^{\ell}_{\varsigma_{12}}, \mathcal{J}^{\mho}_{\varsigma_{11}} \mathcal{J}^{\mho}_{\varsigma_{12}}])$

2) $\mathcal{F}_{\varsigma_{11}} \otimes \mathcal{F}_{\varsigma_{12}} = ([\mathcal{T}^{\ell}_{\varsigma_{11}} \mathcal{T}^{\ell}_{\varsigma_{12}}, \mathcal{T}^{\mho}_{\varsigma_{11}} \mathcal{T}^{\mho}_{\varsigma_{12}}], [\sqrt{(\mathcal{J}^{\ell}_{\varsigma_{11}})^2 + (\mathcal{J}^{\ell}_{\varsigma_{12}})^2 - (\mathcal{J}^{\ell}_{\varsigma_{11}})^2 (\mathcal{J}^{\ell}_{\varsigma_{12}})^2},$
$\sqrt{(\mathcal{J}^{\mho}_{\varsigma_{11}})^2 + (\mathcal{J}^{\mho}_{\varsigma_{12}})^2 - (\mathcal{J}^{\mho}_{\varsigma_{11}})^2 (\mathcal{J}^{\mho}_{\varsigma_{12}})^2}])$

3) $\beta \mathcal{F}_{\varsigma} = ([\sqrt{1 - (1 - (\mathcal{T}^{\ell})^2)^{\beta}}, \sqrt{1 - (1 - (\mathcal{T}^{\mho})^2)^{\beta}}], [(\mathcal{J}^{\ell})^{\beta}, (\mathcal{J}^{\mho})^{\beta}]) =$
$(\sqrt{1 - (1 - [\mathcal{T}^{\ell}, \mathcal{T}^{\mho}]^2)^{\beta}}, [(\mathcal{J}^{\ell})^{\beta}, (\mathcal{J}^{\mho})^{\beta}])$

4) $\mathcal{F}^{\beta}_{\varsigma} = ([(\mathcal{T}^{\ell})^{\beta}, (\mathcal{T}^{\mho})^{\beta}], [\sqrt{1 - (1 - (\mathcal{J}^{\ell})^2)^{\beta}}, \sqrt{1 - (1 - (\mathcal{J}^{\mho})^2)^{\beta}}]) =$
$([(\mathcal{T}^{\ell})^{\beta}, (\mathcal{T}^{\mho})^{\beta}], \sqrt{1 - (1 - [\mathcal{J}^{\ell}, \mathcal{J}^{\mho}]^2)^{\beta}})$

$$\text{IVPFSWA}(\mathcal{F}_{\varsigma_{11}}, \mathcal{F}_{\varsigma_{12}}, \ldots\ldots\ldots, \mathcal{F}_{\varsigma_{nm}})$$
$$= (\sqrt{1 - \prod_{j=1}^{m} (\prod_{i=1}^{n} (1 - [\mathcal{T}^{\ell}_{ij}, \mathcal{T}^{\mho}_{ij}]^2)^{\Omega_i})^{\gamma_j}}, \prod_{j=1}^{m} (\prod_{i=1}^{n} ([\mathcal{J}^{\ell}_{ij}, \mathcal{J}^{\mho}_{ij}])^{\Omega_i})^{\gamma_j}) \qquad (2.1)$$

$$\text{IVPFSWG}(\mathcal{F}_{\varsigma_{11}}, \mathcal{F}_{\varsigma_{12}}, \ldots\ldots\ldots, \mathcal{F}_{\varsigma_{nm}})$$
$$= (\prod_{j=1}^{m} (\prod_{i=1}^{n} ([\mathcal{T}^{\ell}_{ij}, \mathcal{T}^{\mho}_{ij}])^{\Omega_i})^{\gamma_j}, \sqrt{1 - \prod_{j=1}^{m} (\prod_{i=1}^{n} (1 - [\mathcal{J}^{\ell}_{ij}, \mathcal{J}^{\mho}_{ij}]^2)^{\Omega_i})^{\gamma_j}}) \qquad (2.2)$$

The weight vectors for experts and attributes are denoted as $\Omega_i$ and $\gamma_j$ respectively, subject to the given conditions of $\Omega_i > 0, \ \sum_{i=1}^{n} \Omega_i = 1; \ \gamma_j > 0, \ \sum_{j=1}^{m} \gamma_j = 1$.

## 3. Correlation coefficient for interval valued Pythagorean fuzzy soft set

This section presents the correlation coefficient and WCC for IVPFSS, highlighting their desirable features.

**Definition 3.1.** Let $(\mathcal{F}, \mathcal{A}) = \{(\mathfrak{u}_i, (\mathcal{T}_{\mathcal{A}_j}(\mathfrak{u}_i), \mathcal{J}_{\mathcal{A}_j}(\mathfrak{u}_i)))|\mathfrak{u}_i \in U\}$ and $(\mathcal{G}, \mathcal{B}) = \{(\mathfrak{u}_i, (\mathcal{T}_{\mathcal{B}_j}(\mathfrak{u}_i), \mathcal{J}_{\mathcal{B}_j}(\mathfrak{u}_i)))|\mathfrak{u}_i \in U\}$ be two IVPFSS over a set of attributes $\varsigma = \{1, 2, 3, \ldots, m\}$, where $\mathcal{T}_{\mathcal{A}_j}(\mathfrak{u}_i) = [\mathcal{T}^\ell_{\mathcal{A}_j}(\mathfrak{u}_i), \mathcal{T}^\upsilon_{\mathcal{A}_j}(\mathfrak{u}_i)], \mathcal{J}_{\mathcal{A}_j}(\mathfrak{u}_i) = [\mathcal{J}^\ell_{\mathcal{A}_j}(\mathfrak{u}_i), \mathcal{J}^\upsilon_{\mathcal{A}_j}(\mathfrak{u}_i)], \mathcal{T}_{\mathcal{B}_j}(\mathfrak{u}_i) = [\mathcal{T}^\ell_{\mathcal{B}_j}(\mathfrak{u}_i), \mathcal{T}^\upsilon_{\mathcal{B}_j}(\mathfrak{u}_i)], \mathcal{J}_{\mathcal{B}_j}(\mathfrak{u}_i) = [\mathcal{J}^\ell_{\mathcal{B}_j}(\mathfrak{u}_i), \mathcal{J}^\upsilon_{\mathcal{B}_j}(\mathfrak{u}_i)]$. Then the informational energies of $(\mathcal{F}, \mathcal{A})$ and $(\mathcal{G}, \mathcal{B})$ are defined as:

$$\mathcal{E}_{IVPFSS}(\mathcal{F}, \mathcal{A}) = \sum_{j=1}^m \sum_{i=1}^n ((\mathcal{T}^\ell_{\mathcal{A}_j}(\mathfrak{u}_i))^4 + (\mathcal{T}^\upsilon_{\mathcal{A}_j}(\mathfrak{u}_i))^4 + (\mathcal{J}^\ell_{\mathcal{A}_j}(\mathfrak{u}_i))^4 + (\mathcal{J}^\upsilon_{\mathcal{A}_j}(\mathfrak{u}_i))^4) \quad (3.1)$$

$$\mathcal{E}_{IVPFSS}(\mathcal{G}, \mathcal{B}) = \sum_{j=1}^m \sum_{i=1}^n ((\mathcal{T}^\ell_{\mathcal{B}_j}(\mathfrak{u}_i))^4 + (\mathcal{T}^\upsilon_{\mathcal{B}_j}(\mathfrak{u}_i))^4 + (\mathcal{J}^\ell_{\mathcal{B}_j}(\mathfrak{u}_i))^4 + (\mathcal{J}^\upsilon_{\mathcal{B}_j}(\mathfrak{u}_i))^4). \quad (3.2)$$

**Definition 3.2.** Let $(\mathcal{F}, \mathcal{A}) = \{(\mathfrak{u}_i, ([\mathcal{T}^\ell_{\mathcal{A}_j}(\mathfrak{u}_i), \mathcal{T}^\upsilon_{\mathcal{A}_j}(\mathfrak{u}_i)], [\mathcal{J}^\ell_{\mathcal{A}_j}(\mathfrak{u}_i), \mathcal{J}^\upsilon_{\mathcal{A}_j}(\mathfrak{u}_i)]))|\mathfrak{u}_i \in U\}$ and $(\mathcal{G}, \mathcal{B}) = \{(\mathfrak{u}_i, ([\mathcal{T}^\ell_{\mathcal{B}_j}(\mathfrak{u}_i), \mathcal{T}^\upsilon_{\mathcal{B}_j}(\mathfrak{u}_i)], [\mathcal{J}^\ell_{\mathcal{B}_j}(\mathfrak{u}_i), \mathcal{J}^\upsilon_{\mathcal{B}_j}(\mathfrak{u}_i)]))|\mathfrak{u}_i \in U\}$ be two IVPFSS. Then, the correlation of $(\mathcal{F}, \mathcal{A})$ and $(\mathcal{G}, \mathfrak{B})$ is defined as:

$$\mathcal{C}_{IVPFSS}((\mathcal{F}, \mathcal{A}), (\mathcal{G}, \mathcal{B}))$$
$$= \sum_{j=1}^m \sum_{i=1}^n \begin{pmatrix} (\mathcal{T}^\ell_{\mathcal{A}_j}(\mathfrak{u}_i))^2 * (\mathcal{T}^\ell_{\mathcal{B}_j}(\mathfrak{u}_i))^2 + (\mathcal{T}^\upsilon_{\mathcal{A}_j}(\mathfrak{u}_i))^2 * (\mathcal{T}^\upsilon_{\mathcal{B}_j}(\mathfrak{u}_i))^2 + \\ (\mathcal{J}^\ell_{\mathcal{A}_j}(\mathfrak{u}_i))^2 * (\mathcal{J}^\ell_{\mathcal{B}_j}(\mathfrak{u}_i))^2 + (\mathcal{J}^\upsilon_{\mathcal{A}_j}(\mathfrak{u}_i))^2 * (\mathcal{J}^\upsilon_{\mathcal{B}_j}(\mathfrak{u}_i))^2 \end{pmatrix}. \quad (3.3)$$

**Proposition 1:** Let $(\mathcal{F}, \mathcal{A}) = \{(\mathfrak{u}_i, ([\mathcal{T}^\ell_{\mathcal{A}_j}(\mathfrak{u}_i), \mathcal{T}^\upsilon_{\mathcal{A}_j}(\mathfrak{u}_i)], [\mathcal{J}^\ell_{\mathcal{A}_j}(\mathfrak{u}_i), \mathcal{J}^\upsilon_{\mathcal{A}_j}(\mathfrak{u}_i)]))|\mathfrak{u}_i \in U\}$ and $(\mathcal{G}, \mathcal{B}) = \{(\mathfrak{u}_i, ([\mathcal{T}^\ell_{\mathcal{B}_j}(\mathfrak{u}_i), \mathcal{T}^\upsilon_{\mathcal{B}_j}(\mathfrak{u}_i)], [\mathcal{J}^\ell_{\mathcal{B}_j}(\mathfrak{u}_i), \mathcal{J}^\upsilon_{\mathcal{B}_j}(\mathfrak{u}_i)]))|\mathfrak{u}_i \in U\}$ be two IVPFSS. Then

1. $\mathcal{C}_{IVPFSS}((\mathcal{F}, \mathcal{A}), (\mathcal{F}, \mathcal{A})) = (\mathcal{F}, \mathcal{A})$

2. $\mathcal{C}_{IVPFSS}((\mathcal{F}, \mathcal{A}), (\mathcal{G}, \mathcal{B})) = \mathcal{C}_{IVPFSS}((\mathcal{G}, \mathcal{B}), (\mathcal{F}, \mathcal{A}))$.

**Proof:** The proof is simple and easy to follow.

**Definition 3.3.** Let $(\mathcal{F}, \mathcal{A}) = \{(\mathfrak{u}_i, ([\mathcal{T}^\ell_{\mathcal{A}_j}(\mathfrak{u}_i), \mathcal{T}^\upsilon_{\mathcal{A}_j}(\mathfrak{u}_i)], [\mathcal{J}^\ell_{\mathcal{A}_j}(\mathfrak{u}_i), \mathcal{J}^\upsilon_{\mathcal{A}_j}(\mathfrak{u}_i)]))|\mathfrak{u}_i \in U\}$ and $(\mathcal{G}, \mathcal{B}) = \{(\mathfrak{u}_i, ([\mathcal{T}^\ell_{\mathcal{B}_j}(\mathfrak{u}_i), \mathcal{T}^\upsilon_{\mathcal{B}_j}(\mathfrak{u}_i)], [\mathcal{J}^\ell_{\mathcal{B}_j}(\mathfrak{u}_i), \mathcal{J}^\upsilon_{\mathcal{B}_j}(\mathfrak{u}_i)]))|\mathfrak{u}_i \in U\}$ be two IVPFSS. Then, CC is defined as:

$$\mathbb{C}_{IVPFSS}((\mathcal{F}, \mathcal{A}), (\mathcal{G}, \mathcal{B})) = \frac{\mathcal{C}_{IVPFSS}((\mathcal{F}, \mathcal{A}), (\mathcal{G}, \mathcal{B}))}{\sqrt{\mathcal{E}_{IVPFSS}(\mathcal{F}, \mathcal{A})}\sqrt{\mathcal{E}_{IVPFSS}(\mathcal{G}, \mathcal{B})}}$$

$$= \frac{\sum_{j=1}^m \sum_{i=1}^n ((\mathcal{T}^\ell_{\mathcal{A}_j}(\mathfrak{u}_i))^2 * (\mathcal{T}^\ell_{\mathcal{B}_j}(\mathfrak{u}_i))^2 + (\mathcal{T}^\upsilon_{\mathcal{A}_j}(\mathfrak{u}_i))^2 * (\mathcal{T}^\upsilon_{\mathcal{B}_j}(\mathfrak{u}_i))^2 + (\mathcal{J}^\ell_{\mathcal{A}_j}(\mathfrak{u}_i))^2 * (\mathcal{J}^\ell_{\mathcal{B}_j}(\mathfrak{u}_i))^2 + (\mathcal{J}^\upsilon_{\mathcal{A}_j}(\mathfrak{u}_i))^2 * (\mathcal{J}^\upsilon_{\mathcal{B}_j}(\mathfrak{u}_i))^2).}{\sqrt{\sum_{j=1}^m \sum_{i=1}^n ((\mathcal{T}^\ell_{\mathcal{A}_j}(\mathfrak{u}_i))^4 + (\mathcal{T}^\upsilon_{\mathcal{A}_j}(\mathfrak{u}_i))^4 + (\mathcal{J}^\ell_{\mathcal{A}_j}(\mathfrak{u}_i))^4 + (\mathcal{J}^\upsilon_{\mathcal{A}_j}(\mathfrak{u}_i))^4)}\sqrt{\sum_{j=1}^m \sum_{i=1}^n ((\mathcal{T}^\ell_{\mathcal{B}_j}(\mathfrak{u}_i))^4 + (\mathcal{T}^\upsilon_{\mathcal{B}_j}(\mathfrak{u}_i))^4 + (\mathcal{J}^\ell_{\mathcal{B}_j}(\mathfrak{u}_i))^4 + (\mathcal{J}^\upsilon_{\mathcal{B}_j}(\mathfrak{u}_i))^4)}}. \quad (3.4)$$

**Theorem 1.** Let $(\mathcal{F}, \mathcal{A}) = \{(\mathfrak{u}_i, ([\mathcal{T}^\ell_{\mathcal{A}_j}(\mathfrak{u}_i), \mathcal{T}^\upsilon_{\mathcal{A}_j}(\mathfrak{u}_i)], [\mathcal{J}^\ell_{\mathcal{A}_j}(\mathfrak{u}_i), \mathcal{J}^\upsilon_{\mathcal{A}_j}(\mathfrak{u}_i)]))|\mathfrak{u}_i \in U\}$ and $(\mathcal{G}, \mathcal{B}) = \{(\mathfrak{u}_i, ([\mathcal{T}^\ell_{\mathcal{B}_j}(\mathfrak{u}_i), \mathcal{T}^\upsilon_{\mathcal{B}_j}(\mathfrak{u}_i)], [\mathcal{J}^\ell_{\mathcal{B}_j}(\mathfrak{u}_i), \mathcal{J}^\upsilon_{\mathcal{B}_j}(\mathfrak{u}_i)]))|\mathfrak{u}_i \in U\}$ be two IVPFSS, then the following properties are hold:

1. $0 \leq \mathbb{C}_{IVPFSS}((\mathcal{F}, \mathcal{A}), (\mathcal{G}, \mathcal{B})) \leq 1$

2. $\mathbb{C}_{IVPFSS}((\mathcal{F}, \mathcal{A}), (\mathcal{G}, \mathcal{B})) = \mathbb{C}_{IVPFSS}((\mathcal{G}, \mathcal{B}), (\mathcal{F}, \mathcal{A}))$

3. If $(\mathcal{F}, \mathcal{A}) = (\mathcal{G}, \mathcal{B})$, i.e., $\forall i, j, \mathcal{T}^{\ell}_{\mathcal{A}_j}(\mathfrak{u}_i) = \mathcal{T}^{\ell}_{\mathcal{B}_j}(\mathfrak{u}_i), \mathcal{T}^{\mho}_{\mathcal{A}_j}(\mathfrak{u}_i) = \mathcal{T}^{\mho}_{\mathcal{B}_j}(\mathfrak{u}_i), \mathcal{J}^{\ell}_{\mathcal{A}_j}(\mathfrak{u}_i) = \mathcal{J}^{\ell}_{\mathcal{B}_j}(\mathfrak{u}_i)$, and $\mathcal{J}^{\mho}_{\mathcal{A}_j}(\mathfrak{u}_i) = \mathcal{J}^{\mho}_{\mathcal{B}_j}(\mathfrak{u}_i)$, then $\mathbb{C}_{IVPFSS}((\mathcal{F}, \mathcal{A}), (\mathcal{G}, \mathcal{B})) = 1$.

**Proof 1.** $\mathbb{C}_{IVPFSS}((\mathcal{F}, \mathcal{A}), (\mathcal{G}, \mathcal{B})) \geq 0$ is obvious. Now, we will demonstrate $\mathbb{C}_{IVPFSS}((\mathcal{F}, \mathcal{A}), (\mathcal{G}, \mathcal{B})) \leq 1$. Using Eq 3.

$\mathcal{C}_{IVPFSS}((\mathcal{F}, \mathcal{A}), (\mathcal{G}, \mathcal{B}))$

$$= \sum_{j=1}^{m} \sum_{i=1}^{n} ((\mathcal{T}^{\ell}_{\mathcal{A}_j}(\mathfrak{u}_i))^2 * (\mathcal{T}^{\ell}_{\mathcal{B}_j}(\mathfrak{u}_i))^2 + (\mathcal{T}^{\mho}_{\mathcal{A}_j}(\mathfrak{u}_i))^2 * (\mathcal{T}^{\mho}_{\mathcal{B}_j}(\mathfrak{u}_i))^2 + (\mathcal{J}^{\ell}_{\mathcal{A}_j}(\mathfrak{u}_i))^2 * (\mathcal{J}^{\ell}_{\mathcal{B}_j}(\mathfrak{u}_i))^2 + (\mathcal{J}^{\mho}_{\mathcal{A}_j}(\mathfrak{u}_i))^2 * (\mathcal{J}^{\mho}_{\mathcal{B}_j}(\mathfrak{u}_i))^2)$$

$$= \sum_{j=1}^{m} ((\mathcal{T}^{\ell}_{\mathcal{A}_j}(\mathfrak{u}_1))^2 * (\mathcal{T}^{\ell}_{\mathcal{B}_j}(\mathfrak{u}_1))^2 + (\mathcal{T}^{\mho}_{\mathcal{A}_j}(\mathfrak{u}_1))^2 * (\mathcal{T}^{\mho}_{\mathcal{B}_j}(\mathfrak{u}_1))^2 + (\mathcal{J}^{\ell}_{\mathcal{A}_j}(\mathfrak{u}_1))^2 * (\mathcal{J}^{\ell}_{\mathcal{B}_j}(\mathfrak{u}_1))^2 + (\mathcal{J}^{\mho}_{\mathcal{A}_j}(\mathfrak{u}_1))^2 * (\mathcal{J}^{\mho}_{\mathcal{B}_j}(\mathfrak{u}_1))^2)$$

$$+ \sum_{j=1}^{m} ((\mathcal{T}^{\ell}_{\mathcal{A}_j}(\mathfrak{u}_2))^2 * (\mathcal{T}^{\ell}_{\mathcal{B}_j}(\mathfrak{u}_2))^2 + (\mathcal{T}^{\mho}_{\mathcal{A}_j}(\mathfrak{u}_2))^2 * (\mathcal{T}^{\mho}_{\mathcal{B}_j}(\mathfrak{u}_2))^2 + (\mathcal{J}^{\ell}_{\mathcal{A}_j}(\mathfrak{u}_2))^2 * (\mathcal{J}^{\ell}_{\mathcal{B}_j}(\mathfrak{u}_2))^2 + (\mathcal{J}^{\mho}_{\mathcal{A}_j}(\mathfrak{u}_2))^2 * (\mathcal{J}^{\mho}_{\mathcal{B}_j}(\mathfrak{u}_2))^2)$$

$$+$$

$$\vdots$$

$$+$$

$$\sum_{j=1}^{m} ((\mathcal{T}^{\ell}_{\mathcal{A}_j}(\mathfrak{u}_n))^2 * (\mathcal{T}^{\ell}_{\mathcal{B}_j}(\mathfrak{u}_n))^2 + (\mathcal{T}^{\mho}_{\mathcal{A}_j}(\mathfrak{u}_n))^2 * (\mathcal{T}^{\mho}_{\mathcal{B}_j}(\mathfrak{u}_n))^2 + (\mathcal{J}^{\ell}_{\mathcal{A}_j}(\mathfrak{u}_n))^2 * (\mathcal{J}^{\ell}_{\mathcal{B}_j}(\mathfrak{u}_n))^2 + (\mathcal{J}^{\mho}_{\mathcal{A}_j}(\mathfrak{u}_n))^2 * (\mathcal{J}^{\mho}_{\mathcal{B}_j}(\mathfrak{u}_n))^2)$$

$\mathcal{C}_{IVPFSS}((\mathcal{F}, \mathcal{A}), (\mathcal{G}, \mathcal{B}))$

$$= \left\{ \begin{array}{l} ((\mathcal{T}^{\ell}_{\mathcal{A}_1}(\mathfrak{u}_1))^2 * (\mathcal{T}^{\ell}_{\mathcal{B}_1}(\mathfrak{u}_1))^2 + (\mathcal{T}^{\mho}_{\mathcal{A}_1}(\mathfrak{u}_1))^2 * (\mathcal{T}^{\mho}_{\mathcal{B}_1}(\mathfrak{u}_1))^2 + (\mathcal{J}^{\ell}_{\mathcal{A}_1}(\mathfrak{u}_1))^2 * (\mathcal{J}^{\ell}_{\mathcal{B}_1}(\mathfrak{u}_1))^2 + (\mathcal{J}^{\mho}_{\mathcal{A}_1}(\mathfrak{u}_1))^2 * (\mathcal{J}^{\mho}_{\mathcal{B}_1}(\mathfrak{u}_1))^2) + \\ ((\mathcal{T}^{\ell}_{\mathcal{A}_2}(\mathfrak{u}_1))^2 * (\mathcal{T}^{\ell}_{\mathcal{B}_2}(\mathfrak{u}_1))^2 + (\mathcal{T}^{\mho}_{\mathcal{A}_2}(\mathfrak{u}_1))^2 * (\mathcal{T}^{\mho}_{\mathcal{B}_2}(\mathfrak{u}_1))^2 + (\mathcal{J}^{\ell}_{\mathcal{A}_2}(\mathfrak{u}_1))^2 * (\mathcal{J}^{\ell}_{\mathcal{B}_2}(\mathfrak{u}_1))^2 + (\mathcal{J}^{\mho}_{\mathcal{A}_2}(\mathfrak{u}_1))^2 * (\mathcal{J}^{\mho}_{\mathcal{B}_2}(\mathfrak{u}_1))^2) + \\ \vdots \\ + \\ ((\mathcal{T}^{\ell}_{\mathcal{A}_m}(\mathfrak{u}_1))^2 * (\mathcal{T}^{\ell}_{\mathcal{B}_m}(\mathfrak{u}_1))^2 + (\mathcal{T}^{\mho}_{\mathcal{A}_m}(\mathfrak{u}_1))^2 * (\mathcal{T}^{\mho}_{\mathcal{B}_m}(\mathfrak{u}_1))^2 + (\mathcal{J}^{\ell}_{\mathcal{A}_m}(\mathfrak{u}_1))^2 * (\mathcal{J}^{\ell}_{\mathcal{B}_m}(\mathfrak{u}_1))^2 + (\mathcal{J}^{\mho}_{\mathcal{A}_m}(\mathfrak{u}_1))^2 * (\mathcal{J}^{\mho}_{\mathcal{B}_m}(\mathfrak{u}_1))^2) \end{array} \right\}$$

$$+$$

$$\left\{ \begin{array}{l} ((\mathcal{T}^{\ell}_{\mathcal{A}_1}(\mathfrak{u}_2))^2 * (\mathcal{T}^{\ell}_{\mathcal{B}_1}(\mathfrak{u}_2))^2 + (\mathcal{T}^{\mho}_{\mathcal{A}_1}(\mathfrak{u}_2))^2 * (\mathcal{T}^{\mho}_{\mathcal{B}_1}(\mathfrak{u}_2))^2 + (\mathcal{J}^{\ell}_{\mathcal{A}_1}(\mathfrak{u}_2))^2 * (\mathcal{J}^{\ell}_{\mathcal{B}_1}(\mathfrak{u}_2))^2 + (\mathcal{J}^{\mho}_{\mathcal{A}_1}(\mathfrak{u}_2))^2 * (\mathcal{J}^{\mho}_{\mathcal{B}_1}(\mathfrak{u}_2))^2) + \\ ((\mathcal{T}^{\ell}_{\mathcal{A}_2}(\mathfrak{u}_2))^2 * (\mathcal{T}^{\ell}_{\mathcal{B}_2}(\mathfrak{u}_2))^2 + (\mathcal{T}^{\mho}_{\mathcal{A}_2}(\mathfrak{u}_2))^2 * (\mathcal{T}^{\mho}_{\mathcal{B}_2}(\mathfrak{u}_2))^2 + (\mathcal{J}^{\ell}_{\mathcal{A}_2}(\mathfrak{u}_2))^2 * (\mathcal{J}^{\ell}_{\mathcal{B}_2}(\mathfrak{u}_2))^2 + (\mathcal{J}^{\mho}_{\mathcal{A}_2}(\mathfrak{u}_2))^2 * (\mathcal{J}^{\mho}_{\mathcal{B}_2}(\mathfrak{u}_2))^2) + \\ \vdots \\ + \\ ((\mathcal{T}^{\ell}_{\mathcal{A}_m}(\mathfrak{u}_2))^2 * (\mathcal{T}^{\ell}_{\mathcal{B}_m}(\mathfrak{u}_2))^2 + (\mathcal{T}^{\mho}_{\mathcal{A}_m}(\mathfrak{u}_2))^2 * (\mathcal{T}^{\mho}_{\mathcal{B}_m}(\mathfrak{u}_2))^2 + (\mathcal{J}^{\ell}_{\mathcal{A}_m}(\mathfrak{u}_2))^2 * (\mathcal{J}^{\ell}_{\mathcal{B}_m}(\mathfrak{u}_2))^2 + (\mathcal{J}^{\mho}_{\mathcal{A}_m}(\mathfrak{u}_2))^2 * (\mathcal{J}^{\mho}_{\mathcal{B}_m}(\mathfrak{u}_2))^2) \end{array} \right\}$$

$$+$$

$$\vdots$$

$$+$$

$$\left\{\begin{array}{l} ((\mathcal{T}^\ell_{\mathcal{A}_1}(\mathfrak{u}_n))^2 * (\mathcal{T}^\ell_{\mathcal{B}_1}(\mathfrak{u}_n))^2 + (\mathcal{T}^\mho_{\mathcal{A}_1}(\mathfrak{u}_n))^2 * (\mathcal{T}^\mho_{\mathcal{B}_1}(\mathfrak{u}_n))^2 + (\mathcal{J}^\ell_{\mathcal{A}_1}(\mathfrak{u}_n))^2 * (\mathcal{J}^\ell_{\mathcal{B}_1}(\mathfrak{u}_n))^2 + (\mathcal{J}^\mho_{\mathcal{A}_1}(\mathfrak{u}_n))^2 * (\mathcal{J}^\mho_{\mathcal{B}_1}(\mathfrak{u}_n))^2) + \\ ((\mathcal{T}^\ell_{\mathcal{A}_2}(\mathfrak{u}_n))^2 * (\mathcal{T}^\ell_{\mathcal{B}_2}(\mathfrak{u}_n))^2 + (\mathcal{T}^\mho_{\mathcal{A}_2}(\mathfrak{u}_n))^2 * (\mathcal{T}^\mho_{\mathcal{B}_2}(\mathfrak{u}_n))^2 + (\mathcal{J}^\ell_{\mathcal{A}_2}(\mathfrak{u}_n))^2 * (\mathcal{J}^\ell_{\mathcal{B}_2}(\mathfrak{u}_n))^2 + (\mathcal{J}^\mho_{\mathcal{A}_2}(\mathfrak{u}_n))^2 * (\mathcal{J}^\mho_{\mathcal{B}_2}(\mathfrak{u}_n))^2) + \\ \vdots \\ + \\ ((\mathcal{T}^\ell_{\mathcal{A}_m}(\mathfrak{u}_n))^2 * (\mathcal{T}^\ell_{\mathcal{B}_m}(\mathfrak{u}_n))^2 + (\mathcal{T}^\mho_{\mathcal{A}_m}(\mathfrak{u}_n))^2 * (\mathcal{T}^\mho_{\mathcal{B}_m}(\mathfrak{u}_n))^2 + (\mathcal{J}^\ell_{\mathcal{A}_m}(\mathfrak{u}_n))^2 * (\mathcal{J}^\ell_{\mathcal{B}_m}(\mathfrak{u}_n))^2 + (\mathcal{J}^\mho_{\mathcal{A}_m}(\mathfrak{u}_n))^2 * (\mathcal{J}^\mho_{\mathcal{B}_m}(\mathfrak{u}_n))^2) \end{array}\right\}$$

$$= \sum_{j=1}^{m} \left( \begin{array}{c} ((\mathcal{T}^\ell_{\mathcal{A}_j}(\mathfrak{u}_1))^2 * (\mathcal{T}^\ell_{\mathcal{B}_j}(\mathfrak{u}_1))^2 + (\mathcal{T}^\mho_{\mathcal{A}_j}(\mathfrak{u}_1))^2 * (\mathcal{T}^\mho_{\mathcal{B}_j}(\mathfrak{u}_1))^2) + ((\mathcal{T}^\ell_{\mathcal{A}_j}(\mathfrak{u}_2))^2 * (\mathcal{T}^\ell_{\mathcal{B}_j}(\mathfrak{u}_2))^2 + (\mathcal{T}^\mho_{\mathcal{A}_j}(\mathfrak{u}_2))^2 * (\mathcal{T}^\mho_{\mathcal{B}_j}(\mathfrak{u}_2))^2) \\ + \cdots + ((\mathcal{T}^\ell_{\mathcal{A}_j}(\mathfrak{u}_n))^2 * (\mathcal{T}^\ell_{\mathcal{B}_j}(\mathfrak{u}_n))^2 + (\mathcal{T}^\mho_{\mathcal{A}_j}(\mathfrak{u}_n))^2 * (\mathcal{T}^\mho_{\mathcal{B}_j}(\mathfrak{u}_n))^2) \end{array} \right)$$
$$+ \sum_{j=1}^{m} \left( \begin{array}{c} ((\mathcal{J}^\ell_{\mathcal{A}_j}(\mathfrak{u}_1))^2 * (\mathcal{J}^\ell_{\mathcal{B}_j}(\mathfrak{u}_1))^2 + (\mathcal{J}^\mho_{\mathcal{A}_j}(\mathfrak{u}_1))^2 * (\mathcal{J}^\mho_{\mathcal{B}_j}(\mathfrak{u}_1))^2) + ((\mathcal{J}^\ell_{\mathcal{A}_j}(\mathfrak{u}_2))^2 * (\mathcal{J}^\ell_{\mathcal{B}_j}(\mathfrak{u}_2))^2 + (\mathcal{J}^\mho_{\mathcal{A}_j}(\mathfrak{u}_2))^2 * (\mathcal{J}^\mho_{\mathcal{B}_j}(\mathfrak{u}_2))^2) \\ + \cdots + ((\mathcal{J}^\ell_{\mathcal{A}_j}(\mathfrak{u}_n))^2 * (\mathcal{J}^\ell_{\mathcal{B}_j}(\mathfrak{u}_n))^2 + (\mathcal{J}^\mho_{\mathcal{A}_j}(\mathfrak{u}_n))^2 * (\mathcal{J}^\mho_{\mathcal{B}_j}(\mathfrak{u}_n))^2) \end{array} \right)$$

Using Cauchy-Schwarz inequality

$$\mathcal{C}_{IVPFSS}((\mathcal{F}, \mathcal{A}), (\mathcal{G}, \mathcal{B}))^2$$
$$\leq \sum_{j=1}^{m} \left\{ \begin{array}{l} ((\mathcal{T}^\ell_{\mathcal{A}_j}(\mathfrak{u}_1))^4 + (\mathcal{T}^\mho_{\mathcal{A}_j}(\mathfrak{u}_1))^4) + ((\mathcal{T}^\ell_{\mathcal{A}_j}(\mathfrak{u}_2))^4 + (\mathcal{T}^\mho_{\mathcal{A}_j}(\mathfrak{u}_2))^4) + \ldots + ((\mathcal{T}^\ell_{\mathcal{A}_j}(\mathfrak{u}_n))^4 + (\mathcal{T}^\mho_{\mathcal{A}_j}(\mathfrak{u}_n))^4) + \\ ((\mathcal{J}^\ell_{\mathcal{A}_j}(\mathfrak{u}_1))^4 + (\mathcal{J}^\mho_{\mathcal{A}_j}(\mathfrak{u}_1))^4) + ((\mathcal{J}^\ell_{\mathcal{A}_j}(\mathfrak{u}_2))^4 + (\mathcal{J}^\mho_{\mathcal{A}_j}(\mathfrak{u}_2))^4) + \ldots + ((\mathcal{J}^\ell_{\mathcal{A}_j}(\mathfrak{u}_n))^4 + (\mathcal{J}^\mho_{\mathcal{A}_j}(\mathfrak{u}_n))^4) \end{array} \right\}$$
$$\times \sum_{j=1}^{m} \left\{ \begin{array}{l} ((\mathcal{T}^\ell_{\mathcal{B}_j}(\mathfrak{u}_1))^4 + (\mathcal{T}^\mho_{\mathcal{B}_j}(\mathfrak{u}_1))^4) + ((\mathcal{T}^\ell_{\mathcal{B}_j}(\mathfrak{u}_2))^4 + (\mathcal{T}^\mho_{\mathcal{B}_j}(\mathfrak{u}_2))^4) + \ldots + ((\mathcal{T}^\ell_{\mathcal{B}_j}(\mathfrak{u}_n))^4 + (\mathcal{T}^\mho_{\mathcal{B}_j}(\mathfrak{u}_n))^4) + \\ ((\mathcal{J}^\ell_{\mathcal{B}_j}(\mathfrak{u}_1))^4 + (\mathcal{J}^\mho_{\mathcal{B}_j}(\mathfrak{u}_1))^4) + ((\mathcal{J}^\ell_{\mathcal{B}_j}(\mathfrak{u}_2))^4 + (\mathcal{J}^\mho_{\mathcal{B}_j}(\mathfrak{u}_2))^4) + \ldots + ((\mathcal{J}^\ell_{\mathcal{B}_j}(\mathfrak{u}_n))^4 + (\mathcal{J}^\mho_{\mathcal{B}_j}(\mathfrak{u}_n))^4) \end{array} \right\}$$

$$\mathcal{C}_{IVPFSS}((\mathcal{F}, \mathcal{A}), (\mathcal{G}, \mathcal{B}))^2$$
$$\leq \sum_{j=1}^{m} \sum_{i=1}^{n} \{((\mathcal{T}^\ell_{\mathcal{A}_j}(\mathfrak{u}_i))^4 + (\mathcal{T}^\mho_{\mathcal{A}_j}(\mathfrak{u}_i))^4) + ((\mathcal{J}^\ell_{\mathcal{A}_j}(\mathfrak{u}_i))^4 + (\mathcal{J}^\mho_{\mathcal{A}_j}(\mathfrak{u}_i))^4)\}$$
$$\times \sum_{j=1}^{m} \sum_{i=1}^{n} \{((\mathcal{T}^\ell_{\mathcal{B}_j}(\mathfrak{u}_i))^4 + (\mathcal{T}^\mho_{\mathcal{B}_j}(\mathfrak{u}_i))^4) + ((\mathcal{J}^\ell_{\mathcal{B}_j}(\mathfrak{u}_i))^4 + (\mathcal{J}^\mho_{\mathcal{B}_j}(\mathfrak{u}_i))^4)\}$$

$$\mathcal{C}_{IVPFSS}((\mathcal{F}, \mathcal{A}), (\mathcal{G}, \mathcal{B}))^2 \leq \mathcal{E}_{IVPFSS}(\mathcal{F}, \mathcal{A}) \times \mathcal{E}_{IVPFSS}(\mathcal{G}, \mathcal{B}).$$

Using Definition 3.3, we get

$$\mathbb{C}_{IVPFSS}((\mathcal{F}, \mathcal{A}), (\mathcal{G}, \mathcal{B})) \leq 1.$$

So, it is verified that $0 \leq \mathbb{C}_{IVPFSS}((\mathcal{F}, \mathcal{A}), (\mathcal{G}, \mathcal{B})) \leq 1$.
**Proof 2.** The proof is simple and easy to follow.

**Proof 3.** It is well-known that

$$\mathbb{C}_{IVPFSS}((\mathcal{F}, \mathcal{A}), (\mathcal{G}, \mathcal{B}))$$

$$= \frac{\sum_{j=1}^{m} \sum_{i=1}^{n} ((\mathcal{T}_{\mathcal{A}_j}^{\ell}(\mathfrak{u}_i))^2 * (\mathcal{T}_{\mathcal{B}_j}^{\ell}(\mathfrak{u}_i))^2 + (\mathcal{T}_{\mathcal{A}_j}^{\mathcal{V}}(\mathfrak{u}_i))^2 * (\mathcal{T}_{\mathcal{B}_j}^{\mathcal{V}}(\mathfrak{u}_i))^2 + (\mathcal{J}_{\mathcal{A}_j}^{\ell}(\mathfrak{u}_i))^2 * (\mathcal{J}_{\mathcal{B}_j}^{\ell}(\mathfrak{u}_i))^2 + (\mathcal{J}_{\mathcal{A}_j}^{\mathcal{V}}(\mathfrak{u}_i))^2 * (\mathcal{J}_{\mathcal{B}_j}^{\mathcal{V}}(\mathfrak{u}_i))^2.}{\sqrt{\sum_{j=1}^{m} \sum_{i=1}^{n} ((\mathcal{T}_{\mathcal{A}_j}^{\ell}(\mathfrak{u}_i))^4 + (\mathcal{T}_{\mathcal{A}_j}^{\mathcal{V}}(\mathfrak{u}_i))^4 + (\mathcal{J}_{\mathcal{A}_j}^{\ell}(\mathfrak{u}_i))^4 + (\mathcal{J}_{\mathcal{A}_j}^{\mathcal{V}}(\mathfrak{u}_i))^4)} \sqrt{\sum_{j=1}^{m} \sum_{i=1}^{n} ((\mathcal{T}_{\mathcal{B}_j}^{\ell}(\mathfrak{u}_i))^4 + (\mathcal{T}_{\mathcal{B}_j}^{\mathcal{V}}(\mathfrak{u}_i))^4 + (\mathcal{J}_{\mathcal{B}_j}^{\ell}(\mathfrak{u}_i))^4 + (\mathcal{J}_{\mathcal{B}_j}^{\mathcal{V}}(\mathfrak{u}_i))^4)}}$$

As
$$\mathcal{T}_{\mathcal{A}_j}^{\ell}(\mathfrak{u}_i) = \mathcal{T}_{\mathcal{B}_j}^{\ell}(\mathfrak{u}_i), \mathcal{T}_{\mathcal{A}_j}^{\mathcal{V}}(\mathfrak{u}_i) = \mathcal{T}_{\mathcal{B}_j}^{\mathcal{V}}(\mathfrak{u}_i), \mathcal{J}_{\mathcal{A}_j}^{\ell}(\mathfrak{u}_i) = \mathcal{J}_{\mathcal{B}_j}^{\ell}(\mathfrak{u}_i), \text{ and } \mathcal{J}_{\mathcal{A}_j}^{\mathcal{V}}(\mathfrak{u}_i) = \mathcal{J}_{\mathcal{B}_j}^{\mathcal{V}}(\mathfrak{u}_i). \text{ So,}$$

$$\mathbb{C}_{IVPFSS}((\mathcal{F}, \mathcal{A}), (\mathcal{G}, \mathcal{B}))$$

$$= \frac{\sum_{j=1}^{m} \sum_{i=1}^{n} ((\mathcal{T}_{\mathcal{A}_j}^{\ell}(\mathfrak{u}_i))^4 + (\mathcal{T}_{\mathcal{A}_j}^{\mathcal{V}}(\mathfrak{u}_i))^4 + (\mathcal{J}_{\mathcal{A}_j}^{\ell}(\mathfrak{u}_i))^4 + (\mathcal{J}_{\mathcal{A}_j}^{\mathcal{V}}(\mathfrak{u}_i))^4.}{\sqrt{\sum_{j=1}^{m} \sum_{i=1}^{n} ((\mathcal{T}_{\mathcal{A}_j}^{\ell}(\mathfrak{u}_i))^4 + (\mathcal{T}_{\mathcal{A}_j}^{\mathcal{V}}(\mathfrak{u}_i))^4 + (\mathcal{J}_{\mathcal{A}_j}^{\ell}(\mathfrak{u}_i))^4 + (\mathcal{J}_{\mathcal{A}_j}^{\mathcal{V}}(\mathfrak{u}_i))^4)} \sqrt{\sum_{j=1}^{m} \sum_{i=1}^{n} ((\mathcal{T}_{\mathcal{A}_j}^{\ell}(\mathfrak{u}_i))^4 + (\mathcal{T}_{\mathcal{A}_j}^{\mathcal{V}}(\mathfrak{u}_i))^4 + (\mathcal{J}_{\mathcal{A}_j}^{\ell}(\mathfrak{u}_i))^4 + (\mathcal{J}_{\mathcal{A}_j}^{\mathcal{V}}(\mathfrak{u}_i))^4)}}$$

$$\mathbb{C}_{IVPFSS}((\mathcal{F}, \mathcal{A}), (\mathcal{G}, \mathcal{B})) = 1.$$

**Definition 3.4.** Let $(\mathcal{F}, \mathcal{A}) = \{(\mathfrak{u}_i, ([\mathcal{T}_{\mathcal{A}_j}^{\ell}(\mathfrak{u}_i), \mathcal{T}_{\mathcal{A}_j}^{\mathcal{V}}(\mathfrak{u}_i)], [\mathcal{J}_{\mathcal{A}_j}^{\ell}(\mathfrak{u}_i), \mathcal{J}_{\mathcal{A}_j}^{\mathcal{V}}(\mathfrak{u}_i)]))|\mathfrak{u}_i \in U\}$ and
$(\mathcal{G}, \mathcal{B}) = \{(\mathfrak{u}_i, ([\mathcal{T}_{\mathcal{B}_j}^{\ell}(\mathfrak{u}_i), \mathcal{T}_{\mathcal{B}_j}^{\mathcal{V}}(\mathfrak{u}_i)], [\mathcal{J}_{\mathcal{B}_j}^{\ell}(\mathfrak{u}_i), \mathcal{J}_{\mathcal{B}_j}^{\mathcal{V}}(\mathfrak{u}_i)]))|\mathfrak{u}_i \in U\}$ be two IVPFSS. Then, CC is
also defined as:

$$\mathbb{C}_{IVPFSS}^{1}((\mathcal{F}, \mathcal{A}), (\mathcal{G}, \mathcal{B})) = \frac{\mathcal{C}_{IVPFSS}((\mathcal{F}, \mathcal{A}), (\mathcal{G}, \mathcal{B}))}{max\{\mathcal{E}_{IVPFSS}(\mathcal{F}, \mathcal{A}), \mathcal{E}_{IVPFSS}(\mathcal{G}, \mathcal{B})\}}$$

$$\mathbb{C}_{IVPFSS}^{1}((\mathcal{F}, \mathcal{A}), (\mathcal{G}, \mathcal{B}))$$

$$= \frac{\sum_{j=1}^{m} \sum_{i=1}^{n} ((\mathcal{T}_{\mathcal{A}_j}^{\ell}(\mathfrak{u}_i))^2 * (\mathcal{T}_{\mathcal{B}_j}^{\ell}(\mathfrak{u}_i))^2 + (\mathcal{T}_{\mathcal{A}_j}^{\mathcal{V}}(\mathfrak{u}_i))^2 * (\mathcal{T}_{\mathcal{B}_j}^{\mathcal{V}}(\mathfrak{u}_i))^2 + (\mathcal{J}_{\mathcal{A}_j}^{\ell}(\mathfrak{u}_i))^2 * (\mathcal{J}_{\mathcal{B}_j}^{\ell}(\mathfrak{u}_i))^2 + (\mathcal{J}_{\mathcal{A}_j}^{\mathcal{V}}(\mathfrak{u}_i))^2 * (\mathcal{J}_{\mathcal{B}_j}^{\mathcal{V}}(\mathfrak{u}_i))^2)}{max\{\sum_{j=1}^{m} \sum_{i=1}^{n} ((\mathcal{T}_{\mathcal{A}_j}^{\ell}(\mathfrak{u}_i))^4 + (\mathcal{T}_{\mathcal{A}_j}^{\mathcal{V}}(\mathfrak{u}_i))^4 + (\mathcal{J}_{\mathcal{A}_j}^{\ell}(\mathfrak{u}_i))^4 + (\mathcal{J}_{\mathcal{A}_j}^{\mathcal{V}}(\mathfrak{u}_i))^4), \sum_{j=1}^{m} \sum_{i=1}^{n} ((\mathcal{T}_{\mathcal{B}_j}^{\ell}(\mathfrak{u}_i))^4 + (\mathcal{T}_{\mathcal{B}_j}^{\mathcal{V}}(\mathfrak{u}_i))^4 + (\mathcal{J}_{\mathcal{B}_j}^{\ell}(\mathfrak{u}_i))^4 + (\mathcal{J}_{\mathcal{B}_j}^{\mathcal{V}}(\mathfrak{u}_i))^4)\}} \quad (3.5)$$

**Theorem 2.** Let $(\mathcal{F}, \mathcal{A}) = \{(\mathfrak{u}_i, ([\mathcal{T}_{\mathcal{A}_j}^{\ell}(\mathfrak{u}_i), \mathcal{T}_{\mathcal{A}_j}^{\mathcal{V}}(\mathfrak{u}_i)], [\mathcal{J}_{\mathcal{A}_j}^{\ell}(\mathfrak{u}_i), \mathcal{J}_{\mathcal{A}_j}^{\mathcal{V}}(\mathfrak{u}_i)]))|\mathfrak{u}_i \in U\}$ and
$(\mathcal{G}, \mathcal{B}) = \{(\mathfrak{u}_i, ([\mathcal{T}_{\mathcal{B}_j}^{\ell}(\mathfrak{u}_i), \mathcal{T}_{\mathcal{B}_j}^{\mathcal{V}}(\mathfrak{u}_i)], [\mathcal{J}_{\mathcal{B}_j}^{\ell}(\mathfrak{u}_i), \mathcal{J}_{\mathcal{B}_j}^{\mathcal{V}}(\mathfrak{u}_i)]))|\mathfrak{u}_i \in U\}$ be two IVPFSS, then the following properties are hold:

1. $0 \leq \mathbb{C}_{IVPFSS}^{1}((\mathcal{F}, \mathcal{A}), (\mathcal{G}, \mathcal{B})) \leq 1$.

2. $\mathbb{C}_{IVPFSS}^{1}((\mathcal{F}, \mathcal{A}), (\mathcal{G}, \mathcal{B})) = \mathbb{C}_{IVPFSS}^{1}((\mathcal{G}, \mathcal{B}), (\mathcal{F}, \mathcal{A}))$

3. If $\mathcal{T}_{\mathcal{A}_j}^{\ell}(\mathfrak{u}_i) = \mathcal{T}_{\mathcal{B}_j}^{\ell}(\mathfrak{u}_i), \mathcal{T}_{\mathcal{A}_j}^{\mathcal{V}}(\mathfrak{u}_i) = \mathcal{T}_{\mathcal{B}_j}^{\mathcal{V}}(\mathfrak{u}_i), \mathcal{J}_{\mathcal{A}_j}^{\ell}(\mathfrak{u}_i) = \mathcal{J}_{\mathcal{B}_j}^{\ell}(\mathfrak{u}_i), \text{ and } \mathcal{J}_{\mathcal{A}_j}^{\mathcal{V}}(\mathfrak{u}_i) = \mathcal{J}_{\mathcal{B}_j}^{\mathcal{V}}(\mathfrak{u}_i) \forall i, j.$
Then $\mathbb{C}_{IVPFSS}^{1}((\mathcal{F}, \mathcal{A}), (\mathcal{G}, \mathcal{B})) = 1$.

**Proof.** The proof for case 2 is simple and can be easily demonstrated. The proof for case 3 follows a similar pattern as shown in Theorem 1 for case 3. Also, $\mathbb{C}_{IVPFSS}^{1}((\mathcal{F}, \mathcal{A}), (\mathcal{G}, \mathcal{B})) \geq 0$ is trivial in case 1. Here, we only need to prove $\mathbb{C}_{IVPFSS}^{1}((\mathcal{F}, \mathcal{A}), (\mathcal{G}, \mathcal{B})) \leq 1$. Since, $\mathcal{C}_{IVPFSS}((\mathcal{F}, \mathcal{A}), (\mathcal{G}, \mathcal{B}))^2 \leq \mathcal{E}_{IVPFSS}(\mathcal{F}, \mathcal{A}) \times \mathcal{E}_{IVPFSS}(\mathcal{G}, \mathcal{B})$. So, $\mathcal{C}_{IVPFSS}((\mathcal{F}, \mathcal{A}), (\mathcal{G}, \mathcal{B})) \leq max\{\mathcal{E}_{IVPFSS}(\mathcal{F}, \mathcal{A}), \mathcal{E}_{IVPFSS}(\mathcal{G}, \mathcal{B})\}$. Hence, $\mathbb{C}_{IVPFSS}^{1}((\mathcal{F}, \mathcal{A}), (\mathcal{G}, \mathcal{B})) \leq 1$.

## 4. Weighted correlation coefficient for interval valued Pythagorean fuzzy soft set

In today's world, it is crucial to consider the significance of IVPFSS in practical decision-making. The results may vary depending on policymakers' weights to different alternatives during

the planning process. Hence, determining the weights of decision-makers and alternatives is crucial before drawing any conclusions. To address this, we introduce the WCC for IVPFSS. Let $\Omega = \{\Omega_1, \Omega_2, \Omega_3,\ldots,\Omega_n\}^T$ and $\gamma = \{\gamma_1, \gamma_2, \gamma_3,\ldots,\gamma_m\}^T$ represent the weights for experts and parameters, where $\Omega_i > 0, \sum_{i=1}^{m} \Omega_i = 1$ and $\gamma_j > 0, \sum_{j=1}^{m} \gamma_j = 1$.

**Definition 4.1.** Let $(\mathcal{F}, \mathcal{A}) = \{(\mathfrak{u}_i, (\mathcal{T}_{\mathcal{A}_j}(\mathfrak{u}_i), \mathcal{J}_{\mathcal{A}_j}(\mathfrak{u}_i)))|\mathfrak{u}_i \in U\}$ and $(\mathcal{G}, \mathcal{B}) = \{(\mathfrak{u}_i, (\mathcal{T}_{\mathcal{B}_j}(\mathfrak{u}_i), \mathcal{J}_{\mathcal{B}_j}(\mathfrak{u}_i)))|\mathfrak{u}_i \in U\}$ be two IVPFSS over a set of attributes

$\varsigma = \{\varsigma_1, \varsigma_2, \varsigma_3,\ldots,\varsigma_m)$, where $\mathcal{T}_{\mathcal{A}_j}(\mathfrak{u}_i) = [\mathcal{T}_{\mathcal{A}_j}^{\ell}(\mathfrak{u}_i), \mathcal{T}_{\mathcal{A}_j}^{\mho}(\mathfrak{u}_i)], \mathcal{J}_{\mathcal{A}_j}(\mathfrak{u}_i) = [\mathcal{J}_{\mathcal{A}_j}^{\ell}(\mathfrak{u}_i), \mathcal{J}_{\mathcal{A}_j}^{\mho}(\mathfrak{u}_i)], \mathcal{T}_{\mathcal{B}_j}(\mathfrak{u}_i) = [\mathcal{T}_{\mathcal{B}_j}^{\ell}(\mathfrak{u}_i), \mathcal{T}_{\mathcal{B}_j}^{\mho}(\mathfrak{u}_i)], \mathcal{J}_{\mathcal{B}_j}(\mathfrak{u}_i) = [\mathcal{J}_{\mathcal{B}_j}^{\ell}(\mathfrak{u}_i), \mathcal{J}_{\mathcal{B}_j}^{\mho}(\mathfrak{u}_i)]$. Then the weighted informational energies of $(\mathcal{F}, \mathcal{A})$ and $(\mathcal{G}, \mathcal{B})$ are defined as:

$$\mathcal{E}_{WIVPFSS}(\mathcal{F}, \mathcal{A}) = \sum_{j=1}^{m} \gamma_j \left(\sum_{i=1}^{n} \Omega_i((\mathcal{T}_{\mathcal{A}_j}^{\ell}(\mathfrak{u}_i))^4 + (\mathcal{T}_{\mathcal{A}_j}^{\mho}(\mathfrak{u}_i))^4 + (\mathcal{J}_{\mathcal{A}_j}^{\ell}(\mathfrak{u}_i))^4 + (\mathcal{J}_{\mathcal{A}_j}^{\mho}(\mathfrak{u}_i))^4)\right) \quad (4.1)$$

$$\mathcal{E}_{WIVPFSS}(\mathcal{G}, \mathcal{B}) = \sum_{j=1}^{m} \gamma_j \left(\sum_{i=1}^{n} \Omega_i((\mathcal{T}_{\mathcal{B}_j}^{\ell}(\mathfrak{u}_i))^4 + (\mathcal{T}_{\mathcal{B}_j}^{\mho}(\mathfrak{u}_i))^4 + (\mathcal{J}_{\mathcal{B}_j}^{\ell}(\mathfrak{u}_i))^4 + (\mathcal{J}_{\mathcal{B}_j}^{\mho}(\mathfrak{u}_i))^4)\right). \quad (4.2)$$

**Definition 4.2.** Let $(\mathcal{F}, \mathcal{A}) = \{(\mathfrak{u}_i, ([\mathcal{T}_{\mathcal{A}_j}^{\ell}(\mathfrak{u}_i), \mathcal{T}_{\mathcal{A}_j}^{\mho}(\mathfrak{u}_i)], [\mathcal{J}_{\mathcal{A}_j}^{\ell}(\mathfrak{u}_i), \mathcal{J}_{\mathcal{A}_j}^{\mho}(\mathfrak{u}_i)]))|\mathfrak{u}_i \in U\}$ and $(\mathcal{G}, \mathcal{B}) = \{(\mathfrak{u}_i, ([\mathcal{T}_{\mathcal{B}_j}^{\ell}(\mathfrak{u}_i), \mathcal{T}_{\mathcal{B}_j}^{\mho}(\mathfrak{u}_i)], [\mathcal{J}_{\mathcal{B}_j}^{\ell}(\mathfrak{u}_i), \mathcal{J}_{\mathcal{B}_j}^{\mho}(\mathfrak{u}_i)]))|\mathfrak{u}_i \in U\}$ be two IVPFSS. Then, the WCC between $(\mathcal{F}, \mathcal{A})$ and $(\mathcal{G}, \mathfrak{B})$ is defined as:

$$\mathcal{C}_{WIVPFSS}((\mathcal{F}, \mathcal{A}), (\mathcal{G}, \mathcal{B})) = \sum_{j=1}^{m} \gamma_j \left( \sum_{i=1}^{n} \Omega_i\left(\begin{matrix} (\mathcal{T}_{\mathcal{A}_j}^{\ell}(\mathfrak{u}_i))^2 * (\mathcal{T}_{\mathcal{B}_j}^{\ell}(\mathfrak{u}_i))^2 + (\mathcal{T}_{\mathcal{A}_j}^{\mho}(\mathfrak{u}_i))^2 * (\mathcal{T}_{\mathcal{B}_j}^{\mho}(\mathfrak{u}_i))^2 + \\ (\mathcal{J}_{\mathcal{A}_j}^{\ell}(\mathfrak{u}_i))^2 * (\mathcal{J}_{\mathcal{B}_j}^{\ell}(\mathfrak{u}_i))^2 + (\mathcal{J}_{\mathcal{A}_j}^{\mho}(\mathfrak{u}_i))^2 * (\mathcal{J}_{\mathcal{B}_j}^{\mho}(\mathfrak{u}_i))^2 \end{matrix}\right) \right). \quad (4.3)$$

**Proposition 1:** Let $(\mathcal{F}, \mathcal{A}) = \{(\mathfrak{u}_i, ([\mathcal{T}_{\mathcal{A}_j}^{\ell}(\mathfrak{u}_i), \mathcal{T}_{\mathcal{A}_j}^{\mho}(\mathfrak{u}_i)], [\mathcal{J}_{\mathcal{A}_j}^{\ell}(\mathfrak{u}_i), \mathcal{J}_{\mathcal{A}_j}^{\mho}(\mathfrak{u}_i)]))|\mathfrak{u}_i \in U\}$ and $(\mathcal{G}, \mathcal{B}) = \{(\mathfrak{u}_i, ([\mathcal{T}_{\mathcal{B}_j}^{\ell}(\mathfrak{u}_i), \mathcal{T}_{\mathcal{B}_j}^{\mho}(\mathfrak{u}_i)], [\mathcal{J}_{\mathcal{B}_j}^{\ell}(\mathfrak{u}_i), \mathcal{J}_{\mathcal{B}_j}^{\mho}(\mathfrak{u}_i)]))|\mathfrak{u}_i \in U\}$ be two IVPFSS. Then

1. $\mathcal{C}_{WIVPFSS}((\mathcal{F}, \mathcal{A}), (\mathcal{F}, \mathcal{A})) = (\mathcal{F}, \mathcal{A})$

2. $\mathcal{C}_{WIVPFSS}((\mathcal{F}, \mathcal{A}), (\mathcal{G}, \mathcal{B})) = \mathcal{C}_{WIVPFSS}((\mathcal{G}, \mathcal{B}), (\mathcal{F}, \mathcal{A}))$.

**Proof:** The proof is simple and straightforward.

**Definition 4.3.** Let $(\mathcal{F}, \mathcal{A}) = \{(\mathfrak{u}_i, ([\mathcal{T}_{\mathcal{A}_j}^{\ell}(\mathfrak{u}_i), \mathcal{T}_{\mathcal{A}_j}^{\mho}(\mathfrak{u}_i)], [\mathcal{J}_{\mathcal{A}_j}^{\ell}(\mathfrak{u}_i), \mathcal{J}_{\mathcal{A}_j}^{\mho}(\mathfrak{u}_i)]))|\mathfrak{u}_i \in U\}$ and $(\mathcal{G}, \mathcal{B}) = \{(\mathfrak{u}_i, ([\mathcal{T}_{\mathcal{B}_j}^{\ell}(\mathfrak{u}_i), \mathcal{T}_{\mathcal{B}_j}^{\mho}(\mathfrak{u}_i)], [\mathcal{J}_{\mathcal{B}_j}^{\ell}(\mathfrak{u}_i), \mathcal{J}_{\mathcal{B}_j}^{\mho}(\mathfrak{u}_i)]))|\mathfrak{u}_i \in U\}$ be two IVPFSS. Then, the WCC is defined as:

$$\mathbb{C}_{WIVPFSS}((\mathcal{F}, \mathcal{A}), (\mathcal{G}, \mathcal{B})) = \frac{\mathcal{C}_{WIVPFSS}((\mathcal{F}, \mathcal{A}), (\mathcal{G}, \mathcal{B}))}{\sqrt{\mathcal{E}_{WIVPFSS}(\mathcal{F}, \mathcal{A})}\sqrt{\mathcal{E}_{WIVPFSS}(\mathcal{G}, \mathcal{B})}}$$

$$= \frac{\sum_{j=1}^{m} \gamma_j \left( \sum_{i=1}^{n} \Omega_i\left(\begin{matrix} (\mathcal{T}_{\mathcal{A}_j}^{\ell}(\mathfrak{u}_i))^2 * (\mathcal{T}_{\mathcal{B}_j}^{\ell}(\mathfrak{u}_i))^2 + (\mathcal{T}_{\mathcal{A}_j}^{\mho}(\mathfrak{u}_i))^2 * (\mathcal{T}_{\mathcal{B}_j}^{\mho}(\mathfrak{u}_i))^2 + \\ (\mathcal{J}_{\mathcal{A}_j}^{\ell}(\mathfrak{u}_i))^2 * (\mathcal{J}_{\mathcal{B}_j}^{\ell}(\mathfrak{u}_i))^2 + (\mathcal{J}_{\mathcal{A}_j}^{\mho}(\mathfrak{u}_i))^2 * (\mathcal{J}_{\mathcal{B}_j}^{\mho}(\mathfrak{u}_i))^2 \end{matrix}\right) \right).}{\sqrt{\sum_{j=1}^{m} \gamma_j (\sum_{i=1}^{n} \Omega_i((\mathcal{T}_{\mathcal{A}_j}^{\ell}(\mathfrak{u}_i))^4 + (\mathcal{T}_{\mathcal{A}_j}^{\mho}(\mathfrak{u}_i))^4 + (\mathcal{J}_{\mathcal{A}_j}^{\ell}(\mathfrak{u}_i))^4 + (\mathcal{J}_{\mathcal{A}_j}^{\mho}(\mathfrak{u}_i))^4))} \cdot \sqrt{\sum_{j=1}^{m} \gamma_j (\sum_{i=1}^{n} \Omega_i((\mathcal{T}_{\mathcal{B}_j}^{\ell}(\mathfrak{u}_i))^4 + (\mathcal{T}_{\mathcal{B}_j}^{\mho}(\mathfrak{u}_i))^4 + (\mathcal{J}_{\mathcal{B}_j}^{\ell}(\mathfrak{u}_i))^4 + (\mathcal{J}_{\mathcal{B}_j}^{\mho}(\mathfrak{u}_i))^4))}} \quad (4.4)$$

Where $\Omega = \{\Omega_1, \Omega_2, \Omega_3,\ldots,\Omega_m\}^T$ and $\gamma = \{\gamma_1, \gamma_2, \gamma_3,\ldots,\gamma_m\}^T$ be the weight vectors for experts and attributes, respectively, such as $\Omega_i > 0, \sum_{i=1}^{m} \Omega_i = 1$ and $\gamma_j > 0, \sum_{j=1}^{m} \gamma_j = 1$.

**Theorem 3.** Let $(\mathcal{F}, \mathcal{A}) = \{(\mathfrak{u}_i, ([\mathcal{T}_{\mathcal{A}_j}^{\ell}(\mathfrak{u}_i), \mathcal{T}_{\mathcal{A}_j}^{\mho}(\mathfrak{u}_i)], [\mathcal{J}_{\mathcal{A}_j}^{\ell}(\mathfrak{u}_i), \mathcal{J}_{\mathcal{A}_j}^{\mho}(\mathfrak{u}_i)]))|\mathfrak{u}_i \in U\}$ and $(\mathcal{G}, \mathcal{B}) = \{(\mathfrak{u}_i, ([\mathcal{T}_{\mathcal{B}_j}^{\ell}(\mathfrak{u}_i), \mathcal{T}_{\mathcal{B}_j}^{\mho}(\mathfrak{u}_i)], [\mathcal{J}_{\mathcal{B}_j}^{\ell}(\mathfrak{u}_i), \mathcal{J}_{\mathcal{B}_j}^{\mho}(\mathfrak{u}_i)]))|\mathfrak{u}_i \in U\}$ be two IVPFSS. If $\Omega = \{\Omega_1, \Omega_2,$

$\Omega_3, \ldots, \Omega_m\}^T$ and $\gamma = \{\gamma_1, \gamma_2, \gamma_3, \ldots, \gamma_m\}^T$ be the weight vectors for experts and attributes, respectively, such as $\Omega_i > 0$, $\sum_{i=1}^m \Omega_i = 1$ and $\gamma_j > 0$, $\sum_{j=1}^m \gamma_j = 1$. Then, WCC satisfied the following properties:

1. $0 \leq \mathbb{C}_{WIVPFSS}((\mathcal{F}, \mathcal{A}), (\mathcal{G}, \mathcal{B})) \leq 1$

2. $\mathbb{C}_{WIVPFSS}((\mathcal{F}, \mathcal{A}), (\mathcal{G}, \mathcal{B})) = \mathbb{C}_{WIVPFSS}((\mathcal{G}, \mathcal{B}), (\mathcal{F}, \mathcal{A}))$

3. If $(\mathcal{F}, \mathcal{A}) = (\mathcal{G}, \mathcal{B})$, i.e., $\forall i, j, \mathcal{T}^{\ell}_{\mathcal{A}_j}(\mathfrak{u}_i) = \mathcal{T}^{\ell}_{\mathcal{B}_j}(\mathfrak{u}_i), \mathcal{T}^{\mho}_{\mathcal{A}_j}(\mathfrak{u}_i) = \mathcal{T}^{\mho}_{\mathcal{B}_j}(\mathfrak{u}_i), \mathcal{J}^{\ell}_{\mathcal{A}_j}(\mathfrak{u}_i) = \mathcal{J}^{\ell}_{\mathcal{B}_j}(\mathfrak{u}_i)$, and $\mathcal{J}^{\mho}_{\mathcal{A}_j}(\mathfrak{u}_i) = \mathcal{J}^{\mho}_{\mathcal{B}_j}(\mathfrak{u}_i)$, then $\mathbb{C}_{WIVPFSS}((\mathcal{F}, \mathcal{A}), (\mathcal{G}, \mathcal{B})) = 1$.

**Proof 1.** $\mathbb{C}_{WIVPFSS}((\mathcal{F}, \mathcal{A}), (\mathcal{G}, \mathcal{B})) \geq 0$ is trivial. Now, we will prove $\mathbb{C}_{WIVPFSS}((\mathcal{F}, \mathcal{A}), (\mathcal{G}, \mathcal{B})) \leq 1$. Using Eq 4.3.

$$\mathcal{C}_{WIVPFSS}((\mathcal{F}, \mathcal{A}), (\mathcal{G}, \mathcal{B})) = \sum_{j=1}^m \gamma_j \left( \sum_{i=1}^n \Omega_i \left( \begin{array}{c} (\mathcal{T}^{\ell}_{\mathcal{A}_j}(\mathfrak{u}_i))^2 * (\mathcal{T}^{\ell}_{\mathcal{B}_j}(\mathfrak{u}_i))^2 + (\mathcal{T}^{\mho}_{\mathcal{A}_j}(\mathfrak{u}_i))^2 * (\mathcal{T}^{\mho}_{\mathcal{B}_j}(\mathfrak{u}_i))^2 + \\ (\mathcal{J}^{\ell}_{\mathcal{A}_j}(\mathfrak{u}_i))^2 * (\mathcal{J}^{\ell}_{\mathcal{B}_j}(\mathfrak{u}_i))^2 + (\mathcal{J}^{\mho}_{\mathcal{A}_j}(\mathfrak{u}_i))^2 * (\mathcal{J}^{\mho}_{\mathcal{B}_j}(\mathfrak{u}_i))^2 \end{array} \right) \right)$$

$$= \sum_{j=1}^m \gamma_j (\Omega_1 ((\mathcal{T}^{\ell}_{\mathcal{A}_j}(\mathfrak{u}_1))^2 * (\mathcal{T}^{\ell}_{\mathcal{B}_j}(\mathfrak{u}_1))^2 + (\mathcal{T}^{\mho}_{\mathcal{A}_j}(\mathfrak{u}_1))^2 * (\mathcal{T}^{\mho}_{\mathcal{B}_j}(\mathfrak{u}_1))^2 + (\mathcal{J}^{\ell}_{\mathcal{A}_j}(\mathfrak{u}_1))^2 * (\mathcal{J}^{\ell}_{\mathcal{B}_j}(\mathfrak{u}_1))^2$$

$$+ (\mathcal{J}^{\mho}_{\mathcal{A}_j}(\mathfrak{u}_1))^2 * (\mathcal{J}^{\mho}_{\mathcal{B}_j}(\mathfrak{u}_1))^2)) + \sum_{j=1}^m \gamma_j (\Omega_2 ((\mathcal{T}^{\ell}_{\mathcal{A}_j}(\mathfrak{u}_2))^2 * (\mathcal{T}^{\ell}_{\mathcal{B}_j}(\mathfrak{u}_2))^2 + (\mathcal{T}^{\mho}_{\mathcal{A}_j}(\mathfrak{u}_2))^2 * (\mathcal{T}^{\mho}_{\mathcal{B}_j}(\mathfrak{u}_2))^2$$

$$+ (\mathcal{J}^{\ell}_{\mathcal{A}_j}(\mathfrak{u}_2))^2 * (\mathcal{J}^{\ell}_{\mathcal{B}_j}(\mathfrak{u}_2))^2 + (\mathcal{J}^{\mho}_{\mathcal{A}_j}(\mathfrak{u}_2))^2 * (\mathcal{J}^{\mho}_{\mathcal{B}_j}(\mathfrak{u}_2))^2))$$

$$+$$

$$\vdots$$

$$+$$

$$\sum_{j=1}^m \gamma_j (\Omega_n ((\mathcal{T}^{\ell}_{\mathcal{A}_j}(\mathfrak{u}_n))^2 * (\mathcal{T}^{\ell}_{\mathcal{B}_j}(\mathfrak{u}_n))^2 + (\mathcal{T}^{\mho}_{\mathcal{A}_j}(\mathfrak{u}_n))^2 * (\mathcal{T}^{\mho}_{\mathcal{B}_j}(\mathfrak{u}_n))^2 + (\mathcal{J}^{\ell}_{\mathcal{A}_j}(\mathfrak{u}_n))^2 * (\mathcal{J}^{\ell}_{\mathcal{B}_j}(\mathfrak{u}_n))^2$$

$$+ (\mathcal{J}^{\mho}_{\mathcal{A}_j}(\mathfrak{u}_n))^2 * (\mathcal{J}^{\mho}_{\mathcal{B}_j}(\mathfrak{u}_n))^2))$$

$$\mathcal{C}_{WIVPFSS}((\mathcal{F}, \mathcal{A}), (\mathcal{G}, \mathcal{B}))$$

$$= \left\{ \begin{array}{l} \gamma_1 (\Omega_1 ((\mathcal{T}^{\ell}_{\mathcal{A}_1}(\mathfrak{u}_1))^2 * (\mathcal{T}^{\ell}_{\mathcal{B}_1}(\mathfrak{u}_1))^2 + (\mathcal{T}^{\mho}_{\mathcal{A}_1}(\mathfrak{u}_1))^2 * (\mathcal{T}^{\mho}_{\mathcal{B}_1}(\mathfrak{u}_1))^2 + (\mathcal{J}^{\ell}_{\mathcal{A}_1}(\mathfrak{u}_1))^2 * (\mathcal{J}^{\ell}_{\mathcal{B}_1}(\mathfrak{u}_1))^2 + (\mathcal{J}^{\mho}_{\mathcal{A}_1}(\mathfrak{u}_1))^2 * (\mathcal{J}^{\mho}_{\mathcal{B}_1}(\mathfrak{u}_1))^2)) + \\ \gamma_2 (\Omega_1 ((\mathcal{T}^{\ell}_{\mathcal{A}_2}(\mathfrak{u}_1))^2 * (\mathcal{T}^{\ell}_{\mathcal{B}_2}(\mathfrak{u}_1))^2 + (\mathcal{T}^{\mho}_{\mathcal{A}_2}(\mathfrak{u}_1))^2 * (\mathcal{T}^{\mho}_{\mathcal{B}_2}(\mathfrak{u}_1))^2 + (\mathcal{J}^{\ell}_{\mathcal{A}_2}(\mathfrak{u}_1))^2 * (\mathcal{J}^{\ell}_{\mathcal{B}_2}(\mathfrak{u}_1))^2 + (\mathcal{J}^{\mho}_{\mathcal{A}_2}(\mathfrak{u}_1))^2 * (\mathcal{J}^{\mho}_{\mathcal{B}_2}(\mathfrak{u}_1))^2)) + \\ \vdots \\ + \\ \gamma_m (\Omega_1 ((\mathcal{T}^{\ell}_{\mathcal{A}_m}(\mathfrak{u}_1))^2 * (\mathcal{T}^{\ell}_{\mathcal{B}_m}(\mathfrak{u}_1))^2 + (\mathcal{T}^{\mho}_{\mathcal{A}_m}(\mathfrak{u}_1))^2 * (\mathcal{T}^{\mho}_{\mathcal{B}_m}(\mathfrak{u}_1))^2 + (\mathcal{J}^{\ell}_{\mathcal{A}_m}(\mathfrak{u}_1))^2 * (\mathcal{J}^{\ell}_{\mathcal{B}_m}(\mathfrak{u}_1))^2 + (\mathcal{J}^{\mho}_{\mathcal{A}_m}(\mathfrak{u}_1))^2 * (\mathcal{J}^{\mho}_{\mathcal{B}_m}(\mathfrak{u}_1))^2)) \end{array} \right\}$$

$$+\left\{\begin{array}{l}\gamma_1(\Omega_2((\mathcal{T}^\ell_{\mathcal{A}_1}(\mathfrak{u}_2))^2*(\mathcal{T}^\ell_{\mathcal{B}_1}(\mathfrak{u}_2))^2+(\mathcal{T}^\upsilon_{\mathcal{A}_1}(\mathfrak{u}_2))^2*(\mathcal{T}^\upsilon_{\mathcal{B}_1}(\mathfrak{u}_2))^2+(\mathcal{J}^\ell_{\mathcal{A}_1}(\mathfrak{u}_2))^2*(\mathcal{J}^\ell_{\mathcal{B}_1}(\mathfrak{u}_2))^2+(\mathcal{J}^\upsilon_{\mathcal{A}_1}(\mathfrak{u}_2))^2*(\mathcal{J}^\upsilon_{\mathcal{B}_1}(\mathfrak{u}_2))^2))+\\ \gamma_2(\Omega_2((\mathcal{T}^\ell_{\mathcal{A}_2}(\mathfrak{u}_2))^2*(\mathcal{T}^\ell_{\mathcal{B}_2}(\mathfrak{u}_2))^2+(\mathcal{T}^\upsilon_{\mathcal{A}_2}(\mathfrak{u}_2))^2*(\mathcal{T}^\upsilon_{\mathcal{B}_2}(\mathfrak{u}_2))^2+(\mathcal{J}^\ell_{\mathcal{A}_2}(\mathfrak{u}_2))^2*(\mathcal{J}^\ell_{\mathcal{B}_2}(\mathfrak{u}_2))^2+(\mathcal{J}^\upsilon_{\mathcal{A}_2}(\mathfrak{u}_2))^2*(\mathcal{J}^\upsilon_{\mathcal{B}_2}(\mathfrak{u}_2))^2))+\\ \vdots \\ + \\ \gamma_m(\Omega_2((\mathcal{T}^\ell_{\mathcal{A}_m}(\mathfrak{u}_2))^2*(\mathcal{T}^\ell_{\mathcal{B}_m}(\mathfrak{u}_2))^2+(\mathcal{T}^\upsilon_{\mathcal{A}_m}(\mathfrak{u}_2))^2*(\mathcal{T}^\upsilon_{\mathcal{B}_m}(\mathfrak{u}_2))^2+(\mathcal{J}^\ell_{\mathcal{A}_m}(\mathfrak{u}_2))^2*(\mathcal{J}^\ell_{\mathcal{B}_m}(\mathfrak{u}_2))^2+(\mathcal{J}^\upsilon_{\mathcal{A}_m}(\mathfrak{u}_2))^2*(\mathcal{J}^\upsilon_{\mathcal{B}_m}(\mathfrak{u}_2))^2))\end{array}\right\}$$

$$+$$

$$\vdots$$

$$+$$

$$\left\{\begin{array}{l}\gamma_1(\Omega_n((\mathcal{T}^\ell_{\mathcal{A}_1}(\mathfrak{u}_n))^2*(\mathcal{T}^\ell_{\mathcal{B}_1}(\mathfrak{u}_n))^2+(\mathcal{T}^\upsilon_{\mathcal{A}_1}(\mathfrak{u}_n))^2*(\mathcal{T}^\upsilon_{\mathcal{B}_1}(\mathfrak{u}_n))^2+(\mathcal{J}^\ell_{\mathcal{A}_1}(\mathfrak{u}_n))^2*(\mathcal{J}^\ell_{\mathcal{B}_1}(\mathfrak{u}_n))^2+(\mathcal{J}^\upsilon_{\mathcal{A}_1}(\mathfrak{u}_n))^2*(\mathcal{J}^\upsilon_{\mathcal{B}_1}(\mathfrak{u}_n))^2))+\\ \gamma_2(\Omega_n((\mathcal{T}^\ell_{\mathcal{A}_2}(\mathfrak{u}_n))^2*(\mathcal{T}^\ell_{\mathcal{B}_2}(\mathfrak{u}_n))^2+(\mathcal{T}^\upsilon_{\mathcal{A}_2}(\mathfrak{u}_n))^2*(\mathcal{T}^\upsilon_{\mathcal{B}_2}(\mathfrak{u}_n))^2+(\mathcal{J}^\ell_{\mathcal{A}_2}(\mathfrak{u}_n))^2*(\mathcal{J}^\ell_{\mathcal{B}_2}(\mathfrak{u}_n))^2+(\mathcal{J}^\upsilon_{\mathcal{A}_2}(\mathfrak{u}_n))^2*(\mathcal{J}^\upsilon_{\mathcal{B}_2}(\mathfrak{u}_n))^2))+\\ \vdots \\ + \\ \gamma_m(\Omega_n((\mathcal{T}^\ell_{\mathcal{A}_m}(\mathfrak{u}_n))^2*(\mathcal{T}^\ell_{\mathcal{B}_m}(\mathfrak{u}_n))^2+(\mathcal{T}^\upsilon_{\mathcal{A}_m}(\mathfrak{u}_n))^2*(\mathcal{T}^\upsilon_{\mathcal{B}_m}(\mathfrak{u}_n))^2+(\mathcal{J}^\ell_{\mathcal{A}_m}(\mathfrak{u}_n))^2*(\mathcal{J}^\ell_{\mathcal{B}_m}(\mathfrak{u}_n))^2+(\mathcal{J}^\upsilon_{\mathcal{A}_m}(\mathfrak{u}_n))^2*(\mathcal{J}^\upsilon_{\mathcal{B}_m}(\mathfrak{u}_n))^2))\end{array}\right\}$$

$$=\left\{\begin{array}{l}\gamma_1\left(\begin{array}{l}\sqrt{\Omega_1}(\mathcal{T}^\ell_{\mathcal{A}_1}(\mathfrak{u}_1))^2*\sqrt{\Omega_1}(\mathcal{T}^\ell_{\mathcal{B}_1}(\mathfrak{u}_1))^2+\sqrt{\Omega_1}(\mathcal{T}^\upsilon_{\mathcal{A}_1}(\mathfrak{u}_1))^2*\sqrt{\Omega_1}(\mathcal{T}^\upsilon_{\mathcal{B}_1}(\mathfrak{u}_1))^2+\\ \sqrt{\Omega_1}(\mathcal{J}^\ell_{\mathcal{A}_1}(\mathfrak{u}_1))^2*\sqrt{\Omega_1}(\mathcal{J}^\ell_{\mathcal{B}_1}(\mathfrak{u}_1))^2+\sqrt{\Omega_1}(\mathcal{J}^\upsilon_{\mathcal{A}_1}(\mathfrak{u}_1))^2*\sqrt{\Omega_1}(\mathcal{J}^\upsilon_{\mathcal{B}_1}(\mathfrak{u}_1))^2\end{array}\right)+\\ \gamma_2\left(\begin{array}{l}\sqrt{\Omega_1}(\mathcal{T}^\ell_{\mathcal{A}_2}(\mathfrak{u}_1))^2*\sqrt{\Omega_1}(\mathcal{T}^\ell_{\mathcal{B}_2}(\mathfrak{u}_1))^2+\sqrt{\Omega_1}(\mathcal{T}^\upsilon_{\mathcal{A}_2}(\mathfrak{u}_1))^2*\sqrt{\Omega_1}(\mathcal{T}^\upsilon_{\mathcal{B}_2}(\mathfrak{u}_1))^2+\\ \sqrt{\Omega_1}(\mathcal{J}^\ell_{\mathcal{A}_2}(\mathfrak{u}_1))^2*\sqrt{\Omega_1}(\mathcal{J}^\ell_{\mathcal{B}_2}(\mathfrak{u}_1))^2+\sqrt{\Omega_1}(\mathcal{J}^\upsilon_{\mathcal{A}_2}(\mathfrak{u}_1))^2*\sqrt{\Omega_1}(\mathcal{J}^\upsilon_{\mathcal{B}_2}(\mathfrak{u}_1))^2\end{array}\right)+\\ \vdots \\ + \\ \gamma_m\left(\begin{array}{l}\sqrt{\Omega_1}(\mathcal{T}^\ell_{\mathcal{A}_m}(\mathfrak{u}_1))^2*\sqrt{\Omega_1}(\mathcal{T}^\ell_{\mathcal{B}_m}(\mathfrak{u}_1))^2+\sqrt{\Omega_1}(\mathcal{T}^\upsilon_{\mathcal{A}_m}(\mathfrak{u}_1))^2*\sqrt{\Omega_1}(\mathcal{T}^\upsilon_{\mathcal{B}_m}(\mathfrak{u}_1))^2+\\ \sqrt{\Omega_1}(\mathcal{J}^\ell_{\mathcal{A}_m}(\mathfrak{u}_1))^2*\sqrt{\Omega_1}(\mathcal{J}^\ell_{\mathcal{B}_m}(\mathfrak{u}_1))^2+\sqrt{\Omega_1}(\mathcal{J}^\upsilon_{\mathcal{A}_m}(\mathfrak{u}_1))^2*\sqrt{\Omega_1}(\mathcal{J}^\upsilon_{\mathcal{B}_m}(\mathfrak{u}_1))^2\end{array}\right)\end{array}\right\}$$

$$+\left\{\begin{array}{l}\gamma_1\left(\begin{array}{l}\sqrt{\Omega_2}(\mathcal{T}^\ell_{\mathcal{A}_1}(\mathfrak{u}_2))^2*\sqrt{\Omega_2}(\mathcal{T}^\ell_{\mathcal{B}_1}(\mathfrak{u}_2))^2+\sqrt{\Omega_2}(\mathcal{T}^\upsilon_{\mathcal{A}_1}(\mathfrak{u}_2))^2*\sqrt{\Omega_2}(\mathcal{T}^\upsilon_{\mathcal{B}_1}(\mathfrak{u}_2))^2+\\ \sqrt{\Omega_2}(\mathcal{J}^\ell_{\mathcal{A}_1}(\mathfrak{u}_2))^2*\sqrt{\Omega_2}(\mathcal{J}^\ell_{\mathcal{B}_1}(\mathfrak{u}_2))^2+\sqrt{\Omega_2}(\mathcal{J}^\upsilon_{\mathcal{A}_1}(\mathfrak{u}_2))^2*\sqrt{\Omega_2}(\mathcal{J}^\upsilon_{\mathcal{B}_1}(\mathfrak{u}_2))^2\end{array}\right)+\\ \gamma_2\left(\begin{array}{l}\sqrt{\Omega_2}(\mathcal{T}^\ell_{\mathcal{A}_2}(\mathfrak{u}_2))^2*\sqrt{\Omega_2}(\mathcal{T}^\ell_{\mathcal{B}_2}(\mathfrak{u}_2))^2+\sqrt{\Omega_2}(\mathcal{T}^\upsilon_{\mathcal{A}_2}(\mathfrak{u}_2))^2*\sqrt{\Omega_2}(\mathcal{T}^\upsilon_{\mathcal{B}_2}(\mathfrak{u}_2))^2+\\ \sqrt{\Omega_2}(\mathcal{J}^\ell_{\mathcal{A}_2}(\mathfrak{u}_2))^2*\sqrt{\Omega_2}(\mathcal{J}^\ell_{\mathcal{B}_2}(\mathfrak{u}_2))^2+\sqrt{\Omega_2}(\mathcal{J}^\upsilon_{\mathcal{A}_2}(\mathfrak{u}_2))^2*\sqrt{\Omega_2}(\mathcal{J}^\upsilon_{\mathcal{B}_2}(\mathfrak{u}_2))^2\end{array}\right)+\\ \vdots \\ + \\ \gamma_m\left(\begin{array}{l}\sqrt{\Omega_2}(\mathcal{T}^\ell_{\mathcal{A}_m}(\mathfrak{u}_2))^2*\sqrt{\Omega_2}(\mathcal{T}^\ell_{\mathcal{B}_m}(\mathfrak{u}_2))^2+\sqrt{\Omega_2}(\mathcal{T}^\upsilon_{\mathcal{A}_m}(\mathfrak{u}_2))^2*\sqrt{\Omega_2}(\mathcal{T}^\upsilon_{\mathcal{B}_m}(\mathfrak{u}_2))^2+\\ \sqrt{\Omega_2}(\mathcal{J}^\ell_{\mathcal{A}_m}(\mathfrak{u}_2))^2*\sqrt{\Omega_2}(\mathcal{J}^\ell_{\mathcal{B}_m}(\mathfrak{u}_2))^2+\sqrt{\Omega_2}(\mathcal{J}^\upsilon_{\mathcal{A}_m}(\mathfrak{u}_2))^2*\sqrt{\Omega_2}(\mathcal{J}^\upsilon_{\mathcal{B}_m}(\mathfrak{u}_2))^2\end{array}\right)\end{array}\right\}$$

$$+$$

$$\vdots$$

$$+$$

$$\left\{ \begin{array}{l} \gamma_1 \left( \begin{array}{l} \sqrt{\Omega_n}(\mathcal{T}^{\ell}_{\mathcal{A}_1}(\mathfrak{u}_n))^2 * \sqrt{\Omega_n}(\mathcal{T}^{\ell}_{\mathcal{B}_1}(\mathfrak{u}_n))^2 + \sqrt{\Omega_n}(\mathcal{T}^{\mho}_{\mathcal{A}_1}(\mathfrak{u}_n))^2 * \sqrt{\Omega_n}(\mathcal{T}^{\mho}_{\mathcal{B}_1}(\mathfrak{u}_n))^2 + \\ \sqrt{\Omega_n}(\mathcal{J}^{\ell}_{\mathcal{A}_1}(\mathfrak{u}_n))^2 * \sqrt{\Omega_n}(\mathcal{J}^{\ell}_{\mathcal{B}_1}(\mathfrak{u}_n))^2 + \sqrt{\Omega_n}(\mathcal{J}^{\mho}_{\mathcal{A}_1}(\mathfrak{u}_n))^2 * \sqrt{\Omega_n}(\mathcal{J}^{\mho}_{\mathcal{B}_1}(\mathfrak{u}_n))^2 \end{array} \right) + \\ \gamma_2 \left( \begin{array}{l} \sqrt{\Omega_n}(\mathcal{T}^{\ell}_{\mathcal{A}_2}(\mathfrak{u}_n))^2 * \sqrt{\Omega_n}(\mathcal{T}^{\ell}_{\mathcal{B}_2}(\mathfrak{u}_n))^2 + \sqrt{\Omega_n}(\mathcal{T}^{\mho}_{\mathcal{A}_2}(\mathfrak{u}_n))^2 * \sqrt{\Omega_n}(\mathcal{T}^{\mho}_{\mathcal{B}_2}(\mathfrak{u}_n))^2 + \\ \sqrt{\Omega_n}(\mathcal{J}^{\ell}_{\mathcal{A}_2}(\mathfrak{u}_n))^2 * \sqrt{\Omega_n}(\mathcal{J}^{\ell}_{\mathcal{B}_2}(\mathfrak{u}_n))^2 + \sqrt{\Omega_n}(\mathcal{J}^{\mho}_{\mathcal{A}_2}(\mathfrak{u}_n))^2 * \sqrt{\Omega_n}(\mathcal{J}^{\mho}_{\mathcal{B}_2}(\mathfrak{u}_n))^2 \end{array} \right) + \\ \vdots \\ + \\ \gamma_m \left( \begin{array}{l} \sqrt{\Omega_n}(\mathcal{T}^{\ell}_{\mathcal{A}_m}(\mathfrak{u}_n))^2 * \sqrt{\Omega_n}(\mathcal{T}^{\ell}_{\mathcal{B}_m}(\mathfrak{u}_n))^2 + \sqrt{\Omega_n}(\mathcal{T}^{\mho}_{\mathcal{A}_m}(\mathfrak{u}_n))^2 * \sqrt{\Omega_n}(\mathcal{T}^{\mho}_{\mathcal{B}_m}(\mathfrak{u}_n))^2 + \\ \sqrt{\Omega_n}(\mathcal{J}^{\ell}_{\mathcal{A}_m}(\mathfrak{u}_n))^2 * \sqrt{\Omega_n}(\mathcal{J}^{\ell}_{\mathcal{B}_m}(\mathfrak{u}_n))^2 + \sqrt{\Omega_n}(\mathcal{J}^{\mho}_{\mathcal{A}_m}(\mathfrak{u}_n))^2 * \sqrt{\Omega_n}(\mathcal{J}^{\mho}_{\mathcal{B}_m}(\mathfrak{u}_n))^2 \end{array} \right) \end{array} \right\}$$

$$= \left\{ \begin{array}{l} \left( \begin{array}{l} \sqrt{\gamma_1}\sqrt{\Omega_1}(\mathcal{T}^{\ell}_{\mathcal{A}_1}(\mathfrak{u}_1))^2 * \sqrt{\gamma_1}\sqrt{\Omega_1}(\mathcal{T}^{\ell}_{\mathcal{B}_1}(\mathfrak{u}_1))^2 + \sqrt{\gamma_1}\sqrt{\Omega_1}(\mathcal{T}^{\mho}_{\mathcal{A}_1}(\mathfrak{u}_1))^2 * \sqrt{\gamma_1}\sqrt{\Omega_1}(\mathcal{T}^{\mho}_{\mathcal{B}_1}(\mathfrak{u}_1))^2 + \\ \sqrt{\gamma_1}\sqrt{\Omega_1}(\mathcal{J}^{\ell}_{\mathcal{A}_1}(\mathfrak{u}_1))^2 * \sqrt{\gamma_1}\sqrt{\Omega_1}(\mathcal{J}^{\ell}_{\mathcal{B}_1}(\mathfrak{u}_1))^2 + \sqrt{\gamma_1}\sqrt{\Omega_1}(\mathcal{J}^{\mho}_{\mathcal{A}_1}(\mathfrak{u}_1))^2 * \sqrt{\gamma_1}\sqrt{\Omega_1}(\mathcal{J}^{\mho}_{\mathcal{B}_1}(\mathfrak{u}_1))^2 \end{array} \right) + \\ \left( \begin{array}{l} \sqrt{\gamma_2}\sqrt{\Omega_1}(\mathcal{T}^{\ell}_{\mathcal{A}_2}(\mathfrak{u}_1))^2 * \sqrt{\gamma_2}\sqrt{\Omega_1}(\mathcal{T}^{\ell}_{\mathcal{B}_2}(\mathfrak{u}_1))^2 + \sqrt{\gamma_2}\sqrt{\Omega_1}(\mathcal{T}^{\mho}_{\mathcal{A}_2}(\mathfrak{u}_1))^2 * \sqrt{\gamma_2}\sqrt{\Omega_1}(\mathcal{T}^{\mho}_{\mathcal{B}_2}(\mathfrak{u}_1))^2 + \\ \sqrt{\gamma_2}\sqrt{\Omega_1}(\mathcal{J}^{\ell}_{\mathcal{A}_2}(\mathfrak{u}_1))^2 * \sqrt{\gamma_2}\sqrt{\Omega_1}(\mathcal{J}^{\ell}_{\mathcal{B}_2}(\mathfrak{u}_1))^2 + \sqrt{\gamma_2}\sqrt{\Omega_1}(\mathcal{J}^{\mho}_{\mathcal{A}_2}(\mathfrak{u}_1))^2 * \sqrt{\gamma_2}\sqrt{\Omega_1}(\mathcal{J}^{\mho}_{\mathcal{B}_2}(\mathfrak{u}_1))^2 \end{array} \right) + \\ \vdots \\ + \\ \left( \begin{array}{l} \sqrt{\gamma_m}\sqrt{\Omega_1}(\mathcal{T}^{\ell}_{\mathcal{A}_m}(\mathfrak{u}_1))^2 * \sqrt{\gamma_m}\sqrt{\Omega_1}(\mathcal{T}^{\ell}_{\mathcal{B}_m}(\mathfrak{u}_1))^2 + \sqrt{\gamma_m}\sqrt{\Omega_1}(\mathcal{T}^{\mho}_{\mathcal{A}_m}(\mathfrak{u}_1))^2 * \sqrt{\gamma_m}\sqrt{\Omega_1}(\mathcal{T}^{\mho}_{\mathcal{B}_m}(\mathfrak{u}_1))^2 + \\ \sqrt{\gamma_m}\sqrt{\Omega_1}(\mathcal{J}^{\ell}_{\mathcal{A}_m}(\mathfrak{u}_1))^2 * \sqrt{\gamma_m}\sqrt{\Omega_1}(\mathcal{J}^{\ell}_{\mathcal{B}_m}(\mathfrak{u}_1))^2 + \sqrt{\gamma_m}\sqrt{\Omega_1}(\mathcal{J}^{\mho}_{\mathcal{A}_m}(\mathfrak{u}_1))^2 * \sqrt{\gamma_m}\sqrt{\Omega_1}(\mathcal{J}^{\mho}_{\mathcal{B}_m}(\mathfrak{u}_1))^2 \end{array} \right) \end{array} \right\}$$

$$+ \left\{ \begin{array}{l} \left( \begin{array}{l} \sqrt{\gamma_1}\sqrt{\Omega_2}(\mathcal{T}^{\ell}_{\mathcal{A}_1}(\mathfrak{u}_2))^2 * \sqrt{\gamma_1}\sqrt{\Omega_2}(\mathcal{T}^{\ell}_{\mathcal{B}_1}(\mathfrak{u}_2))^2 + \sqrt{\gamma_1}\sqrt{\Omega_2}(\mathcal{T}^{\mho}_{\mathcal{A}_1}(\mathfrak{u}_2))^2 * \sqrt{\gamma_1}\sqrt{\Omega_2}(\mathcal{T}^{\mho}_{\mathcal{B}_1}(\mathfrak{u}_2))^2 + \\ \sqrt{\gamma_1}\sqrt{\Omega_2}(\mathcal{J}^{\ell}_{\mathcal{A}_1}(\mathfrak{u}_2))^2 * \sqrt{\gamma_1}\sqrt{\Omega_2}(\mathcal{J}^{\ell}_{\mathcal{B}_1}(\mathfrak{u}_2))^2 + \sqrt{\gamma_1}\sqrt{\Omega_2}(\mathcal{J}^{\mho}_{\mathcal{A}_1}(\mathfrak{u}_2))^2 * \sqrt{\gamma_1}\sqrt{\Omega_2}(\mathcal{J}^{\mho}_{\mathcal{B}_1}(\mathfrak{u}_2))^2 \end{array} \right) + \\ \left( \begin{array}{l} \sqrt{\gamma_2}\sqrt{\Omega_2}(\mathcal{T}^{\ell}_{\mathcal{A}_2}(\mathfrak{u}_2))^2 * \sqrt{\gamma_2}\sqrt{\Omega_2}(\mathcal{T}^{\ell}_{\mathcal{B}_2}(\mathfrak{u}_2))^2 + \sqrt{\gamma_2}\sqrt{\Omega_2}(\mathcal{T}^{\mho}_{\mathcal{A}_2}(\mathfrak{u}_2))^2 * \sqrt{\gamma_2}\sqrt{\Omega_2}(\mathcal{T}^{\mho}_{\mathcal{B}_2}(\mathfrak{u}_2))^2 + \\ \sqrt{\gamma_2}\sqrt{\Omega_2}(\mathcal{J}^{\ell}_{\mathcal{A}_2}(\mathfrak{u}_2))^2 * \sqrt{\gamma_2}\sqrt{\Omega_2}(\mathcal{J}^{\ell}_{\mathcal{B}_2}(\mathfrak{u}_2))^2 + \sqrt{\gamma_2}\sqrt{\Omega_2}(\mathcal{J}^{\mho}_{\mathcal{A}_2}(\mathfrak{u}_2))^2 * \sqrt{\gamma_2}\sqrt{\Omega_2}(\mathcal{J}^{\mho}_{\mathcal{B}_2}(\mathfrak{u}_2))^2 \end{array} \right) + \\ \vdots \\ + \\ \left( \begin{array}{l} \sqrt{\gamma_m}\sqrt{\Omega_2}(\mathcal{T}^{\ell}_{\mathcal{A}_m}(\mathfrak{u}_2))^2 * \sqrt{\gamma_m}\sqrt{\Omega_2}(\mathcal{T}^{\ell}_{\mathcal{B}_m}(\mathfrak{u}_2))^2 + \sqrt{\gamma_m}\sqrt{\Omega_2}(\mathcal{T}^{\mho}_{\mathcal{A}_m}(\mathfrak{u}_2))^2 * \sqrt{\gamma_m}\sqrt{\Omega_2}(\mathcal{T}^{\mho}_{\mathcal{B}_m}(\mathfrak{u}_2))^2 + \\ \sqrt{\gamma_m}\sqrt{\Omega_2}(\mathcal{J}^{\ell}_{\mathcal{A}_m}(\mathfrak{u}_2))^2 * \sqrt{\gamma_m}\sqrt{\Omega_2}(\mathcal{J}^{\ell}_{\mathcal{B}_m}(\mathfrak{u}_2))^2 + \sqrt{\gamma_m}\sqrt{\Omega_2}(\mathcal{J}^{\mho}_{\mathcal{A}_m}(\mathfrak{u}_2))^2 * \sqrt{\gamma_m}\sqrt{\Omega_2}(\mathcal{J}^{\mho}_{\mathcal{B}_m}(\mathfrak{u}_2))^2 \end{array} \right) \end{array} \right\}$$

$$+$$

$$\vdots$$

$$+$$

$$
\left\{
\begin{array}{l}
\left(
\begin{array}{l}
\sqrt{\gamma_1}\sqrt{\Omega_n}(\mathcal{T}^{\ell}_{\mathcal{A}_1}(\mathfrak{u}_n))^2 * \sqrt{\gamma_1}\sqrt{\Omega_n}(\mathcal{T}^{\ell}_{\mathcal{B}_1}(\mathfrak{u}_n))^2 + \sqrt{\gamma_1}\sqrt{\Omega_n}(\mathcal{T}^{\mho}_{\mathcal{A}_1}(\mathfrak{u}_n))^2 * \sqrt{\gamma_1}\sqrt{\Omega_n}(\mathcal{T}^{\mho}_{\mathcal{B}_1}(\mathfrak{u}_n))^2 + \\
\sqrt{\gamma_1}\sqrt{\Omega_n}(\mathcal{J}^{\ell}_{\mathcal{A}_1}(\mathfrak{u}_n))^2 * \sqrt{\gamma_1}\sqrt{\Omega_n}(\mathcal{J}^{\ell}_{\mathcal{B}_1}(\mathfrak{u}_n))^2 + \sqrt{\gamma_1}\sqrt{\Omega_n}(\mathcal{J}^{\mho}_{\mathcal{A}_1}(\mathfrak{u}_n))^2 * \sqrt{\gamma_1}\sqrt{\Omega_n}(\mathcal{J}^{\mho}_{\mathcal{B}_1}(\mathfrak{u}_n))^2
\end{array}
\right)+ \\[2ex]
\left(
\begin{array}{l}
\sqrt{\gamma_2}\sqrt{\Omega_n}(\mathcal{T}^{\ell}_{\mathcal{A}_2}(\mathfrak{u}_n))^2 * \sqrt{\gamma_2}\sqrt{\Omega_n}(\mathcal{T}^{\ell}_{\mathcal{B}_2}(\mathfrak{u}_n))^2 + \sqrt{\gamma_2}\sqrt{\Omega_n}(\mathcal{T}^{\mho}_{\mathcal{A}_2}(\mathfrak{u}_n))^2 * \sqrt{\gamma_2}\sqrt{\Omega_n}(\mathcal{T}^{\mho}_{\mathcal{B}_2}(\mathfrak{u}_n))^2 + \\
\sqrt{\gamma_2}\sqrt{\Omega_n}(\mathcal{J}^{\ell}_{\mathcal{A}_2}(\mathfrak{u}_n))^2 * \sqrt{\gamma_2}\sqrt{\Omega_n}(\mathcal{J}^{\ell}_{\mathcal{B}_2}(\mathfrak{u}_n))^2 + \sqrt{\gamma_2}\sqrt{\Omega_n}(\mathcal{J}^{\mho}_{\mathcal{A}_2}(\mathfrak{u}_n))^2 * \sqrt{\gamma_2}\sqrt{\Omega_n}(\mathcal{J}^{\mho}_{\mathcal{B}_2}(\mathfrak{u}_n))^2
\end{array}
\right)+ \\[2ex]
\vdots \\[1ex]
+ \\[1ex]
\left(
\begin{array}{l}
\sqrt{\gamma_m}\sqrt{\Omega_n}(\mathcal{T}^{\ell}_{\mathcal{A}_m}(\mathfrak{u}_n))^2 * \sqrt{\gamma_m}\sqrt{\Omega_n}(\mathcal{T}^{\ell}_{\mathcal{B}_m}(\mathfrak{u}_n))^2 + \sqrt{\gamma_m}\sqrt{\Omega_n}(\mathcal{T}^{\mho}_{\mathcal{A}_m}(\mathfrak{u}_n))^2 * \sqrt{\gamma_m}\sqrt{\Omega_n}(\mathcal{T}^{\mho}_{\mathcal{B}_m}(\mathfrak{u}_n))^2 + \\
\sqrt{\gamma_m}\sqrt{\Omega_n}(\mathcal{J}^{\ell}_{\mathcal{A}_m}(\mathfrak{u}_n))^2 * \sqrt{\gamma_m}\sqrt{\Omega_n}(\mathcal{J}^{\ell}_{\mathcal{B}_m}(\mathfrak{u}_n))^2 + \sqrt{\gamma_m}\sqrt{\Omega_n}(\mathcal{J}^{\mho}_{\mathcal{A}_m}(\mathfrak{u}_n))^2 * \sqrt{\gamma_m}\sqrt{\Omega_n}(\mathcal{J}^{\mho}_{\mathcal{B}_m}(\mathfrak{u}_n))^2
\end{array}
\right)
\end{array}
\right\}
$$

Using Cauchy-Schwarz inequality

$$\mathcal{C}_{WIVPFSS}((\mathcal{F},\mathcal{A}),(\mathcal{G},\mathcal{B}))^2 \leq$$

$$
\left\{
\begin{array}{l}
\gamma_1\Omega_1\{(\mathcal{T}^{\ell}_{\mathcal{A}_1}(\mathfrak{u}_1))^4 + (\mathcal{T}^{\mho}_{\mathcal{A}_1}(\mathfrak{u}_1))^4 + (\mathcal{J}^{\ell}_{\mathcal{A}_1}(\mathfrak{u}_1))^4 + (\mathcal{J}^{\mho}_{\mathcal{A}_1}(\mathfrak{u}_1))^4\}+ \\[2ex]
\gamma_2\Omega_1\{(\mathcal{T}^{\ell}_{\mathcal{A}_2}(\mathfrak{u}_1))^4 + (\mathcal{T}^{\mho}_{\mathcal{A}_2}(\mathfrak{u}_1))^4 + (\mathcal{J}^{\ell}_{\mathcal{A}_2}(\mathfrak{u}_1))^4 + (\mathcal{J}^{\mho}_{\mathcal{A}_2}(\mathfrak{u}_1))^4\}+ \\[2ex]
\left(\qquad\qquad\qquad\qquad\qquad \vdots \qquad\qquad\qquad\qquad\qquad\right)+ \\[1ex]
+ \\[1ex]
\gamma_m\Omega_1\{(\mathcal{T}^{\ell}_{\mathcal{A}_m}(\mathfrak{u}_1))^4 + (\mathcal{T}^{\mho}_{\mathcal{A}_m}(\mathfrak{u}_1))^4 + (\mathcal{J}^{\ell}_{\mathcal{A}_m}(\mathfrak{u}_1))^4 + (\mathcal{J}^{\mho}_{\mathcal{A}_m}(\mathfrak{u}_1))^4\} \\[2ex]
\gamma_1\Omega_2\{(\mathcal{T}^{\ell}_{\mathcal{A}_1}(\mathfrak{u}_2))^4 + (\mathcal{T}^{\mho}_{\mathcal{A}_1}(\mathfrak{u}_2))^4 + (\mathcal{J}^{\ell}_{\mathcal{A}_1}(\mathfrak{u}_2))^4 + (\mathcal{J}^{\mho}_{\mathcal{A}_1}(\mathfrak{u}_2))^4\}+ \\[2ex]
\gamma_2\Omega_2\{(\mathcal{T}^{\ell}_{\mathcal{A}_2}(\mathfrak{u}_2))^4 + (\mathcal{T}^{\mho}_{\mathcal{A}_2}(\mathfrak{u}_2))^4 + (\mathcal{J}^{\ell}_{\mathcal{A}_2}(\mathfrak{u}_2))^4 + (\mathcal{J}^{\mho}_{\mathcal{A}_2}(\mathfrak{u}_2))^4\}+ \\[2ex]
\left(\qquad\qquad\qquad\qquad\qquad \vdots \qquad\qquad\qquad\qquad\qquad\right)+ \\[1ex]
+ \\[1ex]
\gamma_m\Omega_2\{(\mathcal{T}^{\ell}_{\mathcal{A}_m}(\mathfrak{u}_2))^4 + (\mathcal{T}^{\mho}_{\mathcal{A}_m}(\mathfrak{u}_2))^4 + (\mathcal{J}^{\ell}_{\mathcal{A}_m}(\mathfrak{u}_2))^4 + (\mathcal{J}^{\mho}_{\mathcal{A}_m}(\mathfrak{u}_2))^4\} \\[2ex]
\vdots \\[1ex]
+ \\[1ex]
\gamma_1\Omega_n\{(\mathcal{T}^{\ell}_{\mathcal{A}_1}(\mathfrak{u}_n))^4 + (\mathcal{T}^{\mho}_{\mathcal{A}_1}(\mathfrak{u}_n))^4 + (\mathcal{J}^{\ell}_{\mathcal{A}_1}(\mathfrak{u}_n))^4 + (\mathcal{J}^{\mho}_{\mathcal{A}_1}(\mathfrak{u}_n))^4\}+ \\[2ex]
\gamma_2\Omega_n\{(\mathcal{T}^{\ell}_{\mathcal{A}_2}(\mathfrak{u}_n))^4 + (\mathcal{T}^{\mho}_{\mathcal{A}_2}(\mathfrak{u}_n))^4 + (\mathcal{J}^{\ell}_{\mathcal{A}_2}(\mathfrak{u}_n))^4 + (\mathcal{J}^{\mho}_{\mathcal{A}_2}(\mathfrak{u}_n))^4\}+ \\[2ex]
\left(\qquad\qquad\qquad\qquad\qquad \vdots \qquad\qquad\qquad\qquad\qquad\right) \\[1ex]
+ \\[1ex]
\gamma_m\Omega_n\{(\mathcal{T}^{\ell}_{\mathcal{A}_m}(\mathfrak{u}_n))^4 + (\mathcal{T}^{\mho}_{\mathcal{A}_m}(\mathfrak{u}_n))^4 + (\mathcal{J}^{\ell}_{\mathcal{A}_m}(\mathfrak{u}_n))^4 + (\mathcal{J}^{\mho}_{\mathcal{A}_m}(\mathfrak{u}_n))^4\}
\end{array}
\right\}
$$

$$\times$$

$$\left\{\begin{array}{l} \left(\begin{array}{c} \gamma_1 \Omega_1 \{(\mathcal{T}^\ell_{\mathcal{B}_1}(\mathfrak{u}_1))^4 + (\mathcal{T}^\mathfrak{v}_{\mathcal{B}_1}(\mathfrak{u}_1))^4 + (\mathcal{J}^\ell_{\mathcal{B}_1}(\mathfrak{u}_1))^4 + (\mathcal{J}^\mathfrak{v}_{\mathcal{B}_1}(\mathfrak{u}_1))^4\}+ \\ \gamma_2 \Omega_1 \{(\mathcal{T}^\ell_{\mathcal{B}_2}(\mathfrak{u}_1))^4 + (\mathcal{T}^\mathfrak{v}_{\mathcal{B}_2}(\mathfrak{u}_1))^4 + (\mathcal{J}^\ell_{\mathcal{B}_2}(\mathfrak{u}_1))^4 + (\mathcal{J}^\mathfrak{v}_{\mathcal{B}_2}(\mathfrak{u}_1))^4\}+ \\ \qquad\qquad\qquad\qquad\vdots \\ + \\ \gamma_m \Omega_1 \{(\mathcal{T}^\ell_{\mathcal{B}_m}(\mathfrak{u}_1))^4 + (\mathcal{T}^\mathfrak{v}_{\mathcal{B}_m}(\mathfrak{u}_1))^4 + (\mathcal{J}^\ell_{\mathcal{B}_m}(\mathfrak{u}_1))^4 + (\mathcal{J}^\mathfrak{v}_{\mathcal{B}_m}(\mathfrak{u}_1))^4\} \end{array}\right)+ \\ \left(\begin{array}{c} \gamma_1 \Omega_2 \{(\mathcal{T}^\ell_{\mathcal{B}_1}(\mathfrak{u}_2))^4 + (\mathcal{T}^\mathfrak{v}_{\mathcal{B}_1}(\mathfrak{u}_2))^4 + (\mathcal{J}^\ell_{\mathcal{B}_1}(\mathfrak{u}_2))^4 + (\mathcal{J}^\mathfrak{v}_{\mathcal{B}_1}(\mathfrak{u}_2))^4\}+ \\ \gamma_2 \Omega_2 \{(\mathcal{T}^\ell_{\mathcal{B}_2}(\mathfrak{u}_2))^4 + (\mathcal{T}^\mathfrak{v}_{\mathcal{B}_2}(\mathfrak{u}_2))^4 + (\mathcal{J}^\ell_{\mathcal{B}_2}(\mathfrak{u}_2))^4 + (\mathcal{J}^\mathfrak{v}_{\mathcal{B}_2}(\mathfrak{u}_2))^4\}+ \\ \qquad\qquad\qquad\qquad\vdots \\ + \\ \gamma_m \Omega_2 \{(\mathcal{T}^\ell_{\mathcal{B}_m}(\mathfrak{u}_2))^4 + (\mathcal{T}^\mathfrak{v}_{\mathcal{B}_m}(\mathfrak{u}_2))^4 + (\mathcal{J}^\ell_{\mathcal{B}_m}(\mathfrak{u}_2))^4 + (\mathcal{J}^\mathfrak{v}_{\mathcal{B}_m}(\mathfrak{u}_2))^4\} \end{array}\right)+ \\ \qquad\qquad\qquad\qquad\vdots \\ + \\ \left(\begin{array}{c} \gamma_1 \Omega_n \{(\mathcal{T}^\ell_{\mathcal{B}_1}(\mathfrak{u}_n))^4 + (\mathcal{T}^\mathfrak{v}_{\mathcal{B}_1}(\mathfrak{u}_n))^4 + (\mathcal{J}^\ell_{\mathcal{B}_1}(\mathfrak{u}_n))^4 + (\mathcal{J}^\mathfrak{v}_{\mathcal{B}_1}(\mathfrak{u}_n))^4\}+ \\ \gamma_2 \Omega_n \{(\mathcal{T}^\ell_{\mathcal{B}_2}(\mathfrak{u}_n))^4 + (\mathcal{T}^\mathfrak{v}_{\mathcal{B}_2}(\mathfrak{u}_n))^4 + (\mathcal{J}^\ell_{\mathcal{B}_2}(\mathfrak{u}_n))^4 + (\mathcal{J}^\mathfrak{v}_{\mathcal{B}_2}(\mathfrak{u}_n))^4\}+ \\ \qquad\qquad\qquad\qquad\vdots \\ + \\ \gamma_m \Omega_n \{(\mathcal{T}^\ell_{\mathcal{B}_m}(\mathfrak{u}_n))^4 + (\mathcal{T}^\mathfrak{v}_{\mathcal{B}_m}(\mathfrak{u}_n))^4 + (\mathcal{J}^\ell_{\mathcal{B}_m}(\mathfrak{u}_n))^4 + (\mathcal{J}^\mathfrak{v}_{\mathcal{B}_m}(\mathfrak{u}_n))^4\} \end{array}\right) \end{array}\right\}$$

$$\mathcal{C}_{WIVPFSS}((\mathcal{F},\mathcal{A}),(\mathcal{G},\mathcal{B}))^2$$

$$\leq \sum_{j=1}^m \gamma_m \left(\sum_{i=1}^n \Omega_i \{((\mathcal{T}^\ell_{\mathcal{A}_j}(\mathfrak{u}_i))^4 + (\mathcal{T}^\mathfrak{v}_{\mathcal{A}_j}(\mathfrak{u}_i))^4) + ((\mathcal{J}^\ell_{\mathcal{A}_j}(\mathfrak{u}_i))^4 + (\mathcal{J}^\mathfrak{v}_{\mathcal{A}_j}(\mathfrak{u}_i))^4)\}\right)$$

$$\times \sum_{j=1}^m \gamma_m \left(\sum_{i=1}^n \Omega_i \{((\mathcal{T}^\ell_{\mathcal{B}_j}(\mathfrak{u}_i))^4 + (\mathcal{T}^\mathfrak{v}_{\mathcal{B}_j}(\mathfrak{u}_i))^4) + ((\mathcal{J}^\ell_{\mathcal{B}_j}(\mathfrak{u}_i))^4 + (\mathcal{J}^\mathfrak{v}_{\mathcal{B}_j}(\mathfrak{u}_i))^4)\}\right)$$

$\mathcal{C}_{WIVPFSS}((\mathcal{F},\mathcal{A}),(\mathcal{G},\mathcal{B}))^2 \leq \mathcal{E}_{WIVPFSS}(\mathcal{F},\mathcal{A}) \times \mathcal{E}_{WIVPFSS}(\mathcal{G},\mathcal{B})$.
Using Definition 4.3, we get
$\mathbb{C}_{WIVPFSS}((\mathcal{F},\mathcal{A}),(\mathcal{G},\mathcal{B})) \leq 1$.
So, it is verified that $0 \leq \mathbb{C}_{WIVPFSS}((\mathcal{F},\mathcal{A}),(\mathcal{G},\mathcal{B})) \leq 1$.
**Proof 2.** The proof is simple and easy to follow.

**Proof 3.** It is well-known that

$$\mathbb{C}_{WIVPFSS}((\mathcal{F}, \mathcal{A}), (\mathcal{G}, \mathcal{B}))$$

$$= \frac{\sum_{j=1}^{m} \gamma_j (\sum_{i=1}^{n} \Omega_i (\begin{smallmatrix} (\mathcal{T}_{\mathcal{A}_j}^{\ell}(\mathfrak{u}_i))^2 * (\mathcal{T}_{\mathcal{B}_j}^{\ell}(\mathfrak{u}_i))^2 + (\mathcal{T}_{\mathcal{A}_j}^{\mho}(\mathfrak{u}_i))^2 * (\mathcal{T}_{\mathcal{B}_j}^{\mho}(\mathfrak{u}_i))^2 + \\ (\mathcal{J}_{\mathcal{A}_j}^{\ell}(\mathfrak{u}_i))^2 * (\mathcal{J}_{\mathcal{B}_j}^{\ell}(\mathfrak{u}_i))^2 + (\mathcal{J}_{\mathcal{A}_j}^{\mho}(\mathfrak{u}_i))^2 * (\mathcal{J}_{\mathcal{B}_j}^{\mho}(\mathfrak{u}_i))^2 \end{smallmatrix}))}{\left( \frac{\sqrt{\sum_{j=1}^{m} \gamma_j (\sum_{i=1}^{n} \Omega_i ((\mathcal{T}_{\mathcal{A}_j}^{\ell}(\mathfrak{u}_i))^4 + (\mathcal{T}_{\mathcal{A}_j}^{\mho}(\mathfrak{u}_i))^4 + (\mathcal{J}_{\mathcal{A}_j}^{\ell}(\mathfrak{u}_i))^4 + (\mathcal{J}_{\mathcal{A}_j}^{\mho}(\mathfrak{u}_i))^4))}}{\sqrt{\sum_{j=1}^{m} \gamma_j (\sum_{i=1}^{n} \Omega_i ((\mathcal{T}_{\mathcal{B}_j}^{\ell}(\mathfrak{u}_i))^4 + (\mathcal{T}_{\mathcal{B}_j}^{\mho}(\mathfrak{u}_i))^4 + (\mathcal{J}_{\mathcal{B}_j}^{\ell}(\mathfrak{u}_i))^4 + (\mathcal{J}_{\mathcal{B}_j}^{\mho}(\mathfrak{u}_i))^4))}} \right)}$$

As
$$\mathcal{T}_{\mathcal{A}_j}^{\ell}(\mathfrak{u}_i) = \mathcal{T}_{\mathcal{B}_j}^{\ell}(\mathfrak{u}_i), \mathcal{T}_{\mathcal{A}_j}^{\mho}(\mathfrak{u}_i) = \mathcal{T}_{\mathcal{B}_j}^{\mho}(\mathfrak{u}_i), \mathcal{J}_{\mathcal{A}_j}^{\ell}(\mathfrak{u}_i) = \mathcal{J}_{\mathcal{B}_j}^{\ell}(\mathfrak{u}_i), \text{ and } \mathcal{J}_{\mathcal{A}_j}^{\mho}(\mathfrak{u}_i) = \mathcal{J}_{\mathcal{B}_j}^{\mho}(\mathfrak{u}_i). \text{ So,}$$

$$\mathbb{C}_{WIVPFSS}((\mathcal{F}, \mathcal{A}), (\mathcal{G}, \mathcal{B}))$$

$$= \frac{\sum_{j=1}^{m} \gamma_m (\sum_{i=1}^{n} \Omega_i \{((\mathcal{T}_{\mathcal{A}_j}^{\ell}(\mathfrak{u}_i))^4 + (\mathcal{T}_{\mathcal{A}_j}^{\mho}(\mathfrak{u}_i))^4) + ((\mathcal{J}_{\mathcal{A}_j}^{\ell}(\mathfrak{u}_i))^4 + (\mathcal{J}_{\mathcal{A}_j}^{\mho}(\mathfrak{u}_i))^4)\})}{\left( \frac{\sqrt{\sum_{j=1}^{m} \gamma_m (\sum_{i=1}^{n} \Omega_i \{((\mathcal{T}_{\mathcal{A}_j}^{\ell}(\mathfrak{u}_i))^4 + (\mathcal{T}_{\mathcal{A}_j}^{\mho}(\mathfrak{u}_i))^4) + ((\mathcal{J}_{\mathcal{A}_j}^{\ell}(\mathfrak{u}_i))^4 + (\mathcal{J}_{\mathcal{A}_j}^{\mho}(\mathfrak{u}_i))^4)\})}}{\sqrt{\sum_{j=1}^{m} \gamma_m (\sum_{i=1}^{n} \Omega_i \{((\mathcal{T}_{\mathcal{A}_j}^{\ell}(\mathfrak{u}_i))^4 + (\mathcal{T}_{\mathcal{A}_j}^{\mho}(\mathfrak{u}_i))^4) + ((\mathcal{J}_{\mathcal{A}_j}^{\ell}(\mathfrak{u}_i))^4 + (\mathcal{J}_{\mathcal{A}_j}^{\mho}(\mathfrak{u}_i))^4)\})}} \right)}$$

$$\mathbb{C}_{WIVPFSS}((\mathcal{F}, \mathcal{A}), (\mathcal{G}, \mathcal{B})) = 1.$$

## 5. TOPSIS approach on IVPFSS for MADM problem based on the correlation coefficient

This section proposes a methodology that uses the TOPSIS method to solve DM problems with IVPFSS statistics using CC. Hwang and Yoon [2] initially introduced the TOPSIS technique to evaluate positive and negative ideal solutions to DM problems. Using the TOPSIS methodology, the best alternatives can be identified by determining their distance from the PIS and NIS. This approach confirms that the correlation measure differentiates positive ideals from negative ideals by selecting positions. While academics have used the TOPSIS methodology to bargain closeness coefficients using distance and similarity measures, the TOPSIS technique with CC is preferred for finding closeness coefficients since the correlation measure directly reflects the relationship among the constrained factors. Using the settled CC, a TOPSIS technique can be used to select the most appropriate alternative.

### 5.1. Proposed TOPSIS approach

Let $\mathfrak{L} = \{\mathfrak{L}_1, \mathfrak{L}_2, \mathfrak{L}_3, \ldots, \mathfrak{L}_s\}$ denote a set of alternatives and $\mathfrak{H} = \{\mathfrak{H}^1, \mathfrak{H}^2, \mathfrak{H}^3, \ldots, \mathfrak{H}^n\}$ represents a group of professionals whose weight vectors can be stated as $\Omega = \{\Omega_1, \Omega_1, \ldots, \Omega_n\}^T$ and $\Omega_i > 0, \sum_{i=1}^{n} \Omega_i = 1$. Let $\varsigma = \{_{1, 2, 3, \ldots, m}\}$ be a collection of considered attributes whose weight vector is given as $\gamma = \{\gamma_1, \gamma_2, \gamma_3, \ldots, \gamma_m\}^T$ such as $\gamma_j > 0, \sum_{j=1}^{m} \gamma_j = 1$. The group of experts $\{\mathfrak{H}^i : i = 1, 2, \ldots, n\}$ deliver their opinions in the form of IVPFSNs for each alternative $\{\mathfrak{L}_z : z = 1, 2, 3, \cdots, s\}$ seeing the designated set of attributes $\varsigma = \{_{1, 2, 3, \ldots, m}\}$. The expert's opinion for each alternative in the form of IVPFSNs can be denoted as $\Delta_{ij}^{(z)} = (\mathcal{T}_{ij}^{(z)}, \mathcal{J}_{ij}^{(z)})$, where $\mathcal{T}_{ij}^{(z)} = [\mathcal{T}_{ij}^{\ell}, \mathcal{T}_{ij}^{\mho}], \mathcal{J}_{ij}^{(z)} = [\mathcal{J}_{ij}^{\ell}, \mathcal{J}_{ij}^{\mho}]$, and $0 \leq \mathcal{T}_{ij}^{\ell}, \mathcal{T}_{ij}^{\mho}, \mathcal{J}_{ij}^{\ell}, \mathcal{J}_{ij}^{\mho} \leq 1$ and $(\mathcal{T}_{ij}^{\mho})^2 + (\mathcal{J}_{ij}^{\mho})^2 \leq 1, \forall i, j$.

**Step 1.** Development of decision matrices for alternatives $\{\mathfrak{L}_z : z = 1, 2, \ldots, s\}$ in the form of IVPFSNs under-considered attributes given as:

$$(\mathfrak{L}_z, \varsigma)_{n\times m} = \begin{array}{c} \mathfrak{H}^1 \\ \mathfrak{H}^2 \\ \vdots \\ \mathfrak{H}^n \end{array} \begin{pmatrix} ([\mathcal{T}_{11}^\ell, \mathcal{T}_{11}^\mho], [\mathcal{J}_{11}^\ell, \mathcal{J}_{11}^\mho]) & ([\mathcal{T}_{12}^\ell, \mathcal{T}_{12}^\mho], [\mathcal{J}_{12}^\ell, \mathcal{J}_{12}^\mho]) & \cdots & ([\mathcal{T}_{1m}^\ell, \mathcal{T}_{1m}^\mho], [\mathcal{J}_{1m}^\ell, \mathcal{J}_{1m}^\mho]) \\ ([\mathcal{T}_{21}^\ell, \mathcal{T}_{21}^\mho], [\mathcal{J}_{21}^\ell, \mathcal{J}_{21}^\mho]) & ([\mathcal{T}_{22}^\ell, \mathcal{T}_{22}^\mho], [\mathcal{J}_{22}^\ell, \mathcal{J}_{22}^\mho]) & \cdots & ([\mathcal{T}_{2m}^\ell, \mathcal{T}_{2m}^\mho], [\mathcal{J}_{2m}^\ell, \mathcal{J}_{2m}^\mho]) \\ \vdots & \vdots & \vdots & \vdots \\ ([\mathcal{T}_{n1}^\ell, \mathcal{T}_{n1}^\mho], [\mathcal{J}_{n1}^\ell, \mathcal{J}_{n1}^\mho]) & ([\mathcal{T}_{n2}^\ell, \mathcal{T}_{n2}^\mho], [\mathcal{J}_{n2}^\ell, \mathcal{J}_{n2}^\mho]) & \cdots & ([\mathcal{T}_{nm}^\ell, \mathcal{T}_{nm}^\mho], [\mathcal{J}_{nm}^\ell, \mathcal{J}_{nm}^\mho]) \end{pmatrix}$$

**Step 2.** Obtain the standard interval-valued Pythagorean fuzzy soft decision matrix. Now, the consequent matrix $(\mathfrak{H}^{(z)}, \varsigma)_{n\times m}$ is judged by approving two varieties of parameters, such as benefit and cost type attributes. No need to normalize if attributes are of the same type. But if both parameters are involved, obtain the normalized decision matrices using the normalization rule to convert them into the same type.

$$R_{ij}^{(z)} = \begin{cases} \Delta_{ij}^c = ([\mathcal{J}_{ij}^\ell, \mathcal{J}_{ij}^\mho], [\mathcal{T}_{ij}^\ell, \mathcal{T}_{ij}^\mho]); & \text{cost type parameter} \\ \Delta_{ij} = ([\mathcal{T}_{ij}^\ell, \mathcal{T}_{ij}^\mho], [\mathcal{J}_{ij}^\ell, \mathcal{J}_{ij}^\mho]); & \text{benefit type parameter} \end{cases} \tag{5.1}$$

**Step 3.** Develop the weighted decision matrix for each alternative $\bar{\mathfrak{L}}^{(z)} = (\bar{\Delta}_{ij}^{(z)})_{n\times m}$, where

$$\bar{\mathfrak{L}}^{(z)} = \gamma_j \Omega_i \Delta_{ij}^{(z)} = (\sqrt{1 - ((1 - [\mathcal{T}_{ij}^\ell, \mathcal{T}_{ij}^\mho]^2)^{\Omega_i})^{\gamma_j}}, (([\mathcal{J}_{ij}^\ell, \mathcal{J}_{ij}^\mho])^{\Omega_i})^{\gamma_j})$$

$$= ([\bar{\mathcal{T}}_{ij}^\ell, \bar{\mathcal{T}}_{ij}^\mho], [\bar{\mathcal{J}}_{ij}^\ell, \bar{\mathcal{J}}_{ij}^\mho]) \tag{5.2}$$

Where $\Omega_i$ and $\gamma_j$ be the weights of experts and parameters.

**Step 4.** Calculate the indices $\hbar_{ij} = \arg max_z \{\theta_{ij}^{(z)}\}$ and $\mathfrak{g}_{ij} = \arg min_z \{\theta_{ij}^{(z)}\}$ and regulate the PIA and NIA such as:

$$\Delta^+ = ([\bar{\mathcal{T}}_{ij}^+, \bar{\mathcal{T}}_{ij}^+], [\bar{\mathcal{J}}_{ij}^+, \bar{\mathcal{J}}_{ij}^+])_{n\times m} = ([\bar{\mathcal{T}}_{ij}^\ell, \bar{\mathcal{T}}_{ij}^\mho]^{(\hbar_{ij})}, [\bar{\mathcal{J}}_{ij}^\ell, \bar{\mathcal{J}}_{ij}^\mho]^{(\hbar_{ij})}) \tag{5.3}$$

and

$$\Delta^- = ([\bar{\mathcal{T}}_{ij}^-, \bar{\mathcal{T}}_{ij}^-], [\bar{\mathcal{J}}_{ij}^-, \bar{\mathcal{J}}_{ij}^-])_{n\times m} = ([\bar{\mathcal{T}}_{ij}^\ell, \bar{\mathcal{T}}_{ij}^\mho]^{(\mathfrak{g}_{ij})}, [\bar{\mathcal{J}}_{ij}^\ell, \bar{\mathcal{J}}_{ij}^\mho]^{(\mathfrak{g}_{ij})}) \tag{5.4}$$

**Step 5.** Determine the CC among $\bar{\mathfrak{L}}^{(z)}$ and PIA $\Delta^+$ such as:

$$\kappa^{(z)} = \mathbb{C}_{IVPFSS}(\bar{\mathfrak{L}}^{(z)}, \Delta^+) = \frac{\mathcal{C}_{IVPFSS}(\bar{\mathfrak{L}}^{(z)}, \Delta^+)}{\sqrt{\mathcal{E}_{IVPFSS}\bar{\mathfrak{L}}^{(z)}}\sqrt{\mathcal{E}_{IVPFSS}\Delta^+}}$$

$$= \frac{\sum_{j=1}^m \sum_{i=1}^n (\bar{\mathcal{T}}_{ij}^{(z)} * \mathcal{T}_{ij}^+ + \bar{\mathcal{J}}_{ij}^{(z)} * \mathcal{J}_{ij}^+)}{\sqrt{\sum_{j=1}^m \sum_{i=1}^n ((\bar{\mathcal{T}}_{ij}^{(z)})^2 + (\bar{\mathcal{J}}_{ij}^{(z)})^2)}\sqrt{\sum_{j=1}^m \sum_{i=1}^n ((\mathcal{T}_{ij}^+)^2 + (\mathcal{J}_{ij}^+)^2)}} \tag{5.5}$$

Step 6. Determine the CC among $\bar{\mathfrak{L}}^{(z)}$ and PIA $\Delta^-$ such as:

$$\tau^{(z)} = \mathbb{C}_{IVPFSS}\big(\bar{\mathfrak{L}}^{(z)}, \Delta^-\big) = \frac{\mathcal{C}_{IVPFSS}(\bar{\mathfrak{L}}^{(z)}, \Delta^-)}{\sqrt{\mathcal{E}_{IVPFSS}\mathfrak{L}^{(z)}}\sqrt{\mathcal{E}_{IVPFSS}\Delta^-}}$$

$$= \frac{\sum\limits_{j=1}^{m}\sum\limits_{i=1}^{n}(\bar{\mathcal{T}}_{ij}^{(z)}*\mathcal{T}_{ij}^- + \bar{\mathcal{J}}_{ij}^{(z)}*\mathcal{J}_{ij}^-)}{\sqrt{\sum\limits_{j=1}^{m}\sum\limits_{i=1}^{n}((\bar{\mathcal{T}}_{ij}^{(z)})^2 + (\bar{\mathcal{J}}_{ij}^{(z)})^2)}\sqrt{\sum\limits_{j=1}^{m}\sum\limits_{i=1}^{n}((\mathcal{T}_{ij}^-)^2 + (\mathcal{J}_{ij}^-)^2)}} \tag{5.6}$$

Step 7. Calculate the closeness coefficient:

$$\beth^{(z)} = \frac{\daleth(\bar{\mathfrak{L}}^{(z)}, \Delta^-)}{\daleth(\bar{\mathfrak{L}}^{(z)}, \Delta^+) + \daleth(\bar{\mathfrak{L}}^{(z)}, \Delta^-)} \tag{5.7}$$

Where $\daleth(\bar{\mathfrak{L}}^{(z)}, \Delta^-) = 1 - \kappa^{(z)}$ and $\daleth(\bar{\mathfrak{L}}^{(z)}, \Delta^+) = 1 - \tau^{(z)}$.

Step 8. The maximum value of the closeness coefficient represents the most appropriate alternative.

Step 9. Examine the classification of the substitutes.

The flowchart of the planned TOPSIS algorithm can be presented as follows (See Fig 1).

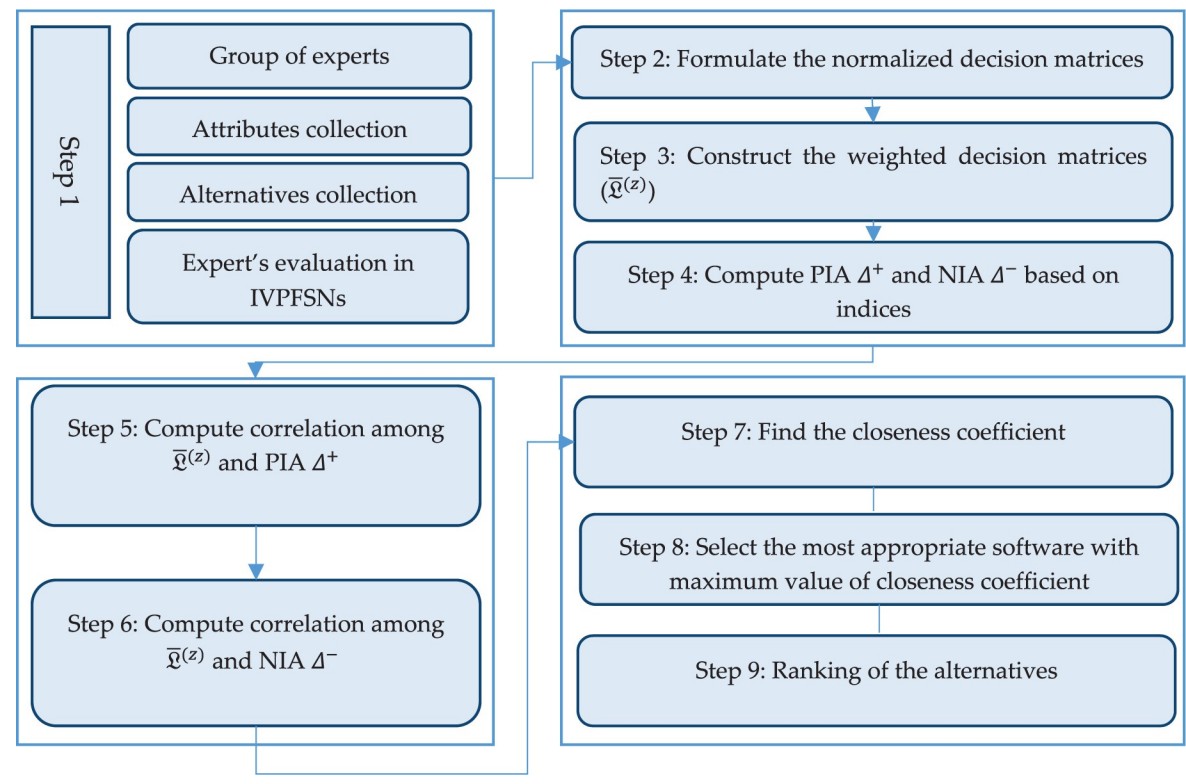

**Fig 1. Flow chart of the proposed TOPSIS model.**

## 6. Application of proposed technique for selection of ETL software in business intelligence

In this section, we showcase the practical application of the proposed TOPSIS method in decision-making by performing numerical calculations.

### 6.1 Selection of ETL software for BI

The problem is selecting an appropriate ETL software for business intelligence projects to maximize benefits. ETL is a vital process in data management. It involves collecting data from various structured and unstructured sources and converting it into a format that meets the operational and intentional requirements of the association. Once transformed, the data is delivered to a desired location. A reliable ETL system ensures that data is retrieved while adhering to data quality guidelines, transformed into a consistent format, and delivered in a way that software developers and business users can use to mark informed decisions. ETL can also perform advanced analyses to improve backend operations or enhance end-user experiences. Additionally, it prepares and cleans data using strategic rules to meet specific business intelligence needs, such as monthly reporting. Business intelligence (BI) is an integral part of data management architecture, which includes data collection, transformation, and restoration methods to support enterprise decision-makers. The core of BI is data warehouses, which are supported by ETL software. Therefore, selecting the best ETL software is crucial for the success of a BI project. The selected ETL software should maximize the benefits of the business and be compatible with its goals to decrease project risks and expenses. However, the decision-making process can be challenging with the abundance of ETL software products in the market. Different approaches have been used to address the selection problem, greatly expanding state-of-the-art. For example, Amiri [55] used the analytic hierarchy process (AHP) approach to determine the order of importance while choosing an ERP system, while Yigit et al. [56] used AHP to create an interactive model that aided in choosing software for Web-based learning objects. Furthermore, Göztepe [57] used the Analytic Network Process (ANP) method to evaluate and choose the optimal operating system considering organizational aspects and strategic performance metrics. Behzadian et al. [58] presented a TOPSIS-based model for multi-criteria decision-making. These methods have helped in the selection of the best ETL software and have improved the success rate of BI projects. To select the most appropriate software, six quality characteristics/attributes are explained as follows:

**Comprehensive functionality and connectivity ($\varsigma_1$).** Comprehensive functionality and connectivity are essential attributes of any effective ETL software. ETL is a data management process that collects data from different sources, transforms it into a format suitable for analysis, and loads it into a destination system. An ETL software must be comprehensive, meaning it should have many features and capabilities to handle diverse data formats, types, and sources. It should be able to handle both structured and unstructured data and support different data integration approaches, including batch processing, real-time streaming, and change data capture. Also, to have comprehensive functionality, ETL software should have strong connectivity. It should be able to connect to multiple data sources, including databases, flat files, cloud services, APIs, and web services. The software should also be able to connect to different types of data warehouses, such as on-premise, cloud-based, and hybrid solutions. This connectivity ensures that data can be easily accessed, integrated, and transformed, regardless of origin or destination.

**Error management ($\varsigma_2$).** Error management is an essential component of any ETL process. Since ETL involves extracting, transforming, and loading data from multiple sources, errors can occur at any process stage. These errors can result in data loss, inconsistencies, or corruption, leading to inaccurate or incomplete results. Effective error management in ETL

involves identifying, handling, and resolving errors promptly and efficiently. This requires a systematic approach, including error detection, handling, and resolution. Error detection involves identifying errors as they occur during the ETL process. This can be done through various techniques such as data profiling, quality checks, and validation. Once an error is detected, it must be handled appropriately to prevent further damage to the data. Error handling involves taking corrective action to mitigate the effects of errors. This may include restarting the ETL process, restoring backup data, or applying data correction techniques. Sometimes, it may be necessary to involve data experts or developers to resolve more complex errors. Error resolution involves identifying the root cause of errors and implementing measures to prevent them from recurring. This may include improving data quality checks, updating ETL processes, or refining data management policies and procedures. Effective error management ensures data accuracy, reliability, and integrity in ETL processes. It minimizes the risk of data loss, inconsistencies, and corruption and ensures that the results of the ETL process are trustworthy and useful for decision-making.

**Security ($\varsigma_3$).** Security is critical to any data management system, and ETL is no exception. ETL processes typically involve accessing, extracting, and loading data from various sources, and this data can often be sensitive and confidential. Therefore, ensuring the security of ETL processes is crucial to prevent unauthorized access, data breaches, and other security-related incidents. There are several ways to ensure security in ETL processes, such as:

❖ Access Control: Implementing strict access control mechanisms such as user authentication, authorization, and role-based access control can prevent unauthorized access to the ETL system.

❖ Encryption: Encrypting data at rest and in transit can prevent data theft and unauthorized access. This includes encrypting data in databases, files, and during transmission between systems.

❖ Auditing and Logging: ETL systems should have auditing and logging mechanisms to track user activities and detect security breaches. This can help identify security threats, analyze security incidents, and take appropriate action to prevent them.

❖ Data Masking: Sensitive data such as personal information, financial data, and health records can be masked or anonymized to protect them from unauthorized access.

❖ Secure Network Architecture: Implementing secure network architecture such as firewalls, virtual private networks (VPNs), and intrusion detection systems (IDS) can provide an additional layer of security for ETL processes.

Generally, ensuring security in ETL processes is critical to maintaining data confidentiality, integrity, and availability. Therefore, organizations should implement a comprehensive security strategy that includes access control, encryption, auditing, data masking, and secure network architecture to protect their ETL systems from security threats.

**Simple user interface ($\varsigma_4$).** A simple user interface (UI) is crucial to any ETL software. It provides an easy-to-use platform for users to interact with the ETL system and perform various tasks. The UI should be intuitive and user-friendly, allowing users to navigate the system and execute tasks easily. A good UI in ETL should have the following characteristics:

❖ Intuitive Navigation: The UI should provide clear navigation options allowing users to move between different system areas quickly.

❖ Clear Feedback: The system should provide clear feedback to users, letting them know that their actions have been executed successfully or if there were any errors.

❖ Error Handling: The UI should have proper error-handling mechanisms in place to help users understand and resolve any errors that occur.

❖ Customization: Users should be able to customize the UI to fit their specific needs and preferences.

❖ Accessibility: The UI should be accessible to all users, including those with disabilities, by adhering to accessibility guidelines.

❖ Training and Documentation: The ETL system should provide adequate training and documentation to help users understand how to use the system effectively.

A simple and user-friendly UI is essential for an ETL system's success. It can increase productivity, reduce errors, and help users manage and manipulate data efficiently. Moreover, a well-designed user interface should provide users with appropriate feedback and error messages, allowing them to quickly identify and correct any issues that arise during the ETL process.

**Durability ($\varsigma_5$).** Durability is an important aspect of ETL, referring to the system's ability to maintain data integrity and consistency over time, even in the face of system failures or disruptions. Durability is achieved through various mechanisms, including data backups, redundancy, and error detection and correction. Data backups involve making copies of data at regular intervals to ensure the data can be easily restored in case of failure or data loss. Redundancy involves duplicating critical components of the ETL system, such as servers or storage devices so that if one component fails, another can take its place without causing disruptions. Error detection and correction mechanisms are used to identify and fix errors or inconsistencies in the data as they occur. This may involve implementing automated checks and validations during the ETL process or using manual review processes to identify and correct errors.

**Accessibility to real-time data ($\varsigma_6$).** Accessibility to real-time data is a critical feature of ETL systems that can greatly enhance the efficiency and effectiveness of business operations. In the modern business environment, organizations require access to real-time data to make informed decisions and respond to changes quickly. ETL systems facilitate the extraction, transformation, and loading of real-time data from various sources, making it easily accessible for analysis and decision-making. To achieve accessibility to real-time data, ETL systems must be designed to handle high volumes of data with minimal delay. This requires the use of advanced data processing techniques, including parallel processing, distributed processing, and event-driven processing. ETL systems should also be able to integrate with real-time data sources, such as sensors and IoT devices, to ensure that data is continuously updated. Another key aspect of accessibility to real-time data is the ability to access data from multiple locations and devices. ETL systems must provide a user-friendly interface allowing users to access real-time data anywhere and anytime. This requires the use of cloud-based solutions that provide a high level of scalability, flexibility, and accessibility.

## 6.2 Numerical example

Suppose $\mathfrak{H} = \left\{ \mathfrak{H}^1, \mathfrak{H}^2, \mathfrak{H}^3, \mathfrak{H}^4 \right\}$ be a team of IT experts with wights $(0.1, 0.2, 0.4, 0.3)^T$ who are asked to select the best ETL software. The experts finalize a list of four softwares (alternatives) for deep exploration, such as $\mathfrak{L} = \left\{ \mathfrak{L}_1, \mathfrak{L}_2, \mathfrak{L}_3, \mathfrak{L}_4 \right\}$. Experts have provided a list of key factors or attributes that must be considered when evaluating ETL software for BI are discussed in section 6.1 and specified as: $\varsigma_1$ = Comprehensive Functionality and Connectivity, $\varsigma_2$ = Error Management, $\varsigma_3$ = Security, $\varsigma_4$ = Simple User Interface, $\varsigma_5$ = Durability and $\varsigma_6$ = Accessibility to Real-Time Data. The weights of the considered parameters for ETL selection in BI are given as follows: $(0.15, 0.1, 0.2, 0.1, 0.25, 0.2)^T$. The experts' opinions are presented as IVPFSNs for each software listed in Tables 1–4 and utilized the MADM methodology based on TOPSIS presented in section 5.1 to identify the most suitable software.

**Table 1. Expert's evaluation for $L_1$ in the form of IVPFSN.**

|  | $\varsigma_1$ | $\varsigma_2$ | $\varsigma_3$ | $\varsigma_4$ | $\varsigma_5$ | $\varsigma_6$ |
|---|---|---|---|---|---|---|
| $H^1$ | ([0.4, 0.5], [0.2, 0.5]) | ([0.7, 0.8], [0.5, 0.6]) | ([0.4, 0.6], [0.2, 0.5]) | ([0.2, 0.4], [0.2, 0.6]) | ([0.2, 0.7], [0.5, 0.6]) | ([0.4, 0.7], [0.5, 0.6]) |
| $H^2$ | ([0.2, 0.7], [0.2, 0.6]) | ([0.1, 0.6], [0.4, 0.5]) | ([0.2, 0.3], [0.4, 0.8]) | ([0.2, 0.5], [0.4, 0.7]) | ([0.4, 0.6], [0.2, 0.5]) | ([0.1, 0.4], [0.3, 0.7]) |
| $H^3$ | ([0.3, 0.5], [0.1, 0.4]) | ([0.4, 0.6], [0.2, 0.7]) | ([0.4, 0.7], [0.3, 0.7]) | ([0.5, 0.7], [0.2, 0.4]) | ([0.3, 0.5], [0.2, 0.8]) | ([0.3, 0.8], [0.2, 0.4]) |
| $H^4$ | ([0.4, 0.6], [0.3, 0.7]) | ([0.4, 0.5], [0.3, 0.7]) | ([0.3, 0.6], [0.3, 0.5]) | ([0.3, 0.6], [0.3, 0.5]) | ([0.3, 0.7], [0.4, 0.6]) | ([0.1, 0.7], [0.1, 0.6]) |

**Table 2. Expert's evaluation for $L_2$ in the form of IVPFSN.**

|  | $\varsigma_1$ | $\varsigma_2$ | $\varsigma_3$ | $\varsigma_4$ | $\varsigma_5$ | $\varsigma_6$ |
|---|---|---|---|---|---|---|
| $H^1$ | ([0.3, 0.6], [0.5, 0.6]) | ([0.2, 0.7], [0.5, 0.7]) | ([0.2, 0.7], [0.4, 0.5]) | ([0.6, 0.7], [0.5, 0.8]) | ([0.5, 0.6], [0.4, 0.8]) | ([0.2, 0.6], [0.5, 0.8]) |
| $H^2$ | ([0.3, 0.5], [0.5, 0.8]) | ([0.1, 0.4], [0.4, 0.5]) | ([0.1, 0.5], [0.3, 0.7]) | ([0.4, 0.5], [0.3, 0.6]) | ([0.2, 0.5], [0.3, 0.7]) | ([0.4, 0.5], [0.3, 0.6]) |
| $H^3$ | ([0.2, 0.6], [0.1, 0.4]) | ([0.1, 0.2], [0.2, 0.9]) | ([0.4, 0.7], [0.3, 0.7]) | ([0.5, 0.8], [0.2, 0.6]) | ([0.3, 0.6], [0.2, 0.5]) | ([0.5, 0.7], [0.2, 0.6]) |
| $H^4$ | ([0.2, 0.3], [0.3, 0.8]) | ([0.3, 0.5], [0.2, 0.8]) | ([0.3, 0.7], [0.2, 0.6]) | ([0.1, 0.7], [0.3, 0.6]) | ([0.1, 0.3], [0.3, 0.8]) | ([0.2, 0.7], [0.4, 0.6]) |

**Table 3. Expert's evaluation for $L_3$ in the form of IVPFSN.**

|  | $\varsigma_1$ | $\varsigma_2$ | $\varsigma_3$ | $\varsigma_4$ | $\varsigma_5$ | $\varsigma_6$ |
|---|---|---|---|---|---|---|
| $H^1$ | ([0.3, 0.4], [0.2, 0.7]) | ([0.3, 0.4], [0.4, 0.6]) | ([0.5, 0.6], [0.4, 0.5]) | ([0.3, 0.4], [0.3, 0.6]) | ([0.4, 0.6], [0.2, 0.5]) | ([0.6, 0.7], [0.5, 0.8]) |
| $H^2$ | ([0.4, 0.6], [0.3, 0.7]) | ([0.3, 0.5], [0.2, 0.3]) | ([0.3, 0.5], [0.5, 0.8]) | ([0.2, 0.6], [0.2, 0.4]) | ([0.3, 0.5], [0.1, 0.6]) | ([0.4, 0.5], [0.3, 0.6]) |
| $H^3$ | ([0.2, 0.4], [0.3, 0.4]) | ([0.3, 0.5], [0.3, 0.7]) | ([0.3, 0.7], [0.3, 0.6]) | ([0.1, 0.3], [0.5, 0.6]) | ([0.5, 0.7], [0.1, 0.6]) | ([0.5, 0.8], [0.2, 0.6]) |
| $H^4$ | ([0.3, 0.7], [0.3, 0.7]) | ([0.3, 0.5], [0.2, 0.4]) | ([0.2, 0.5], [0.3, 0.6]) | ([0.3, 0.4], [0.3, 0.7]) | ([0.2, 0.7], [0.3, 0.6]) | ([0.1, 0.7], [0.3, 0.6]) |

**Table 4. Expert's evaluation for $L_4$ in the form of IVPFSN.**

|  | $\varsigma_1$ | $\varsigma_2$ | $\varsigma_3$ | $\varsigma_4$ | $\varsigma_5$ | $\varsigma_6$ |
|---|---|---|---|---|---|---|
| $H^1$ | ([0.3, 0.5], [0.2, 0.6]) | ([0.2, 0.6], [0.4, 0.7]) | ([0.2, 0.5], [0.3, 0.6]) | ([0.4, 0.5], [0.6, 0.8]) | ([0.4, 0.7], [0.3, 0.6]) | ([0.4, 0.6], [0.2, 0.7]) |
| $H^2$ | ([0.2, 0.6], [0.3, 0.7]) | ([0.1, 0.5], [0.4, 0.7]) | ([0.5, 0.7], [0.4, 0.5]) | ([0.2, 0.5], [0.3, 0.4]) | ([0.1, 0.5], [0.2, 0.6]) | ([0.4, 0.5], [0.3, 0.6]) |
| $H^3$ | ([0.2, 0.5], [0.1, 0.6]) | ([0.2, 0.5], [0.1, 0.5]) | ([0.2, 0.4], [0.2, 0.7]) | ([0.3, 0.5], [0.1, 0.5]) | ([0.3, 0.6], [0.2, 0.6]) | ([0.5, 0.7], [0.2, 0.6]) |
| $H^4$ | ([0.2, 0.4], [0.5, 0.8]) | ([0.2, 0.5], [0.5, 0.8]) | ([0.2, 0.7], [0.3, 0.6]) | ([0.2, 0.5], [0.4, 0.5]) | ([0.2, 0.7], [0.4, 0.6]) | ([0.1, 0.5], [0.3, 0.6]) |

**Table 5. Weighted decision matrix for $\bar{L}_1$.**

|  | $\varsigma_1$ | $\varsigma_2$ | $\varsigma_3$ | $\varsigma_4$ | $\varsigma_5$ | $\varsigma_6$ |
|---|---|---|---|---|---|---|
| $H^1$ | ([.0567, .0745], [.4725, .5173]) | ([.0122, .0448], [.5129, .5343]) | ([.0251, .0642], [.4832, .5463]) | ([.0338, .0477], [.5134, .5351]) | ([.0227, .0443], [.5376, .5658]) | ([.0265, .0547], [.5723, .5173]) |
| $H^2$ | ([.0178, .0543], [.6950, .7965]) | ([.0238, .0499], [.5868, .6776]) | ([.0191, .0540], [.6746, .6372]) | ([.0572, .0676], [.5861, .5960]) | ([.0368, .0385], [.5373, .6852]) | ([.0341, .0525], [.6845, .7532]) |
| $H^3$ | ([.0467, .0696], [.5704, .6381]) | ([.0231, .0578], [.6249, .7129]) | ([.0142, .0549], [.5862, .6872]) | ([.0136, .0561], [.6352, .6532]) | ([.0136, .0561], [.6256, .7135]) | ([.0153, .0357], [.5709, .6385]) |
| $H^4$ | ([.0341, .0476], [.5236, .5752]) | ([.0461, .0565], [.5767, .5964]) | ([.0136, .0561], [.6322, .6432]) | ([.0356, .0545], [.4735, .5267]) | ([.0294, .0537], [.6149, .6578]) | ([.0142, .0549], [.5867, .6872]) |

**Table 6. Weighted decision matrix for $\bar{L}_2$.**

| | $\varsigma_1$ | $\varsigma_2$ | $\varsigma_3$ | $\varsigma_4$ | $\varsigma_5$ | $\varsigma_6$ |
|---|---|---|---|---|---|---|
| $H^1$ | ([.0254, .0451], [.4621, .5279]) | ([.0265, .0345], [.4945, .5567]) | ([.0247, .0379], [.5264, .5458]) | ([.0561, .0746], [.9728, .9863]) | ([.0227, .0283], [.5124, .5648]) | ([.0432, .0477], [.5534, .5748]) |
| $H^2$ | ([.0534, .0613], [.6345, .7637]) | ([.0334, .0367], [.6549, .6878]) | ([.0532, .0596], [.6563, .6860]) | ([.0175, .0534], [.6941, .7981]) | ([.0245, .0387], [.5060, .6372]) | ([.0472, .0476], [.5463, .5762]) |
| $H^3$ | ([.0365, .0377], [.5302, .5389]) | ([.0172, .0239], [.5287, .6162]) | ([.0276, .0461], [.6187, .6252]) | ([.0471, .0642], [.9139, .9382]) | ([.0448, .0587], [.6356, .6445]) | ([.0336, .0561], [.6322, .6431]) |
| $H^4$ | ([.0564, .0641], [.4931, .5068]) | ([.0227, .0453], [.5874, .5948]) | ([.0134, .0219], [.7142, .7235]) | ([.0375, .0379], [.6373, .6852]) | ([.0153, .0351], [.5809, .6085]) | ([.0236, .0381], [.6756, .7235]) |

**Table 7. Weighted decision matrix for $\bar{L}_3$.**

| | $\varsigma_1$ | $\varsigma_2$ | $\varsigma_3$ | $\varsigma_4$ | $\varsigma_5$ | $\varsigma_6$ |
|---|---|---|---|---|---|---|
| $H^1$ | ([.0207, .0568], [.6527, .9249]) | ([.0238, .0547], [.9143, .9265]) | ([.0272, .0291], [.9451, .9733]) | ([.0407, .0654], [.8524, .9247]) | ([.0578, .0772], [.8086, .9074]) | ([.0320, .0354], [.9643, .9836]) |
| $H^2$ | ([.0454, .0821], [.8612, .8975]) | ([.0262, .0348], [.9427, .9652]) | ([.0071, .0105], [.9783, .9962]) | ([.0535, .0923], [.8719, .9069]) | ([.0237, .0478], [.9379, .9526]) | ([.0063, .0107], [.9127, .9722]) |
| $H^3$ | ([.0263.0364], [.9380, .9584]) | ([.0139, .0179], [.9424, .9640]) | ([.0043, .0063], [.9652, .9932]) | ([.0159, .0227], [.9421, .9573]) | ([.0469.0613], [.9289, .9547]) | ([.0143, .0169], [.9414, .9529]) |
| $H^4$ | ([.0227, .0272], [.9713, .9728]) | ([.0264, .0509], [.9436, .9495]) | ([.0218, .0327], [.9549, .9683]) | ([.0067, .0143], [.9809, .9867]) | ([.0203, .0231], [.9661, .9761]) | ([.0057, .0097], [.9631, .9906]) |

**Table 8. Weighted decision matrix for $\bar{L}_4$.**

| | $\varsigma_1$ | $\varsigma_2$ | $\varsigma_3$ | $\varsigma_4$ | $\varsigma_5$ | $\varsigma_6$ |
|---|---|---|---|---|---|---|
| $H^1$ | ([.0057, .0097], [.9631, .9906]) | ([.0272, .0291], [.9451, .9733]) | ([.0218, .0327], [.9549, .9683]) | ([.0159, .0227], [.9421, .9573]) | ([.0227, .0272], [.9713, .9728]) | ([.0454, .0818], [.8612, .8975]) |
| $H^2$ | ([.0237, .0478], [.9379, .9526]) | ([.0238, .0547], [.9143, .9265]) | ([.0254, .0451], [.4621, .5279]) | ([.0334, .0367], [.6549, .6878]) | ([.0134, .0219], [.7142, .7235]) | ([.0175, .0534], [.6941, .7981]) |
| $H^3$ | ([.0448, .0587], [.6356, .6445]) | ([.0172, .0239], [.5287, .6162]) | ([.0472, .0476], [.5463, .5762]) | ([.0175, .0534], [.6941, .7981]) | ([.0142, .0549], [.5867, .6872]) | ([.0336, .0561], [.6322, .6431]) |
| $H^4$ | ([.0136, .0561], [.6256, .7135]) | ([.0231, .0578], [.6249, .7129]) | ([.0191, .0540], [.6146, .6372]) | ([.0375, .0379], [.6373, .6852]) | ([.0153, .0351], [.5809, .6085]) | ([.0136, .0561], [.6256, .7135]) |

Step 1. Development of decision matrices for alternatives $\mathfrak{L} = \{\mathfrak{L}_1, \mathfrak{L}_2, \mathfrak{L}_3, \mathfrak{L}_4\}$ in the form of IVPFSNs under-considered attributes given as

Step 2: The considered parameters are the same type, so normalization is unnecessary.

Step 3: Develop the weighted decision matrix for each alternative $\bar{\mathfrak{L}}_z$ using Eq 5.2 in the following Tables 5–8.

Step 4. Compute the PIA and NIA using Eqs 5.3 and 5.4, respectively.

$$
\Delta^{(z)}_{\varsigma_{ij}}+ = \begin{bmatrix} \begin{pmatrix} [.0567, .0745], \\ [.4725, .5173] \end{pmatrix} & \begin{pmatrix} [.0272, .0291], \\ [.9451, .9733] \end{pmatrix} & \begin{pmatrix} [.0272, .0291], \\ [.9451, .9733] \end{pmatrix} & \begin{pmatrix} [.0159, .0227], \\ [.9421, .9573] \end{pmatrix} & \begin{pmatrix} [.0227, .0272], \\ [.9713, .9728] \end{pmatrix} & \begin{pmatrix} [.0320, .0354], \\ [.9643, .9836] \end{pmatrix} \\[2em] \begin{pmatrix} [.0534, .0613], \\ [.6345, .7637] \end{pmatrix} & \begin{pmatrix} [.0334, .0367], \\ [.6549, .6878] \end{pmatrix} & \begin{pmatrix} [.0071, .0105], \\ [.9783, .9962] \end{pmatrix} & \begin{pmatrix} [.0334, .0367], \\ [.6549, .6878] \end{pmatrix} & \begin{pmatrix} [.0368, .0385], \\ [.5373, .6852] \end{pmatrix} & \begin{pmatrix} [.0472, .0476], \\ [.5463, .5762] \end{pmatrix} \\[2em] \begin{pmatrix} [.0365, .0377], \\ [.5302, .5389] \end{pmatrix} & \begin{pmatrix} [.0139, .0179], \\ [.9424, .9640] \end{pmatrix} & \begin{pmatrix} [.0472, .0476], \\ [.5463, .5762] \end{pmatrix} & \begin{pmatrix} [.0159, .0227], \\ [.9421, .9573] \end{pmatrix} & \begin{pmatrix} [.0448, .0587], \\ [.6356, .6445] \end{pmatrix} & \begin{pmatrix} [.0143, .0169], \\ [.9414, .9529] \end{pmatrix} \\[2em] \begin{pmatrix} [.0227, .0272], \\ [.9713, .9728] \end{pmatrix} & \begin{pmatrix} [.0461, .0565], \\ [.5767, .5964] \end{pmatrix} & \begin{pmatrix} [.0134, .0219], \\ [.7142, .7235] \end{pmatrix} & \begin{pmatrix} [.0375, .0379], \\ [.6373, .6852] \end{pmatrix} & \begin{pmatrix} [.0203, .0231], \\ [.9661, .9761] \end{pmatrix} & \begin{pmatrix} [.0057, .0097], \\ [.9631, .9906] \end{pmatrix} \end{bmatrix}
$$

$$
\Delta^{(z)}_{\varsigma_{ij}}- = \begin{bmatrix} \begin{pmatrix} [.0207, .0568], \\ [.6527, .9249] \end{pmatrix} & \begin{pmatrix} [.0122, .0448], \\ [.5129, .5343] \end{pmatrix} & \begin{pmatrix} [.0251, .0642], \\ [.4832, .5463] \end{pmatrix} & \begin{pmatrix} [.0407, .0654], \\ [.8524, .9247] \end{pmatrix} & \begin{pmatrix} [.0227, .0443], \\ [.5376, .5658] \end{pmatrix} & \begin{pmatrix} [.0454, .0818], \\ [.8612, .8975] \end{pmatrix} \\[2em] \begin{pmatrix} [.0534, .0613], \\ [.6345, .7637] \end{pmatrix} & \begin{pmatrix} [.0238, .0547], \\ [.9143, .9265] \end{pmatrix} & \begin{pmatrix} [.0191, .0540], \\ [.6746, .6372] \end{pmatrix} & \begin{pmatrix} [.0535, .0923], \\ [.8719, .9069] \end{pmatrix} & \begin{pmatrix} [.0237, .0478], \\ [.9379, .9526] \end{pmatrix} & \begin{pmatrix} [.0175, .0534], \\ [.6941, .7981] \end{pmatrix} \\[2em] \begin{pmatrix} [.0467, .0696], \\ [.5704, .6381] \end{pmatrix} & \begin{pmatrix} [.0231, .0578], \\ [.6249, .7129] \end{pmatrix} & \begin{pmatrix} [.0142, .0549], \\ [.5862, .6872] \end{pmatrix} & \begin{pmatrix} [.0136, .0561], \\ [.6352, .6532] \end{pmatrix} & \begin{pmatrix} [.0142, .0549], \\ [.5867, .6872] \end{pmatrix} & \begin{pmatrix} [.0153, .0357], \\ [.5709, .6385] \end{pmatrix} \\[2em] \begin{pmatrix} [.0136, .0561], \\ [.6256, .7135] \end{pmatrix} & \begin{pmatrix} [.0231, .0578], \\ [.6249, .7129] \end{pmatrix} & \begin{pmatrix} [.0191, .0540], \\ [.6146, .6372] \end{pmatrix} & \begin{pmatrix} [.0356, .0545], \\ [.4735, .5267] \end{pmatrix} & \begin{pmatrix} [.0294, .0537], \\ [.6149, .6578] \end{pmatrix} & \begin{pmatrix} [.0136, .0561], \\ [.6256, .7135] \end{pmatrix} \end{bmatrix}
$$

Step 5. Determine the CC among $\bar{\mathfrak{L}}^{(z)}$ and PIA $\Delta^+$ using Eq 5.5, such as:

$$\kappa^{(1)} = 0.99361, \kappa^{(2)} = 0.99628, \kappa^{(3)} = 0.99537, \kappa^{(4)} = 0.99620.$$

Step 6. Determine the CC among $\bar{\mathfrak{L}}^{(z)}$ and NIA $\Delta^-$ using Eq 5.6, such as:

$$\tau^{(1)} = 0.99638, \tau^{(2)} = 0.99651, \tau^{(3)} = 0.99302, \tau^{(4)} = 0.99579.$$

Step 7. Find the closeness coefficient using Eq 5.7.

$$\beth^{(1)} = 0.63836, \beth^{(2)} = 0.51595, \beth^{(3)} = 0.39879, \text{and} \beth^{(4)} = 0.47441.$$

Step 8. The above calculation shows that $\beth^{(1)} = 0.63836$ is the maximum value of the closeness coefficient. So, $\mathfrak{L}_1$ is the most suitable software.

Step 9: Ranking of the alternatives $\mathfrak{L}_1 > \mathfrak{L}_2 > \mathfrak{L}_4 > \mathfrak{L}_3$.

## 7. Discussion and comparative analysis

The following section evaluates the practicality of the proposed method by comparing it to existing approaches.

### 7.1 Superiority of the proposed methodology

A suggested approach presents a correlation-based TOPSIS technique to tackle MADM problems associated with IVPFSS and is very efficient and significant. Our technique can quickly resolve MADM complications and is much more precise than current approaches. It can accommodate multiple results based on different criteria, kinds of reliability, and

**Table 9. Qualitative comparison of the proposed model with the prevalent models.**

| | Set | Parameters | Expert's opinions in interval form | Advantages | Limitations |
|---|---|---|---|---|---|
| **Zadeh** [5] | FS | × | × | Deals uncertainty using MD | Unable to handle NMD |
| **Atanassov** [9] | IFS | × | × | Deals uncertainty using MD and NMD | Unable to handle $MD+NMD>1$ |
| **Yager** [21] | PFS | × | × | Deals uncertainty using MD and NMD | Unable to handle $MD^2+NMD^2>1$ |
| **Turksen** [6] | IVFS | × | ✓ | Deals uncertainty using MD intervals | Unable to handle NMD interval |
| **Atanassov** [10] | IVIFS | × | ✓ | Deals uncertainty using MD and NMD intervals | Unable to handle $MD^{\upsilon}+NMD^{\upsilon}>1$ |
| **Peng & Yang** [30] | IVPFS | × | ✓ | Deals uncertainty using MD and NMD intervals | Unable to handle $(MD^{\upsilon})^2+(NMD^{\upsilon})^2>1$ |
| **Maji et al.** [33] | FSS | ✓ | × | Deals uncertainty using MD | Unable to handle NMD |
| **Maji et al.** [34] | IFSS | ✓ | × | Deals uncertainty using MD and NMD | Unable to handle $MD+NMD>1$ |
| **Peng et al.** [43] | PFSS | ✓ | × | Deals uncertainty using MD and NMD | Unable to handle $MD^2+NMD^2>1$ |
| **Jiang et al.** [37] | IVIFSS | ✓ | ✓ | Deals uncertainty using MD and NMD intervals | Unable to handle $MD^{\upsilon}+NMD^{\upsilon}>1$ |
| **Proposed approach** | IVPFSS | ✓ | ✓ | Deals uncertainty using MD and NMD intervals | Unable to handle $(MD^{\upsilon})^2+(NMD^{\upsilon})^2>1$ |

modifications since it is malleable and adaptable. We alter the evaluation framework for several models with particular classification properties to reflect these viewpoints. Organizational research and assessments indicate that our approach's outcomes are equivalent to other kinds of alliances. Several FS, IFS, and PFS amalgam combinations may cause IVPFSS given particular circumstances. As demonstrated in Table 9, we can indicate object-related facts in a specific and interim way, which makes it a suitable tool for incorporating unclear and imprecise facts in DM agendas. Our assessment and analysis demonstrate that the outcomes of our suggested approach are more extensive compared to other techniques. However, because the DM process uses so much data, it needs greater attention to detail than our suggested DM procedures. Nevertheless, our intended approach is more efficient, adaptable, and superior to previous FS, IFS, and PFS amalgam conformations. Different IVFS combinations can also be changed into IVPFSS by including the proper relationships. How easily confusing and unusual data may be added to the existing system is impressive. We can properly and realistically designate information about well-being while considering fictitious and disturbing data frequently encountered during the DM process. As a result, our suggested method is superior, noteworthy, and extraordinary and can handle different FS combinations. Table 9 lists the features of both our suggested and current techniques.

It turns out that a new problem has arisen recently. This discussion aims to explain the rationale behind introducing a novel TOPSIS method designed to cater to the specific requirements of a particular organization. Although there are various existing methods, the proposed approach stands out due to its unique hybrid structure, which includes FS, IVFS, IFS, IVIFS, PFS, IVPFS, FSS, IVFSS, IFSS, IVIFSS, and PFSS. However, these hybrid structures have limitations in comprehensively analyzing the situation. To overcome these limitations, we have developed a correlation-based TOPSIS technique for IVPFSS that can handle attributes considering the MD and NMD in intervals such as $0\leq(MD^{\upsilon})^2+(NMD^{\upsilon})^2\leq1$. This new approach is unique in its ability to provide a more detailed analysis of the situation compared to existing hybrid structures. Moreover, our proposed hybrid structure of FS is superior to other existing hybrid systems of FS, as presented in Table 9. Choosing the appropriate TOPSIS method is critical for an organization's success, and our novel approach provides a more comprehensive analysis of the situation, which is essential for making informed decisions.

## 7.2 Comparative analysis

Previous research and comparison investigations have proven the validity of the established correlation-based TOPSIS technique, demonstrating that the findings produced using this method are consistent with those obtained using other current methodologies. When paired with other DM approaches, the suggested TOPSIS model can incorporate extra information on the parameters of the alternatives, hence resolving the imprecision in the data. This results in a more precise and empirical representation of object-related facts, making it a valuable tool for determining ambiguous and incomprehensible facts in the DM process. Furthermore, the comparison study shows that the suggested technique's decision-making process varies from earlier methods because PIA and NIA are based on CC inducement at a certain temporary level rather than distance and similarity measurements. This prevents information loss during the process, possibly if score values provided to individual parameters fail to consider the impact on other parameters. The optimal correlation measure for each parameter is derived by evaluating the most beneficial results and determining their correlation. By expressing the degree of perception and similarity across explanations, the planned TOPSIS method outperforms present methodologies and correlated measurements, avoiding conclusions based on negative foundations. Furthermore, present TOPSIS approaches analyze all choices before deciding on ETL software. Several TOPSIS techniques, including those extended by Ansari et al. [7], Ashtiani et al. [8], Rouyendegh et al. [11], Zhang and Yu [18], Rani et al. [59], and Garg [60], have problems in dealing with different parametrization and specific scenarios involving MD and NMD intervals. The TOPSIS technique devised by Garg and Arora [36] and Zulqarnain et al. [40] for IFSS and IVIFSS, respectively, is incapable of addressing scenarios where $MD+NMD>1$ and $MD^{\mho}+NMD^{\mho}>1$. Meanwhile, the TOPSIS technique described in [53] handles parametrized values. Still, it ignores interval information, while the aggregation operators presented in [54] can deal with some problems but fail to calculate the proximity coefficient in some cases. Our suggested TOPSIS model addresses these concerns skillfully and is compared to existing approaches. It provides comparable results and demonstrates its consistency and efficiency in picking the top four ETL software solutions (as shown in Table 10). The shorthand for not applicable is n/a.

The evaluation matrix used to assess multiple TOPSIS techniques and outcomes for each alternative are presented in Table 10. According to the table, the $\mathfrak{L}_1$ the alternative is the ideal ETL software for BI. Table 10 provides an extensive summary of the proposed approach and an investigation. Table 10 indicates that previous works, such as those in [7, 8, 11, 18, 59, 60],

**Table 10. Comparative analysis of the proposed model with existing models under the considered data set.**

| Structure | Alternatives score values or closeness coefficient | | | | Ranking |
|---|---|---|---|---|---|
| **Fuzzy TOPSIS** [7] | n/a | | | | n/a |
| **IVFS TOPSIS** [8] | n/a | | | | n/a |
| **IFS TOPSIS** [11] | n/a | | | | n/a |
| **IVIFS TOPSIS** [18] | n/a | | | | n/a |
| **PFS TOPSIS** [59] | n/a | | | | n/a |
| **IVPFS TOPSIS** [60] | n/a | | | | n/a |
| **IFSS TOPSIS** [36] | n/a | | | | n/a |
| **IVIFSS TOPSIS** [40] | n/a | | | | n/a |
| **PFSS TOPSIS** [53] | n/a | | | | n/a |
| **IVPFSWA** [54] | *Score*(0.69271) | *Score*(0.66247) | *Score*(0.65816) | *Score*(0.6882) | $\mathfrak{L}_1 > \mathfrak{L}_4 > \mathfrak{L}_2 > \mathfrak{L}_3$ |
| **IVPFSWG** [54] | *Score*(0.67352) | *Score*(0.65461) | *Score*(0.64152) | *Score*(0.66517) | $\mathfrak{L}_1 > \mathfrak{L}_4 > \mathfrak{L}_2 > \mathfrak{L}_3$ |
| **Proposed TOPSIS** | 0.63836 | 0.51595 | 0.39879 | 0.47441 | $\mathfrak{L}_1 > \mathfrak{L}_2 > \mathfrak{L}_4 > \mathfrak{L}_3$ |

lack details about parametric analysis. Meanwhile, the TOPSIS methods [36, 40, 53] can manage the parametrized values of the alternatives. However, they are unable to deal with the under-consideration data set. Also, the IVPFSWA [54] and IVPFSWG [54] dealt with the considered data set, but these AOs unable to handle the closeness coefficient. The suggested approach, on the other hand, has the benefit of successfully addressing real-life challenges by acknowledging parametric aspects of the alternatives. As a result, the newly developed method can cope with DM problems that current operators in the IVPFSS environment cannot solve.

### 7.3 Merits and theoretical advantages of the planned model

The proposed model offers several advantages that make it more reliable and effective, including:

❖ Experts often face challenges when selecting attributes and classifying them based on sub-attributes, as these choices are not always absolute. The proposed model addresses this issue by using the concept of interval-valued Pythagorean fuzzy parameterization, which captures the unpredictable behavior of attributes.

❖ The TOPSIS model developed in this study allows specialists to deliver evaluations based on 2D MD and NMD functions. This allows them to express agreement or disagreement with assessments in an interval form, improving accuracy and precision.

❖ The developed model can handle matching attribute values in predictions, providing clarity in uncertainty areas. This is achieved through 2D MD and NMD functions presented in interval form and by incorporating stage values to address duplications.

❖ The developed TOPSIS model enables experts to provide evaluations based on multiple parameters, which is more effective than relying on a single parameter. This feature is particularly advantageous in the case of IVPFSS.

❖ The study provides mathematical explanations for the correlation coefficients used in the proposed algorithm, which demonstrate the symmetry of the algorithm. This symmetry ensures fairness and reliability in decision-making, making the algorithm suitable for several solicitations where accuracy and stability are necessary.

❖ The study's focus on IVPFSS denotes a substantial conjectural encroachment, as it is the best comprehensive system of IVPFSS. The enlargement of informational energies for IVPFSS sceneries and the argument of their rudimentary possessions auxiliary spread the speculative structure for IVPFSS, contributing to more consistent and precise methodologies to DM in the perspective of IVPFSS, an important academic development in FS and DM.

## 8. Conclusion

The main focus of this study is to address the challenges of inadequate data, obscurity, and irregularity in IVPFSS. The substantial objective of this research is to propose a novel approach that utilizes MD and NMD values of the attributes under consideration. Additionally, the study introduces new correlation measures, such as CC and WCC, for IVPFSS and analyses their properties in detail. Furthermore, the study demonstrates that by considering only one parameter, various existing correlation measures in the context of a PFS can be viewed as special cases of the proposed measures. The TOPSIS technique is presented based on the established CC and WCC since the features and specialists decide on MADM contests. The study uses correlation indices and closeness coefficient to determine the PIA (NIA) and rank of alternatives. A numerical example demonstrates the proposed TOPSIS technique's

effectiveness in ETL software selection for BI. A comparative analysis is conducted to validate the approach's efficacy and fairness, demonstrating its exceptional stability and practical potential for decision-makers in the DM process. Future research can explore using VIKOR and MABAC methods to address DM issues and develop other AOs, such as Bonferroni Mean AOs, Einstein AOs, Einstein-ordered AOs, and Einstein hybrid AOs, and their applications in real-world complications. This approach has significant latent presentations in several fields, including engineering, management, medical sciences, and social sciences, where DM under ambiguity and inaccurate statistics are involved.

## Author Contributions

**Conceptualization:** Rana Muhammad Zulqarnain.

**Data curation:** Rana Muhammad Zulqarnain, Imran Siddique.

**Formal analysis:** Rana Muhammad Zulqarnain, Sameh Askar.

**Funding acquisition:** Sameh Askar.

**Investigation:** Imran Siddique, Muhammad Asif, Sameh Askar.

**Methodology:** Imran Siddique, Muhammad Asif.

**Project administration:** Sameh Askar.

**Resources:** Muhammad Asif, Hijaz Ahmad, Shahid Hussain Gurmani.

**Software:** Muhammad Asif.

**Supervision:** Hijaz Ahmad.

**Validation:** Muhammad Asif, Hijaz Ahmad, Shahid Hussain Gurmani.

**Visualization:** Hijaz Ahmad, Shahid Hussain Gurmani.

**Writing – original draft:** Rana Muhammad Zulqarnain, Imran Siddique.

**Writing – review & editing:** Shahid Hussain Gurmani.

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
