## [Decision Letter · Decision Letter 0]

9 May 2023

PONE-D-23-09868Extension of Correlation Coefficient Based TOPSIS Technique for Interval-Valued Pythagorean Fuzzy Soft Set: A Case Study in ETL TechniquesPLOS ONE

Dear Dr. Zulqarnain,

Thank you for submitting your manuscript to PLOS ONE. After careful consideration, we feel that it has merit but does not fully meet PLOS ONE’s publication criteria as it currently stands. Therefore, we invite you to submit a revised version of the manuscript that addresses the points raised during the review process.

We look forward to receiving your revised manuscript.

Kind regards,

Muhammet Gul, Ph.D.

Academic Editor

PLOS ONE

Journal Requirements:

Reviewers' comments:

Reviewer's Responses to Questions

**Comments to the Author**

1. Is the manuscript technically sound, and do the data support the conclusions?

Reviewer #1: Yes

Reviewer #2: Yes

2. Has the statistical analysis been performed appropriately and rigorously? 

Reviewer #1: N/A

Reviewer #2: Yes

3. Have the authors made all data underlying the findings in their manuscript fully available?

Reviewer #1: Yes

Reviewer #2: Yes

4. Is the manuscript presented in an intelligible fashion and written in standard English?

Reviewer #1: Yes

Reviewer #2: Yes

5. Review Comments to the Author

Reviewer #1: The authors perform an interesting concise analysis of a specific problem that concerns the popular TOPSIS technique for multi-attribute decision-making. The baseline model is Pythagorean fuzzy soft sets. The investigation is inspired by references [33] and [45]. The authors motivate their goals with a precise list of setbacks of the approaches presented by these two articles. In this context the problem is the measurement of “correlation” of the concept that arises when the observed membership and non-membership degrees sum up to more than one. As the authors explain, this element is not defined in the case of interval-valued Pythagorean fuzzy soft set (IVPFSS) data.

The article therefore improves the body of knowledge in an active area of research.

Below I give some comments for improvement.

(1) Define all elements properly. I recommend to avoid ETL in the title, or give it in its full form, as it is not common knowledge

(2) There are few recent references. There are 4 papers from 2021, and 3 of them are from authors of this article. And there are only 2 articles from 2022, both written by the first author of this submission. So the selection of literature seems to be too biased and not totally updated. To guarantee that the potential readers find the paper authoritative enough, the article needs to include more papers from reputed journals related to this topic in the summary of literature. Below are some possible references for your perusal (please consider these articles as mere suggestions):

Risk evaluation in failure modes and effects analysis: hybrid TOPSIS and ELECTRE I solutions with Pythagorean fuzzy information. Neural Computing and Applications 33 (2021), 5675-5703

An integrated ELECTRE-I approach for risk evaluation with hesitant Pythagorean fuzzy information. Expert Systems with Applications 200 (2022), 116945

TOPSIS Approach for MAGDM Based on Interval-Valued Hesitant Fuzzy N-Soft Environment. International Journal of Fuzzy Systems 21 (2019), 993-1009

PF-TOPSIS method based on CPFRS models: An application to unconventional emergency events. Computers & Industrial Engineering 139 (2020), 106192

Ranked soft sets. Expert Systems (2023), e13231

(3) Also with respect to the review of literature, note that Pythagorean fuzzy sets had been defined by Prof. Atanassov with the name “type-2 intuitionistic fuzzy sets”. They became popular with Prof. Yager’s paper, but they were not new.

(4) In continuation with existing literature, it is also possible to use dissimilarity measures instead of correlation coefficients or similarity measures (pages 2-3). The discussion in “Improved generalized dissimilarity measure‐based VIKOR method for Pythagorean fuzzy sets”, International Journal of Intelligent Systems 37 (2022), 1807-1845, is quite illustrative for this purpose.

(5) Table 9 is imperfect. The advantages of the “fuzzy soft” versions are identical to the “fuzzy” versions.

(6) Typos and minor suggestions.

Line 6 of Motivation (page 3): delete “upper”.

Def 2.7 uses IVPFSN but the concept has not been defined (this can be done easily in Def 2.6). Delete “and” in Def 2.7.

Figure 1, Step 1: why IT experts? the Figure should not illustrate the case study of Section 6.2 only.

In Table 9, the limitation described for “the proposed approach” seems to be incorrect (compare with its positive feature in line 7 under this table).

Reviewer #2: Title: Extension of Correlation Coefficient Based TOPSIS Technique for Interval-Valued Pythagorean Fuzzy Soft Set: A Case Study in ETL Techniques

This paper investigates the correlation coefficient for Interval-valued Pythagorean fuzzy soft set with their application to solve Multi-attribute decision making problem. Authors developed a novel TOPSIS technique using their developed correlation coefficient and used their developed technique in in Extract, Transform, and Load software selection for business intelligence. The presented technique in this research is very interesting and suitable for publication in PLOS One after some minor modifications.

1. Highlights the importance of your model in abstract.

2. The authors used some abbreviations without explain their proper meanings

3. Authors needs to update the Literature review in light of recent studies. Some Dombi Aggregation Operators Based on Complex q-Rung Orthopair Fuzzy Sets and Their Application to Multi-Attribute Decision Making. Complex Intuitionistic Fuzzy Aczel-Alsina Aggregation Operators and Their Application in Multi-Attribute Decision-Making, Multi-attribute decision-making methods based on Aczel–Alsina power aggregation operators for managing complex intuitionistic fuzzy sets, Power Aggregation Operators of Interval-Valued Atanassov-Intuitionistic Fuzzy Sets Based on Aczel–Alsina t-Norm and t-Conorm and Their Applications in Decision Making, complex T-spherical fuzzy sets.

4. Authors should be revised the manuscript and improve the typos, grammatical mistakes, and some presentation issues.

5. How your proposed model is good than other?

6. Comparison Analysis is the most important part of any manuscript, which is not affectively discussed in this manuscript. This issue should be resolved.

7. The authors should clearly discuss that how their presented work is the need of the time and it will be helpful to the scientific community?

6. PLOS authors have the option to publish the peer review history of their article (what does this mean?). If published, this will include your full peer review and any attached files.

Reviewer #1: No

Reviewer #2: No

---

## [Author Response · Author response to Decision Letter 0]

29 May 2023

Response and revisions Based on Comments of Reviewer 1

Dear editor, associate editor and reviewers:

Thanks for your letter and for the reviewers’ constructive comments concerning our manuscript entitled “Extension of Correlation Coefficient Based TOPSIS Technique for Interval-Valued Pythagorean Fuzzy Soft Set: A Case Study in ETL Techniques” These comments are all valuable and very helpful for improving our paper, as well as the important guiding significance to our research. We have studied the comments carefully and have made a correction which we hope to meet with approval. The reviewers’ comments are in italicized font below and numbered individually. Our response is given in normal font, and changes/additions to the manuscript are given in the yellow color text. Point-by-point responses to the two reviewers are listed as follows.

Reviewer Comments:

The authors perform an interesting concise analysis of a specific problem that concerns the popular TOPSIS technique for multi-attribute decision-making. The baseline model is Pythagorean fuzzy soft sets. The investigation is inspired by references [33] and [45]. The authors motivate their goals with a precise list of setbacks of the approaches presented by these two articles. In this context the problem is the measurement of “correlation” of the concept that arises when the observed membership and non-membership degrees sum up to more than one. As the authors explain, this element is not defined in the case of interval-valued Pythagorean fuzzy soft set (IVPFSS) data.

The article therefore improves the body of knowledge in an active area of research.: 

Response:

Dear respected Reviewer, 

We are thankful to you for your positive comments to improve our manuscript. We revised the manuscript according to the valuable suggestions and highlighted the changes with color (Yellow). Hopefully, our modifications in the revised manuscript will achieve your acceptance. Please see the revised version of the manuscript.

Comment 1. Define all elements properly. I recommend to avoid ETL in the title, or give it in its full form, as it is not common knowledge.

Response: Thanks for your good comment; we write the full form of ETL such as Extract, Transform, and Load in the title of the manuscript. Please see the title in the revised manuscript.

Comment 2. There are few recent references. There are 4 papers from 2021, and 3 of them are from authors of this article. And there are only 2 articles from 2022, both written by the first author of this submission. So the selection of literature seems to be too biased and not totally updated. To guarantee that the potential readers find the paper authoritative enough, the article needs to include more papers from reputed journals related to this topic in the summary of literature. Below are some possible references for your perusal (please consider these articles as mere suggestions): Risk evaluation in failure modes and effects analysis: hybrid TOPSIS and ELECTRE I solutions with Pythagorean fuzzy information. Neural Computing and Applications 33 (2021), 5675-5703. An integrated ELECTRE-I approach for risk evaluation with hesitant Pythagorean fuzzy information. Expert Systems with Applications 200 (2022), 116945. TOPSIS Approach for MAGDM Based on Interval-Valued Hesitant Fuzzy N-Soft Environment. International Journal of Fuzzy Systems 21 (2019), 993-1009. PF-TOPSIS method based on CPFRS models: An application to unconventional emergency events. Computers & Industrial Engineering 139 (2020), 106192. Ranked soft sets. Expert Systems (2023), e13231.

Response: Thanks for your good comment, we revised the suggested literature carefully and observed that suggested literature is very suitable for this research. So, we decisded to include these studies in our literature. Please see the references [24, 26, 27, 29, 35, 41] in the revised manuscript.

Comment 3. Also with respect to the review of literature, note that Pythagorean fuzzy sets had been defined by Prof. Atanassov with the name “type-2 intuitionistic fuzzy sets”. They became popular with Prof. Yager’s paper, but they were not new.

Response: Thanks for yourgood comment, yes you are absolutely right, the Pythagorean fuzzy set was developed by Prof. Atanassov with the name “type-2 intuitionistic fuzzy sets” and first time Pythagorean fuzzy sets presented the first real applications of Pythagorean fuzzy sets in decision-making in 2013 by Prof. Yager’s. but, mostly researchers considered the Pythagorean fuzzy sets from its utilization. Therefore, we also discussed the Pythagorean fuzzy set from the utilization of Prof. Yager’s.

Comment 4. In continuation with existing literature, it is also possible to use dissimilarity measures instead of correlation coefficients or similarity measures (pages 2-3). The discussion in “Improved generalized dissimilarity measure‐based VIKOR method for Pythagorean fuzzy sets”, International Journal of Intelligent Systems 37 (2022), 1807-1845, is quite illustrative for this purpose. 

Response: Thanks for your good comment, by using correlation coefficients we can find the most appropriate alternative using closeness coefficient which is the most suitable comparative dissimilarity and similarity measures. Please see the references [26] in the revised manuscript.

Comment 5. Table 9 is imperfect. The advantages of the “fuzzy soft” versions are identical to the “fuzzy” versions. 

Response: Thanks for your good comment; the advantages of fuzzy soft and fuzzy versions are looking identical. But, actually there is main difference among the fuzzy soft and fuzzy versions. The fuzzy version of cannot deals the parametric values of the alternatives. Meanwhile, fuzzy soft vrsion competently accommodate the parametric values of the alternatives.

Comment 6. Typos and minor suggestions.

Line 6 of Motivation (page 3): delete “upper”.

Def 2.7 uses IVPFSN but the concept has not been defined (this can be done easily in Def 2.6). Delete “and” in Def 2.7.

Figure 1, Step 1: why IT experts? the Figure should not illustrate the case study of Section 6.2 only.

In Table 9, the limitation described for “the proposed approach” seems to be incorrect (compare with its positive feature in line 7 under this table).

Response: Thanks for your valuable suggestions; we remove the word upper from line 6 of motivation section.

We define the concept of IVPFSN in definition 2.6 and also defined the score and accuracy functions in the Definition 2.6.

We replaced the “IT experts” with “Experts” in the step 1 of Figure 1.

We revised the limitations of the proposed model in Table 9. Please see the revised manuscript. 

Response and revisions Based on Comments of Reviewer 2

Reviewer Comments: 

This paper investigates the correlation coefficient for Interval-valued Pythagorean fuzzy soft set with their application to solve Multi-attribute decision making problem. Authors developed a novel TOPSIS technique using their developed correlation coefficient and used their developed technique in in Extract, Transform, and Load software selection for business intelligence. The presented technique in this research is very interesting and suitable for publication in PLOS One after some minor modifications.

Response:

Dear respected Reviewer, 

We are thankful to you for your positive comments to improve our manuscript. We revised the manuscript according to the valuable suggestions and highlighted the changes with color (Yellow). Hopefully, our modifications in the revised manuscript will achieve your acceptance. Please see the revised version of the manuscript.

Comment 1. Highlights the importance of your model in abstract.

Response: Thanks for your good comment; we highlighted the importance of our model in abstract very clearlyWe use the Extract, Transform, and Load (ETL) software selection as an example to demonstrate the application of these measures and construct a prioritization technique for order preference by similarity to the ideal solution (TOPSIS) model. The method investigates the challenge of optimizing ETL software selection for business intelligence (BI). This study offers to illuminate the significance of using correlation measures to make decisions in uncertain and complex settings. The multi-attribute decision-making (MADM) approach is a powerful instrument with many applications. This expansion is predicted to conclude in a more reliable decision-making structure. Using a sensitivity analysis, we contributed empirical studies to determine the most significant decision processes. . Please see the highlighted part of abstract in the revised manuscript.

Comment 2. The authors used some abbreviations without explain their proper meanings

Response: Thanks for your good comment; we revised the manuscript carefully and explained the missing abbreviation in a good manner. Please see the revised manuscript.

Comment 3. Authors needs to update the Literature review in light of recent studies. Some Dombi Aggregation Operators Based on Complex q-Rung Orthopair Fuzzy Sets and Their Application to Multi-Attribute Decision Making. Complex Intuitionistic Fuzzy Aczel-Alsina Aggregation Operators and Their Application in Multi-Attribute Decision-Making, Multi-attribute decision-making methods based on Aczel–Alsina power aggregation operators for managing complex intuitionistic fuzzy sets, Power Aggregation Operators of Interval-Valued Atanassov-Intuitionistic Fuzzy Sets Based on Aczel–Alsina t-Norm and t-Conorm and Their Applications in Decision Making, complex T-spherical fuzzy sets.

Response: Thanks for your good comment; we revised the suggested literature carefully and observed that some of the literature is very suitable for this study. So, we added some more literature in the introduction section. Please see the references [12, 13, 19] in the revised manuscript.

Comment 4. Authors should be revised the manuscript and improve the typos, grammatical mistakes, and some presentation issues. 

Response: Thank you so much for raising this question; we revised the whole manuscript and improved typos, grammatical mistakes, and some presentation issues. Please see the revised manuscript. 

Comment 5. How your proposed model is good than other?. 

Response: Thank you so much for the good question. The proposed approach is highly effective and substantial, offering a correlation-based TOPSIS technique to tackle MADM challenges in IVPFSS. Our system is more accurate than existing approaches and can easily handle MADM concerns. It is adaptable and versatile, allowing for different outcomes with criteria, responsibility, and adjustment variations. We modify the evaluation framework for different models with specific classification characteristics to align with their perspectives. Organizational studies and assessments reveal that our method's outcomes are equivalent to those of mixtures. Under specific conditions, numerous FS, IFS, and PFS amalgam configurations can be converted into IVPFSS. We can designate object-related information in a specific and tentative manner, as shown in Table 9, making it an appropriate tool for merging imprecise and rough facts in DM schedules. Our study and evaluation conclude that our projected scheme's outcomes are more comprehensive than any other approach. But, the DM process requires more attention to detail than our proposed DM techniques, as it involves vast data. Nevertheless, our planned approach is effective, flexible, and higher than other FS, IFS, and PFS amalgam conformations. Additionally, by adding appropriate relationships, various combinations of IVFS can be transformed into IVPFSS. The ability to add unconventional and ambiguous data to the current framework is remarkable. We can designate information about well-being thoroughly and realistically while accommodating fictional and unsettling data often encountered in the DM process. Therefore, our proposed scheme is better, significant, and exceptional, capable of accommodating various combinations of FS. Table 9 outlines the characteristics of both our proposed and existing methods.

Table 9: Qualitative comparison of the proposed model with the prevalent models

 Set Parameters Experts opinions in interval form Advantages Limitations

Zadeh [5] FS × × Deals uncertainty using MD Unable to handle NMD

Atanassov [9] IFS × × Deals uncertainty using MD and NMD Unable to handle MD+NMD>1

Yager [21] PFS × × Deals uncertainty using MD and NMD Unable to handle 〖MD〗^2+〖NMD〗^2>1

Turksen [6] IVFS × ✓ Deals uncertainty using MD intervals Unable to handle NMD interval

Atanassov [10] IVIFS × ✓ Deals uncertainty using MD and NMD intervals Unable to handle 〖MD〗^ひ+〖NMD〗^ひ>1

Peng & Yang [30] IVPFS × ✓ Deals uncertainty using MD and NMD intervals Unable to handle (〖MD〗^ひ )^2+(〖NMD〗^ひ )^2>1

Maji et al. [33] FSS ✓ × Deals uncertainty using MD Unable to handle NMD

Maji et al. [34] IFSS ✓ × Deals uncertainty using MD and NMD Unable to handle MD+NMD>1

Peng et al. [43] PFSS ✓ × Deals uncertainty using MD and NMD Unable to handle 〖MD〗^2+〖NMD〗^2>1

Jiang et al. [37] IVIFSS ✓ ✓ Deals uncertainty using MD and NMD intervals Unable to handle 〖MD〗^ひ+〖NMD〗^ひ>1

Proposed approach IVPFSS ✓ ✓ Deals uncertainty using MD and NMD intervals Unable to handle (〖MD〗^ひ )^2+(〖NMD〗^ひ )^2>1

It turns out that a new problem has arisen recently. This discussion aims to explain the rationale behind introducing a novel TOPSIS method designed to cater to the specific requirements of a particular organization. Although there are various existing methods, the proposed approach stands out due to its unique hybrid structure, which includes FS, IVFS, IFS, IVIFS, PFS, IVPFS, FSS, IVFSS, IFSS, IVIFSS, and PFSS. However, these hybrid structures have limitations in comprehensively analyzing the situation. To overcome these limitations, we have developed a correlation-based TOPSIS technique for IVPFSS that can handle attributes considering the MD and NMD in intervals such as 〖0≤(〖MD〗^ひ )〗^2+(〖NMD〗^ひ )^2≤1. This new approach is unique in its ability to provide a more detailed analysis of the situation compared to existing hybrid structures. Moreover, our proposed hybrid structure of FS is superior to other existing hybrid systems of FS, as presented in Table 9. Choosing the appropriate TOPSIS method is critical for an organization's success, and our novel approach provides a more comprehensive analysis of the situation, which is essential for making informed decisions. 

Comment 6. Comparison Analysis is the most important part of any manuscript, which is not affectively discussed in this manuscript. This issue should be resolved.

Response: Thanks for your good comment; the presented comparative analysis is enough. But we added some more discussion in the comparative analysis section. Please see the subsection 7.2 in the revised manuscript.

The validity of the correlation-based TOPSIS technique settled has been confirmed by previous research and comparative studies, which have shown that the results obtained using this method are consistent with those obtained from other available techniques. The key advantage of the proposed TOPSIS model, when combined with other DM methods, is its ability to incorporate additional information about the parameters of the alternatives, thereby addressing the imprecision present in the data. This results in a clearer and more empirical description of object-related facts, making it an effective tool for determining vague and inexplicable facts in the DM process. Furthermore, the comparative analysis highlights that the decision-making process in the proposed technique differs from previous methods due to the consideration of PIA and NIA based on CC inducement at a specific temporary level rather than distance and similarity measures. This avoids any potential loss of information during the process, which could occur when score values assigned to individual parameters fail to consider the impact on other parameters. The ideal correlation measure for each parameter is determined by assessing the most favorable outcomes and establishing the correlation between them. The premeditated TOPSIS method offers advantages over contemporary approaches and correlated measures by communicating the degree of perception and similarity among explanations, thereby avoiding conclusions based on negative foundations. Additionally, existing TOPSIS methods consider all alternatives, leading to the final judgment for selecting ETL software. However, several TOPSIS methodologies, such as those extended by Ansari et al. [7], Ashtiani et al. [8], Rouyendegh et al. [11], Zhang and Yu [18], Rani et al. [59], and Garg [60], have limitations in dealing with alternative parametrization and certain scenarios involving MD and NMD intervals. The TOPSIS scheme developed by Garg and Arora [36] and Zulqarnain et al. [40] under IFSS and IVIFSS is incapable of handling situations where MD+NMD>1 and 〖MD〗^ひ+〖NMD〗^ひ>1, respectively. Meanwhile, the TOPSIS method proposed in [53] handles parametrized values but neglects interval information, while the aggregation operators presented in [54] can tackle some drawbacks but fail to calculate the closeness coefficient in specific situations. Our proposed TOPSIS model can expertly address these concerns and is compared to existing methods, yielding similar results and demonstrating its consistency and efficiency in selecting the top 4 ETL software options (as shown in Table 10). Where n/a is the abbreviation of the not applicable.

Table 10: Comparative analysis of the proposed model with existing models under the considered data set

Structure Alternatives score values or closeness coefficient Ranking

Fuzzy TOPSIS [7] n/a n/a

IVFS TOPSIS [8] n/a n/a

IFS TOPSIS [11] n/a n/a

IVIFS TOPSIS [18] n/a n/a

PFS TOPSIS [59] n/a n/a

IVPFS TOPSIS [60] n/a n/a

IFSS TOPSIS [36] n/a n/a

IVIFSS TOPSIS [40] n/a n/a

PFSS TOPSIS [53] n/a n/a

IVPFSWA [54] Score(0.69271) Score(0.66247) Score(0.65816) Score(0.6882) L_1>L_4>L_2>L_3

IVPFSWG [54] Score(0.67352) Score(0.65461) Score(0.64152) Score(0.66517) L_1>L_4>L_2>L_3

Proposed TOPSIS 0.63836 0.51595 0.39879 0.47441 L_1>L_2>L_4>L_3

The evaluation matrix used to assess multiple TOPSIS techniques, and outcomes for each alternative are presented in Table 10. According to the table, the L_1 alternative is the ideal ETL software for BI. Table 10 provides an extensive summary of the proposed approach along with investigation. Table 10 indicates that previous works, such as those in [7, 8, 11, 18, 59, 60], lack details about parametric analysis. Meanwhile, the TOPSIS methods [36, 40, 53] are able to manage the parametrized values of the alternatives, however they are unable to deal with the under-consideration data set. Also, the IVPFSWA [54] and IVPFSWG [54] dealt with the considered data set, but these AOs unable to handle the closeness coefficient. The suggested approach, on the other hand, has the benefit of successfully addressing real-life challenges by acknowledging parametric aspects of the alternatives. As a result, the newly developed method can cope with DM problems which current operator in the IVPFSS environment cannot solve.

Comment 7. The authors should clearly discuss that how their presented work is the need of the time and it will be helpful to the scientific community?.

Response: Thank you so much for the constructive comment; Thanks for your good question. Our presented work is the most generalized form of interval valued intuitionistic fuzzy soft sets, interval valued Pythagorean fuzzy sets, and Pythagorean fuzzy soft sets. Our proposed correlation measures are very helpful for scientific community to compute the most suitable alternative in decision making problems using closeness coefficient comparative to other exisiting extensions of fuzzy sets.

---

## [Decision Letter · Decision Letter 1]

30 May 2023

Extension of Correlation Coefficient Based TOPSIS Technique for Interval-Valued Pythagorean Fuzzy Soft Set: A Case Study in Extract, Transform, and Load Techniques

PONE-D-23-09868R1

Dear Dr. Zulqarnain,

We’re pleased to inform you that your manuscript has been judged scientifically suitable for publication and will be formally accepted for publication once it meets all outstanding technical requirements.

Kind regards,

Muhammet Gul, Ph.D.

Academic Editor

PLOS ONE

Additional Editor Comments (optional):

Reviewers' comments:

Reviewer's Responses to Questions

**Comments to the Author**

1. If the authors have adequately addressed your comments raised in a previous round of review and you feel that this manuscript is now acceptable for publication, you may indicate that here to bypass the “Comments to the Author” section, enter your conflict of interest statement in the “Confidential to Editor” section, and submit your "Accept" recommendation.

Reviewer #1: All comments have been addressed

Reviewer #2: All comments have been addressed

2. Is the manuscript technically sound, and do the data support the conclusions?

Reviewer #1: Yes

Reviewer #2: Yes

3. Has the statistical analysis been performed appropriately and rigorously? 

Reviewer #1: N/A

Reviewer #2: Yes

4. Have the authors made all data underlying the findings in their manuscript fully available?

Reviewer #1: Yes

Reviewer #2: Yes

5. Is the manuscript presented in an intelligible fashion and written in standard English?

Reviewer #1: Yes

Reviewer #2: Yes

6. Review Comments to the Author

Reviewer #1: All the comments and suggestions that I made in my report have been addressed by the authors.

Now I have no further comments.

Reviewer #2: The authors have addressed all the comments very carefully and know the paper is ready to published in this journal PLOS ONE, therefor, my decision is accepted.

7. PLOS authors have the option to publish the peer review history of their article (what does this mean?). If published, this will include your full peer review and any attached files.

Reviewer #1: No

Reviewer #2: No

---

## [Editor Report · Acceptance letter]

16 Jun 2023

PONE-D-23-09868R1 

Extension of Correlation Coefficient Based TOPSIS Technique for Interval-Valued Pythagorean Fuzzy Soft Set: A Case Study in Extract, Transform, and Load Techniques 

Dear Dr. Zulqarnain:

I'm pleased to inform you that your manuscript has been deemed suitable for publication in PLOS ONE. Congratulations! Your manuscript is now with our production department. 

Kind regards, 

on behalf of

Dr. Muhammet Gul 

Academic Editor

PLOS ONE